# The neuronal calcium sensor Synaptotagmin-1 and SNARE proteins cooperate to dilate fusion pores

Zhenyong Wu[1,2], Nadiv Dharan[3†], Zachary A McDargh[3†], Sathish Thiyagarajan[3†], Ben O'Shaughnessy[3]*, Erdem Karatekin[1,2,4,5]*

[1]Department of Cellular and Molecular Physiology, Yale University, New Haven, United States; [2]Nanobiology Institute, Yale University, West Haven, United States; [3]Department of Chemical Engineering, Columbia University, New York, United States; [4]Department of Molecular Biophysics and Biochemistry, Yale University, New Haven, United States; [5]Saints-Pères Paris Institute for the Neurosciences (SPPIN), Université de Paris, Centre National de la Recherche Scientifique (CNRS) UMR 8003, Paris, France

**Abstract** All membrane fusion reactions proceed through an initial fusion pore, including calcium-triggered release of neurotransmitters and hormones. Expansion of this small pore to release cargo is energetically costly and regulated by cells, but the mechanisms are poorly understood. Here, we show that the neuronal/exocytic calcium sensor Synaptotagmin-1 (Syt1) promotes expansion of fusion pores induced by SNARE proteins. Pore dilation relied on calcium-induced insertion of the tandem C2 domain hydrophobic loops of Syt1 into the membrane, previously shown to reorient the C2 domain. Mathematical modelling suggests that C2B reorientation rotates a bound SNARE complex so that it exerts force on the membranes in a mechanical lever action that increases the height of the fusion pore, provoking pore dilation to offset the bending energy penalty. We conclude that Syt1 exerts novel non-local calcium-dependent mechanical forces on fusion pores that dilate pores and assist neurotransmitter and hormone release.

*For correspondence:
bo8@columbia.edu (BO'S);
erdem.karatekin@yale.edu (EK)

[†]These authors contributed equally to this work

**Competing interests:** The authors declare that no competing interests exist.

## Introduction

Release of neurotransmitters and hormones occurs through exocytosis in which neurotransmitter-filled synaptic vesicles or hormone-laden secretory vesicles fuse with the plasma membrane to release their cargo to the extracellular space (*Brunger et al., 2018a*). The initial merger of the vesicular and plasma membranes results in a narrow fusion pore only ~1 nm in diameter (*Karatekin, 2018*; *Sharma and Lindau, 2018*; *Chang et al., 2017*; *Alabi and Tsien, 2013*). Dynamics of this key intermediate determine release kinetics and the mode of vesicle recycling. The fusion pore can fluctuate in size, flicker open-closed multiple times and either reseal after partial release of contents or dilate for full cargo release. Because many endocrine cells co-package small and large cargoes, the pore can additionally act as a molecular sieve, controlling the type of cargo released. In pancreatic β-cells, fusion pores that fail to dilate release only small cargo such as ATP, but not insulin, a process that occurs more commonly in type 2 diabetes (*Collins et al., 2016*). Adrenal chromaffin cells release small catecholamines through flickering small pores, or release additional, larger cargo, in an activity-dependent manner (*Fulop et al., 2005*). Fusion pore dynamics also affect release of neurotransmitters and the mode of endocytosis during synaptic vesicle fusion (*Alabi and Tsien, 2013*; *He et al., 2006*; *Pawlu et al., 2004*; *Staal et al., 2004*; *Chapochnikov et al., 2014*; *Gandhi and Stevens, 2003*; *Lisman et al., 2007*; *Verstreken et al., 2002*).

Little is known about the molecular mechanisms that control pore dilation. SNARE proteins, a core component of the release machinery, are known to influence fusion pore dynamics (*Bao et al., 2018*; *Wu et al., 2016*; *Wu et al., 2017b*; *Wu et al., 2017a*; *Han et al., 2004*; *Bretou et al., 2008*; *Kesavan et al., 2007*; *Dhara et al., 2016*; *Ngatchou et al., 2010*). Formation of complexes between the vesicular v-SNARE VAMP2/Syb2 and plasma membrane t-SNAREs Syntaxin-1/SNAP25 is required for fusion (*Weber et al., 1998*). Insertion of flexible linkers between the SNARE domain and the transmembrane domain in VAMP2, or truncation of the last nine residues of SNAP25, retard fusion pore expansion in adrenal chromaffin cells (*Bretou et al., 2008*; *Kesavan et al., 2007*; *Fang et al., 2008*). Mutations in SNARE TMDs also affect fusion pores (*Wu et al., 2017a*). Increasing the number of SNAREs at the fusion site accelerated fusion pore expansion in neurons (*Bao et al., 2018*; *Acuna et al., 2014*), astrocytes (*Guček et al., 2016*), and chromaffin cells (*Zhao et al., 2013*) and led to larger pores in nanodisc-based single-pore fusion assays (*Bao et al., 2018*; *Wu et al., 2017b*). This was interpreted as due to increased molecular crowding at the waist of the pore with increasing SNARE copy numbers (*Wu et al., 2017b*).

Although they are best known for their role as calcium sensors for exocytosis at most synapses and endocrine cells, Synaptotagmins are another component of the release machinery known to affect fusion pore properties (*Segovia et al., 2010*; *Zhang et al., 2010a*; *Zhang et al., 2010b*; *Wang et al., 2006*; *Wang et al., 2003a*; *Wang et al., 2003b*; *Wang et al., 2001*; *Bai et al., 2004a*; *Lynch et al., 2008*; *Rao et al., 2014*; *Lai et al., 2013*). They couple membrane fusion driven by neuronal/exocytic SNAREs to calcium influx (*Geppert et al., 1994*; *Chapman, 2008*). Synaptotagmins are integral membrane proteins possessing two cytosolic C2 domains (C2A and C2B) which can bind $Ca^{2+}$, acidic lipids, SNAREs, and other effectors, but affinities vary widely among the 17 mammalian isoforms (*Chapman, 2008*; *Bhalla et al., 2005*; *Bhalla et al., 2008*; *Pinheiro et al., 2016*; *Volynski and Krishnakumar, 2018*; *Craxton, 2010*; *Sugita et al., 2002*; *Hui et al., 2005*). Synaptotagmin-1 (Syt1) is the major neuronal isoform that mediates fast, synchronous neurotransmitter release (*Chapman, 2008*; *Volynski and Krishnakumar, 2018*; *Xu et al., 2007*). It resides in synaptic vesicles in neurons and secretory granules in neuroendocrine cells and interacts with SNAREs, acidic phospholipids, and calcium (*Brunger et al., 2018a*; *Chapman, 2008*; *Brunger et al., 2018b*; *Südhof, 2013*). How calcium binding to Syt1 leads to the opening of a fusion pore is an area of active research and debate (*Lynch et al., 2008*; *Brunger et al., 2018b*; *Martens et al., 2007*; *Hui et al., 2009*; *Rothman et al., 2017*; *Chang et al., 2018*; *van den Bogaart et al., 2011*; *Seven et al., 2013*; *Lin et al., 2014*; *Bello et al., 2018*; *Tagliatti et al., 2020*). In addition to its role in triggering the opening of a fusion pore, Syt1 also affects the expansion of the fusion pore after it has formed (*Segovia et al., 2010*; *Zhang et al., 2010a*; *Zhang et al., 2010b*; *Wang et al., 2006*; *Wang et al., 2003a*; *Wang et al., 2003b*; *Wang et al., 2001*; *Bai et al., 2004a*; *Lynch et al., 2008*; *Rao et al., 2014*; *Lai et al., 2013*), but mechanisms are even less clear.

Calcium-binding to Syt1 causes hydrophobic residues at the tips of the $Ca^{2+}$-binding loops to insert into the membrane, generating curvature, which may be important for triggering fusion (*Lynch et al., 2008*; *Martens et al., 2007*; *Hui et al., 2009*). Membrane bending has been proposed to facilitate opening of the initial fusion pore by helping to bring the two membranes into close proximity, reducing the repulsive hydration forces by reducing the contact area, and exposing the hydrophobic interior of the two membranes to initiate lipid exchange (*Chernomordik and Kozlov, 2008*; *Kozlov et al., 2010*). After fusion pore opening, Syt1 was suggested to contribute to fusion pore expansion through membrane curvature generation as well, based on the observation that in PC12 cells, membrane-insertion deficient mutants reduced exocytosis, whereas mutants with enhanced insertion led to larger fusion pores (*Lynch et al., 2008*). However, once the initial fusion pore is formed it is not clear whether and how much curvature generation by Syt1 contributes to fusion pore expansion. First, in PC12 cells multiple Syt isoforms reside on the same secretory granule and potentially compete for fusion activity (*Zhang et al., 2011*; *Lynch and Martin, 2007*). Disrupting Syt1 function may allow another isoform to dominate fusion pore dynamics. In adrenal chromaffin cells where Syt1 and Syt7 are sorted to distinct granule populations, fusion pores of Syt7 granules dilate more slowly (*Rao et al., 2014*). Second, compared to Syt1 C2 domains, the higher calcium-affinity Syt7 C2 domains penetrate more avidly and deeply into membranes (*Osterberg et al., 2015*; *Voleti et al., 2017*), which should lead to more efficient membrane bending (*Martens et al., 2007*; *Hui et al., 2009*). This would appear to be inconsistent with the slower dilation of fusion pores

by Syt7. Finally, most previous reconstitutions could not probe the role of Syt1 in fusion pore regulation, as they lacked the required sensitivity and time resolution to detect single pores.

Here, we investigated the mechanism by which Syt1 contributes to fusion pore dynamics, using a single-pore conductance assay (*Wu et al., 2016*; *Wu et al., 2017b*). Compared to SNAREs alone, addition of Syt1 increased the mean pore conductance three-fold. This effect required binding of Syt1 to calcium, the acidic phospholipid PI(4,5)P$_2$, and likely to the SNAREs. In addition, both pore opening and dilation are promoted by insertion of Syt1 C2AB top loops into the membrane in a Ca$^{2+}$-dependent manner, but we propose that membrane curvature generation is not needed to explain fusion pore expansion by Syt1. Mathematical modeling suggests that pore dilation relies on regulation of the intermembrane distance by Syt1. Syt1 penetration into the target membrane upon calcium binding re-orients the C2AB domains and SNARE complexes, forcing the membranes apart in a lever-like action that concomitantly expands the pore.

## Results

### Co-reconstitution of Synaptotagmin-1 and v-SNAREs into nanolipoprotein particles

Previously, using a nanodisc-cell fusion assay, we characterized single, SNARE-induced fusion pores connecting a nanodisc and an engineered cell expressing neuronal 'flipped' t-SNAREs ectopically (*Wu et al., 2016*; *Wu et al., 2017b*). In this assay, a flipped t-SNARE cell is voltage-clamped in the cell-attached configuration. Nanodiscs reconstituted with the neuronal/exocytotic v-SNARE VAMP2 are included in the pipette solution. Fusion of a nanodisc with the cell surface creates a nanometer size pore that connects the cytosol to the exterior, allowing passage of ions under voltage clamp. Direct currents report pore size with sub-millisecond time resolution (*Wu et al., 2016*; *Wu et al., 2017b*). Fusion pore currents fluctuate and may return to baseline transiently multiple times, evidently reflecting pore flickering (*Karatekin, 2018*; *Wu et al., 2016*; *Wu et al., 2017b*; *Dudzinski et al., 2019*). Pore conductance is eventually lost (5–20 s on average after initial appearance), evidently reflecting pore closure (*Karatekin, 2018*; *Wu et al., 2016*; *Wu et al., 2017b*; *Dudzinski et al., 2019*). The mechanism of pore closure is not known, but because pore expansion beyond a maximum size is prevented by the nanodisc scaffold, pore closure is one of the few possible outcomes (*Wu et al., 2016*; *Shi et al., 2012*). To ensure single-pore detection, the rate at which pore currents appear (reported in pores/min, also referred to as the 'fusion rate') is made low by recording from a small area of the cell surface and by tuning the nanodisc concentration (*Karatekin, 2018*; *Wu et al., 2016*; *Wu et al., 2017b*; *Dudzinski et al., 2019*) (see Materials and methods and Appendix 1 for details).

To test whether Syt1 affected fusion pores in this system, we co-reconstituted ~4 copies of recombinant full-length Syt1 together with ~4 copies of VAMP2 (per disc face) into large nanodiscs called nanolipoprotein particles (*Wu et al., 2017b*; *Bello et al., 2016*) (vsNLPs, ~25 nm in diameter, see *Appendix 1—figure 1*). We reasoned that, under these conditions, potential modification of pore properties by Syt1 should be detectable. In the absence of Syt1, we previously found that only ~2 SNARE complexes are sufficient to open a small fusion pore (150-200 pS conductance), but dilation of the pore beyond ~1 nS conductance (~1.7 nm in radius, assuming the pore is a 15 nm long cylinder [*Hille, 2001*]) required the cooperative action of more than ~10 SNARE complexes (*Wu et al., 2017b*). The increase in pore size was heterogeneous with increasing SNARE load; most pores remained small (mean conductance ≤ 1 nS), but an increasing fraction had much larger conductances of a few nS. With ~4 v-SNAREs per NLP face, fusion driven by SNAREs alone results in relatively small pores with ~200 pS average conductance, corresponding to a pore radius of ~0.76 nm (*Wu et al., 2017b*). Larger pores (mean conductance > 1 nS) were rare (< 5%, [*Wu et al., 2017b*]). With ~25 nm NLPs, a fusion pore can in principle grow to > 10 nm diameter (~9 nS conductance) before the scaffold protein stabilizing the edges of the NLP becomes a limitation for further pore dilation (*Wu et al., 2017b*; *Bello et al., 2016*). Thus, at this v-SNARE density, there is a large latitude in pore sizes that can be accommodated, if introduction of Syt1 were to lead to any modification.

We tuned NLP size by varying the lipid-to-scaffold protein (ApoE422k) ratio and adjusted copy numbers of VAMP2 and Syt1 until we obtained the target value of ~4 copies of each per NLP face, similar to previous work with SNAREs alone (*Wu et al., 2017b*; *Bello et al., 2016*). vsNLPs were

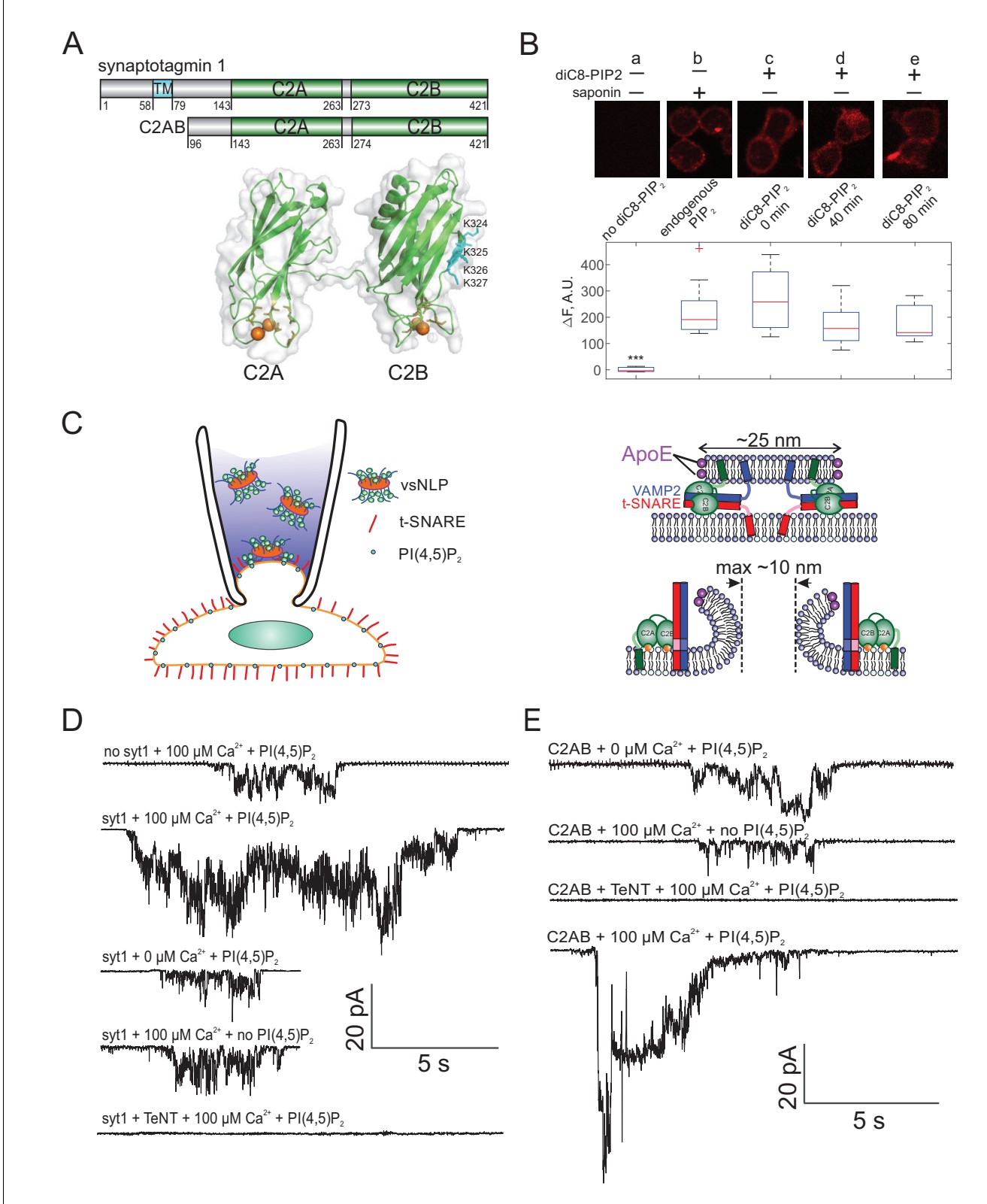

**Figure 1.** Detection of single fusion pore currents mediated by Syt1 or its C2AB domain. (A) Domain structures of the constructs used in this study. The structure of the soluble C2AB domains was rendered using PyMol, from PDB: 5kj7 (*Lyubimov et al., 2016*). The orientations of the C2A and C2B domains relative to each other are not known in the presence of SNAREs and membranes. Conserved aspartate residues coordinating calcium ions are depicted in orange. Calcium ions are shown as orange spheres. A poly-lysine motif on the side of C2B (K324,K325,K326,K327 in the rat sequence) that

*Figure 1 continued on next page*

Figure 1 continued

preferentially interacts with PI(4,5)P$_2$ (**Radhakrishnan et al., 2009**) is highlighted in cyan. (B) Incorporation of exogenous PI(4,5)P$_2$ into the outer leaflet of flipped t-SNARE cells. Top: cells were incubated with diC8-PI(4,5)P$_2$ for 20 min, rinsed, and immunolabeled for PI(4,5)P$_2$ at the indicated time points. Only control cells that were permeabilized with saponin showed immunostaining, confirming absence of PI(4,5)P$_2$ in the outer leaflet, and providing a reference value for inner-leaflet PI(4,5)P$_2$ levels (**a** and **b**). Cells incubated with diC8-PI(4,5)P$_2$ showed immunofluorescence in the absence of permeabilization, indicating successful incorporation of PI(4,5)P$_2$ into the outer leaflet of the cell membrane (**c–e**). The signal was comparable to endogenous inner-leaflet PI(4,5)P$_2$ levels, and persisted at least for 80 min (lower panel). Cells processed similarly, but not treated with saponin or diC8-PI(4,5)P$_2$ served as negative controls (**a**). One-way analysis of variance (ANOVA) followed by multiple comparison test was used to compare the signals from the endogenous PI(4,5)P$_2$ sample (**b**) with all others. *, **, *** indicate p<0.05, 0.01, and 0.001, respectively. (C) Schematic of the single-pore nanodisc-cell fusion assay. A glass micropipette forms a tight seal on a patch of the plasma membrane of a cell expressing 'flipped' t-SNARE proteins on its surface. NLPs co-reconstituted with Syt1 and VAMP2 are included in the pipette solution (left). NLP-cell fusion results in a fusion pore connecting the cytosol to the cell's exterior (right). Under voltage clamp, direct-currents passing through the pore report pore dynamics. With ~25 nm NLPs, the scaffolding ring does not hinder pore expansion up to at least 10 nm diameter. Exogenous PI(4,5)P$_2$ can be added to the cell's outer leaflet as in B, and calcium in the pipette is controlled using calcium buffers. (D) Representative currents that were recorded during vsNLP-tCell fusion, for the indicated conditions. PI(4,5)P$_2$ indicates cells were pre-treated with diC8-PI(4,5)P$_2$. Tetanus neurotoxin (TeNT) light chain cleaves VAMP2 and blocks exocytosis. Currents were larger when all components were present (SNAREs, Syt1, exogenous PI(4,5)P$_2$ and calcium). (E) Similar to D, but instead of full-length Syt1, 10 μM soluble Syt1 C2AB domains were used together with NLPs carrying ~4 copies of VAMP2 per face.

purified by size exclusion chromatography and characterized by SDS-PAGE and transmission electron microscopy (see **Appendix 1—figure 1B-D**). The distribution of NLP diameters was fairly narrow, with mean diameter 25 nm (±5.6 nm SD, see **Appendix 1—figure 1E**), and did not change significantly compared to the distribution when v-SNAREs alone were incorporated at ~4 copies per face (mean diameter = 25 ± 4 nm) (see **Appendix 1—figure 1F-H**, and **Wu et al., 2017b**).

## Syt1 promotes fusion pore expansion

To probe fusion pores, we voltage-clamped a flipped t-SNARE cell in the cell-attached configuration and included NLPs co-loaded with Syt1 and VAMP2 in the pipette solution as shown in **Figure 1** (100 nM vsNLPs, 120 μM lipid). Even in the presence of 100 μM free calcium, a level that elicits robust release in neurons and chromaffin cells (**Pinheiro et al., 2016**; **Schneggenburger and Neher, 2000**; **Schneggenburger and Neher, 2005**; **Voets, 2000**; **Chanaday and Kavalali, 2018**), pore properties were affected only slightly compared to the case when Syt1 was omitted from the NLPs. For example, pore currents (**Figure 1D**) appeared at similar frequency (**Figure 2A**) and the mean single-pore conductance, $\langle G_{po} \rangle$, was only slightly elevated in the presence of Syt1 (See **Appendix 1—figure 1J**, **Appendix 1—figure 2**, and Appendix 1 Supplementary Materials and methods for definitions and other pore parameters). We wondered whether the lack of acidic lipids in the outer leaflet of the cell membrane could be a limitation for Syt1's ability to modulate fusion pores. Syt1 is known to interact with acidic lipids, in particular with PI(4,5)P$_2$, in both calcium-dependent and independent manners, and these interactions are required for Syt1's ability to trigger membrane fusion (**Zhang et al., 2010a**; **Chang et al., 2018**; **Ma et al., 2017**; **Pérez-Lara et al., 2016**; **Honigmann et al., 2013**; **Bai et al., 2004b**). However, the outer leaflet of the plasma membrane which is seen by Syt1 in our assay is poor in such lipids. To test for a requirement for PI(4,5)P$_2$, we incubated flipped t-SNARE cells with 20 μM diC$_8$PI(4,5)P$_2$ for 20 min and rinsed excess exogenous lipid. At different time points after rinsing, we probed incorporation of the short-chain PI(4,5)P$_2$ into the outer leaflet of the cell membrane by immunofluorescence, using a mouse monoclonal anti-PI(4,5)P$_2$ primary antibody, without permeabilizing the cells (**Figure 1B**). The signal decreased slightly as a function of time but persisted for at least 80 min. To compare the level of short-chain PI(4,5)P$_2$ incorporated into the outer leaflet in this manner with endogenous PI(4,5)P$_2$ levels in the inner leaflet, we measured immunofluorescence from permeabilized cells that were not incubated with diC$_8$PI(4,5)P$_2$. Outer leaflet diC$_8$PI(4,5)P$_2$ levels were within 25% of the endogenous inner-leaflet PI(4,5)P$_2$ levels (**Figure 1B**).

When we repeated vsNLP-flipped t-SNARE cell fusion experiments with cells pre-incubated with diC$_8$PI(4,5)P$_2$, the rate of fusion in the absence of calcium was unchanged compared to fusion with SNAREs alone, but increased three- to fourfold when 100 μM calcium was present (**Figure 2A**). Note that our fusion rate estimates throughout should be interpreted with caution, because they are inherently noisy and they systematically underestimate fusion rates when the rates are high. Both effects are due to the fact that in the assay only a few fusion pores can be analyzed per patch (see

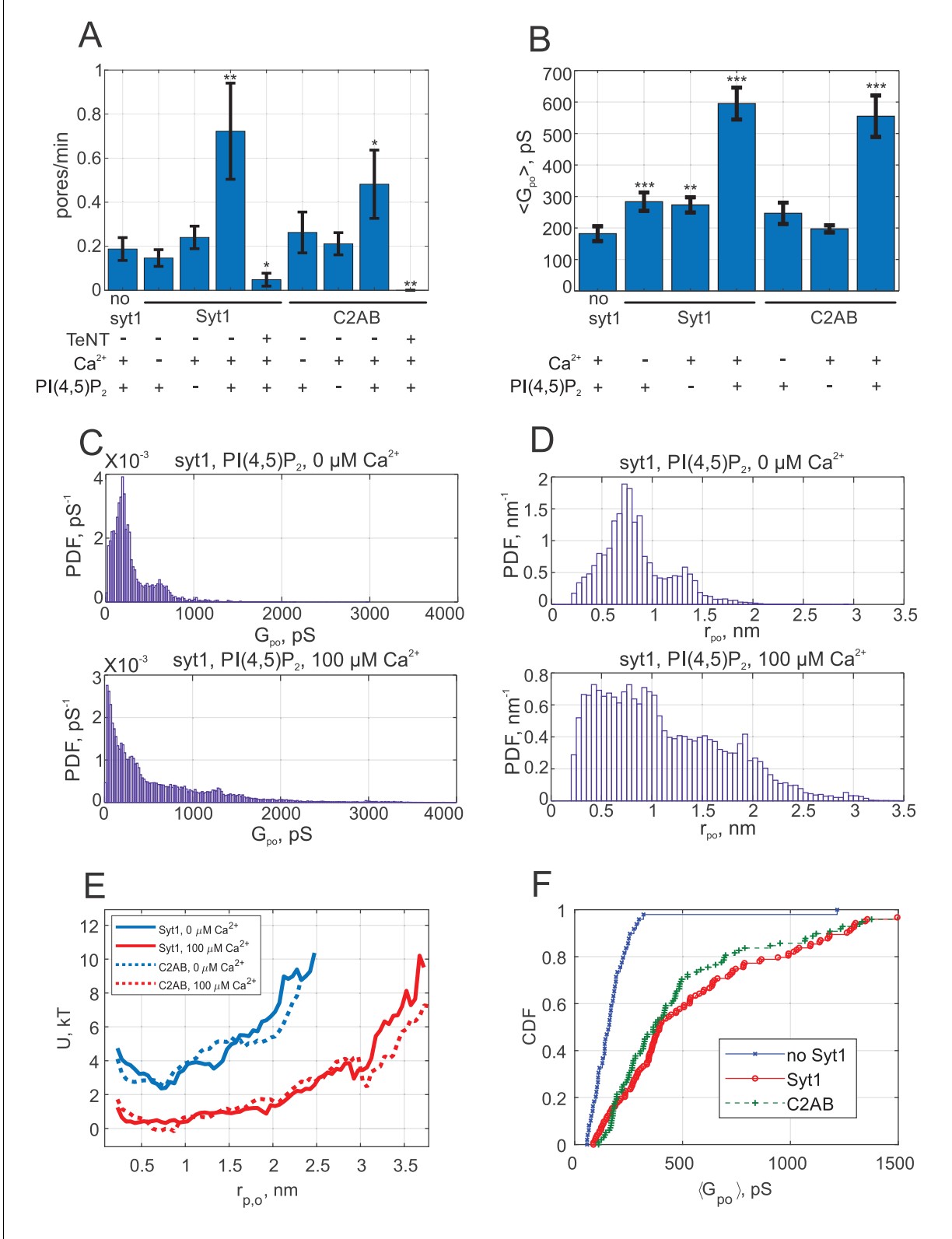

**Figure 2.** Syt1 promotes fusion and expands fusion pores in a calcium and PI(4,5)P$_2$ dependent manner, and soluble Syt1 C2AB largely recapitulates these effects. (**A**) The rate at which current bursts appeared (pore nucleation rate) for the conditions indicated (error bars represent ± S.E.M.). SNARE-induced pores appeared more frequently in the presence of Syt1 or C2AB, when both calcium and PI(4,5)P$_2$ were also present. Student's t-test (one-tailed) was used to assess significant differences between the 'no Syt1' group and the rest. *, **, *** indicate p<0.05, 0.01, and 0.001, respectively.
*Figure 2 continued on next page*

*Figure 2 continued*

There is no difference between the Syt1 and C2AB groups in the presence of calcium and exogenous PI(4,5)P$_2$ (Student's t-test: $p = 0.18$). (B) Mean single fusion pore conductance, $\langle G_{po} \rangle$, for different conditions as indicated ($\pm$ S.E.M.). $\langle G_{po} \rangle$ was three-fold larger in the presence of Syt1 or C2AB, when both calcium and PI(4,5)P$_2$ were also present. Two-sample Kolmogorov-Smirnov test was used to assess significant differences between the 'no Syt1' group and the rest. The same asterisk notation as in A was used. There is no difference between the Syt1 and C2AB groups in the presence of calcium and exogenous PI(4,5)P$_2$ (two-sample Kolmogorov-Smirnov test: $p = 0.29$). (C) Probability density functions (PDFs) for point-by-point open-pore conductances (see Materials and methods) for pores induced in the presence of Syt1, PI(4,5)P$_2$ and with 0 or 100 µM calcium. Notice the higher density at larger conductance values in the presence of 100 µM calcium. (D) Probability density functions for pore radii, calculated from the conductance PDFs in C, assuming a 15-nm long cylindrical pore (*Hille, 2001*). (E) Apparent free energy profiles for Syt1 and soluble Syt1 C2AB domains in the absence or presence of calcium. These profiles were calculated from the pore radii PDFs as in D (see text and Materials and methods) (*Wu et al., 2017b*). The profiles were shifted vertically for clarity. (F) Cumulative density functions (CDFs) for mean single-pore conductances for the conditions indicated. Soluble C2AB recapitulated effects of full-length Syt1 co-reconstituted into NLPs.

Appendix 1- Materials and methods and *Karatekin, 2018*). Compared to SNARE-alone fusion, the mean single-pore conductance increased only slightly in the absence of calcium but was three-fold larger in the presence of 100 µM calcium (*Figure 2B*). Conductance fluctuations around the mean value were larger and flicker frequency lower when Syt1, calcium and PI(4,5)P$_2$ were all present, but no major differences emerged for burst lifetimes, $T_o$, or pore open probability during a burst (the fraction of time the pore was open during a burst), $P_o$ (see *Appendix 1—figure 3*). For all cases tested, the distributions of the number of pore flickers ($N_{flickers}$) and burst durations ($T_o$) were well-described by geometric and exponential distributions, respectively (see *Appendix 1—figure 3*), as would be expected for discrete transitions between open, transiently blocked, and closed states (*Colquhoun and Hawkes, 1995*). Fusion was SNARE-dependent, as treatment with the tetanus neu-rotoxin TeNT, which cleaves VAMP2 at position 76Q-77F and blocks exocytosis (*Schiavo et al., 2000*), dramatically reduced the fusion rate of vsNLPs even in the presence of calcium and exoge-nous PI(4,5)P$_2$ (*Figure 1D* and *Figure 2A*). Thus, Syt1 increases the fusion rate and promotes pore dilation during SNARE-induced fusion, in a calcium- and PI(4,5)P$_2$-dependent manner.

We pooled individual current bursts to obtain the distributions for fusion pore conductances and pore radii as shown in *Figure 2C,D*, and *Appendix 1—figure 4*. The distributions were similar for SNAREs alone, whether calcium or PI(4,5)P$_2$ were added, and with Syt1 when calcium was omitted (*Figure 2C,D*, and see *Appendix 1—figure 4*). By contrast, in the presence of 100 µM free calcium and exogenous PI(4,5)P$_2$, larger conductance values (and corresponding pore radii) became more likely (*Figure 2C,D*).

Even when pores were maximally dilated by Syt1, the mean conductance and pore radius, $G_{po} = 595$ pS (S.E.M. = 51 pS), and $r_{po} = 1.13$ nm (S.E.M = 0.04 nm) were significantly less than the maximum possible value predicted from NLP dimensions (*Wu et al., 2017b*). That is, the geometric constraints imposed by the NLP dimensions were not limiting pore expansion. Instead, there is inher-ent resistance to pore dilation, independent of NLP scaffolding (*Wu et al., 2017b*) as predicted and observed in other systems (*Jackson, 2009*; *Cohen and Melikyan, 2004*; *D'Agostino et al., 2018*). To quantify the resistance, we computed (*Wu et al., 2017b*) the apparent pore free energy $U(r_{po})$ from the distribution of pore radii, $P(r_{po}) \sim e^{-U(r_{po})/kT}$ for fusion with both SNAREs alone and with Syt1 under optimal conditions (with exogenous PI(4,5)P$_2$ and 100 µM free calcium). Invoking the Boltzmann distribution amounts to assuming the membrane-protein system is approximately in equi-librium, that is conductance measurements are approximately passive and only weakly perturb the fusion pore. We cannot exclude substantial non-equilibrium effects, as application of a potential dif-ference may in itself promote pore formation and affect the structure and dynamics of the pores that result, as seen in lipid bilayer electroporation studies (*Melikov et al., 2001*), although the potential difference used in our studies is much lower (<20 mV). Generally, the profiles we report should be interpreted as effective free energies. With SNAREs alone, or with Syt1 but in the absence of calcium, the free energy profile suggested that ~6-7 kT energy was required to expand the pore from 1 to ~2.5 nm radius, whereas calcium-bound Syt1 reduced this resistance to ~2 kT (*Figure 2E*). That is, the force opposing pore expansion decreased from 16-19 pN in the absence of calcium to ~5 pN in the presence of 100 µM calcium.

We tested if the soluble C2AB domains of Syt1 could recapitulate these results. We included 10 µM C2AB together with NLPs reconstituted with ~4 copies per face of VAMP2 in the patch pipette

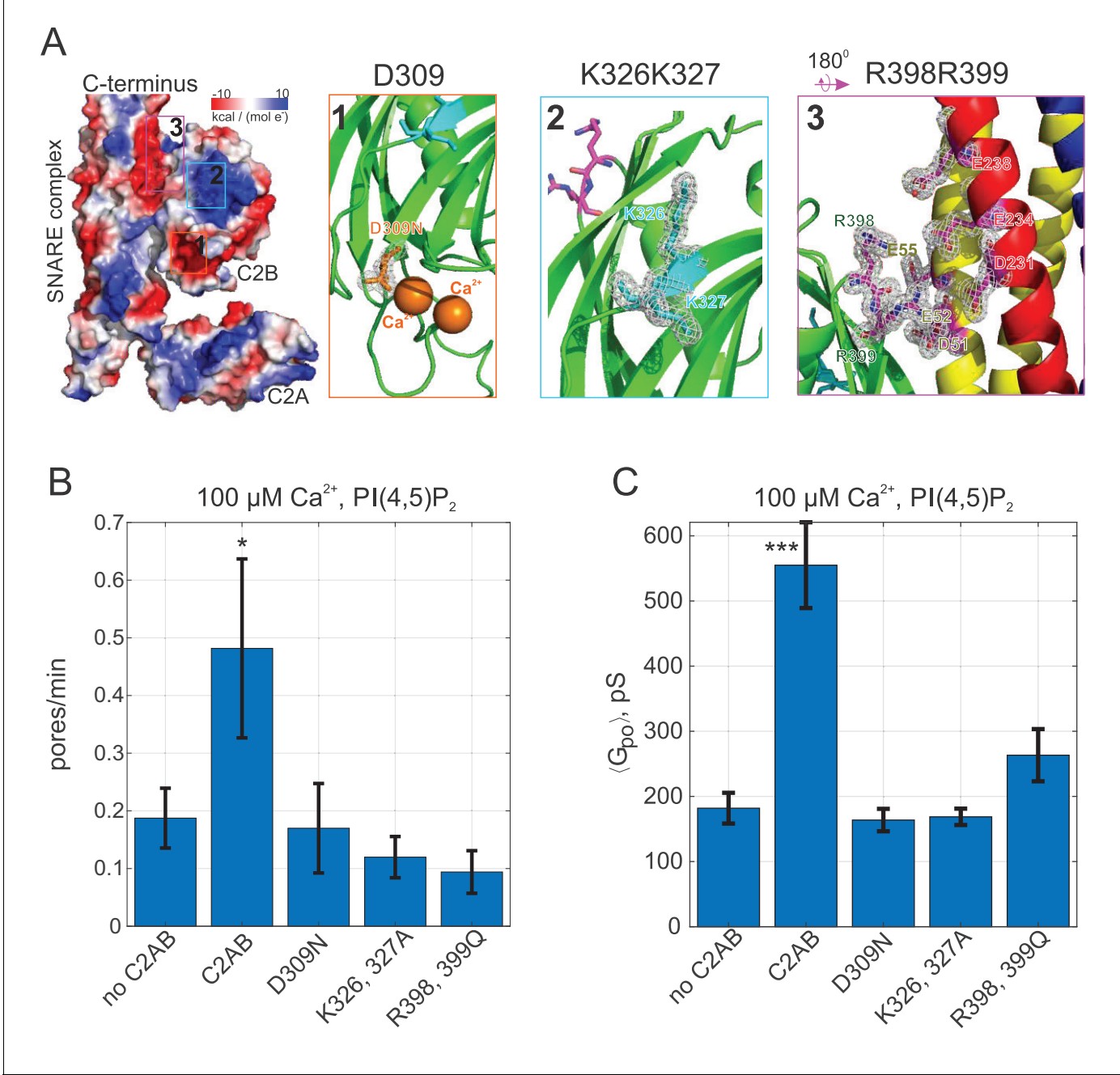

**Figure 3.** Pore expansion by Syt1 C2AB requires calcium, PI(4,5)P$_2$, and putative SNARE binding sites to be intact. (**A**) Overview of the Syt1-SNARE complex (*Lyubimov et al., 2016*). The electrostatic potential of PDB 5kj7 (*Lyubimov et al., 2016*) was rendered using Pymol. The sites mutated in this work are marked by boxes labeled 1–3 on the left and shown in the panels to the right. D309 is a key calcium-binding residue (1), K326, K327 interact with acidic lipids (2), and R398,R399 (3) interact with the t-SNAREs SNAP 25 (E51, E52, and E55) and syntaxin 1A (D231, E234, and E238). VAMP2 is shown in blue, SNAP25 in yellow, and syntaxin 1A in red. (**B**) Pore nucleation rates (+/- SEM) for the indicated conditions. All conditions included 100 µM free calcium and pre-incubation of tCells with exogenous PI(4,5)P$_2$. Pores appeared two to three times less frequently with the mutated proteins compared to wild-type Syt1 C2AB. Student's t-test was used to assess significant differences between the 'no C2AB' group and the rest. (**C**) Mean single open-pore conductance values (± SEM) for the same conditions as in B. Disrupting binding to calcium (D309N), acidic lipids (K326A, K327A), or the SNARE complex (R398, R399) resulted in ~3-fold smaller mean conductance compared to wild-type C2AB, abrogating the effects of Syt1 C2AB. Two-sample Kolmogorov-Smirnov test was used to assess significant differences between the 'no C2AB' group and the rest. *, **, *** indicate $p<0.05$, 0.01, and 0.001, respectively.

and monitored fusion with flipped t-SNARE cells in the cell attached configuration under voltage clamp. Similar to the results with full-length Syt1, there was little change in the fusion rate compared to the SNARE-alone case if either calcium or exogenous PI(4,5)P₂ was omitted (*Figure 2A*). When both calcium (100 μM) and PI(4,5)P₂ were present, the fusion rate was higher, but we are not as confident about this increase as in the case of Syt1. The mean conductance was significantly above the SNARE-only value in the presence of calcium and PI(4,5)P₂, but not when either was omitted (*Figure 2B*). The distributions of average single pore conductances (*Figure 2F*), conductance fluctuations, and other pore parameters were similar whether full-length Syt1 or soluble C2AB were used, except $P_0$ was higher for the +Ca²⁺/+PI(4,5)P₂ case and $T_0$ lower for +Ca²⁺/-PI(4,5)P₂ case for C2AB compared to Syt1 (Figs. S3 and S4). The apparent free energy profile calculated from the pore size distribution was indistinguishable from that of full-length Syt1 (*Figure 2E*). We conclude that soluble Syt1 C2AB largely recapitulates the effect of full-length Syt1 on promoting dilation of SNARE-mediated fusion pores. As they were far easier to manipulate, we used soluble Syt1 C2AB domains for the remainder of this work.

In some cases, a peak at ~200 pS is apparent in open-pore conductance distributions, corresponding to a peak at $r_{po} \approx 0.7$ nm in pore size distributions (e.g. see *Appendix 1—figure 4*). This is

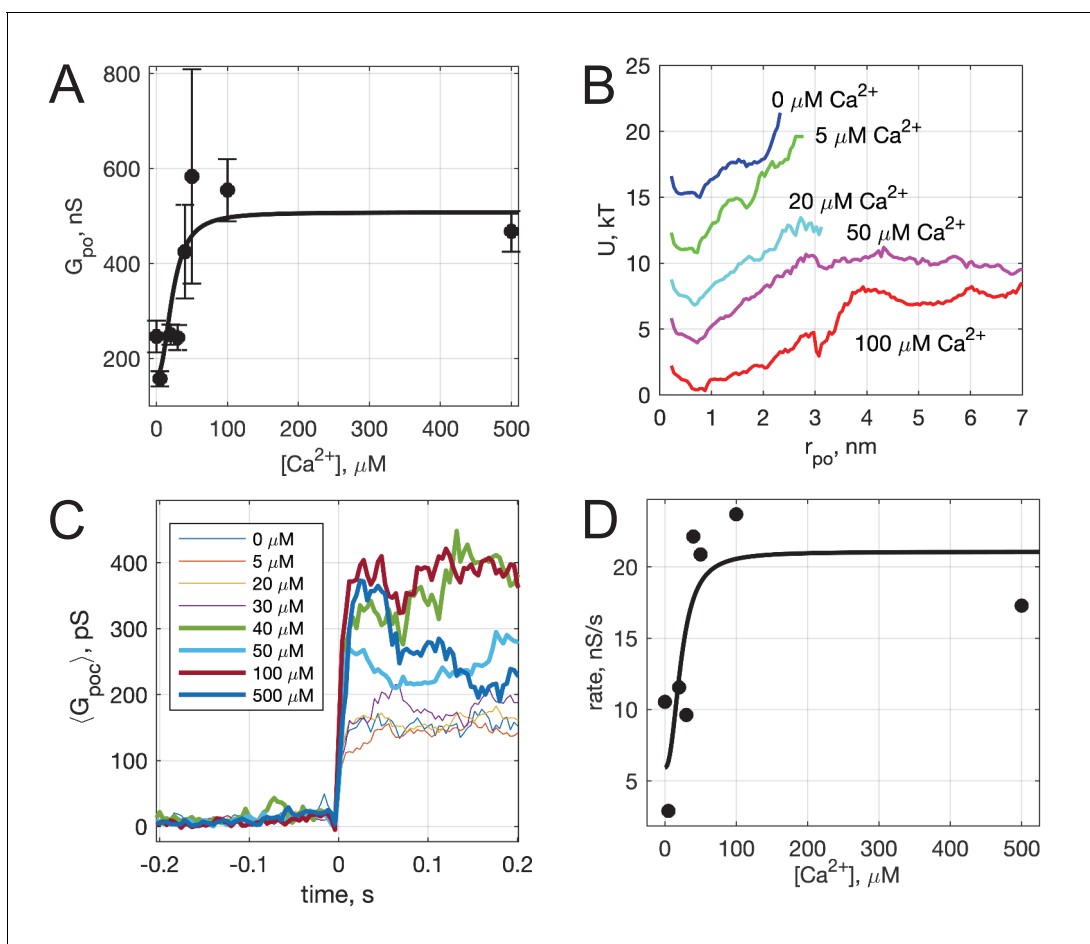

**Figure 4.** Calcium-dependence of pore properties. (A) Mean single open-pore conductance, $\langle G_{po} \rangle$, as a function of free calcium concentration in the pipette solution. Plotted values are mean ± S.E.M. A fit to a Hill equation $f(x) = \frac{a}{\left(\frac{K}{x}\right)^n + 1} + c$ is shown as the black line, where $x = [\text{Ca}^{2+}]_{free}$, $n = 2.3$, and $K = 23$ μM (see text). Best fit parameters (with 95% confidence bounds) were $a = 343.7 (128.5, 558.8)$, $c = 164.2 (14.5, 314)$, and $R^2 = 0.72$. (B) Apparent free energy profiles, calculated as in *Figure 2E*, for different calcium concentrations. (C) Kinetics of pore expansion for different $[\text{Ca}^{2+}]_{free}$ as indicated. Conductance traces were aligned to the first point in a pore and averaged. (D) Expansion rates of time-aligned and averaged conductances as a function of $[\text{Ca}^{2+}]_{free}$. Expansion rates were calculated as the 10-90% rise time from the baseline to the level of conductance reached within the first 100 ms after pore opening, divided by the time it took for this rise (see Appendix 1, Supplementary Materials and methods). A fit to a Hill equation as in A is also shown, using the same $x$ and $n$ parameter values.

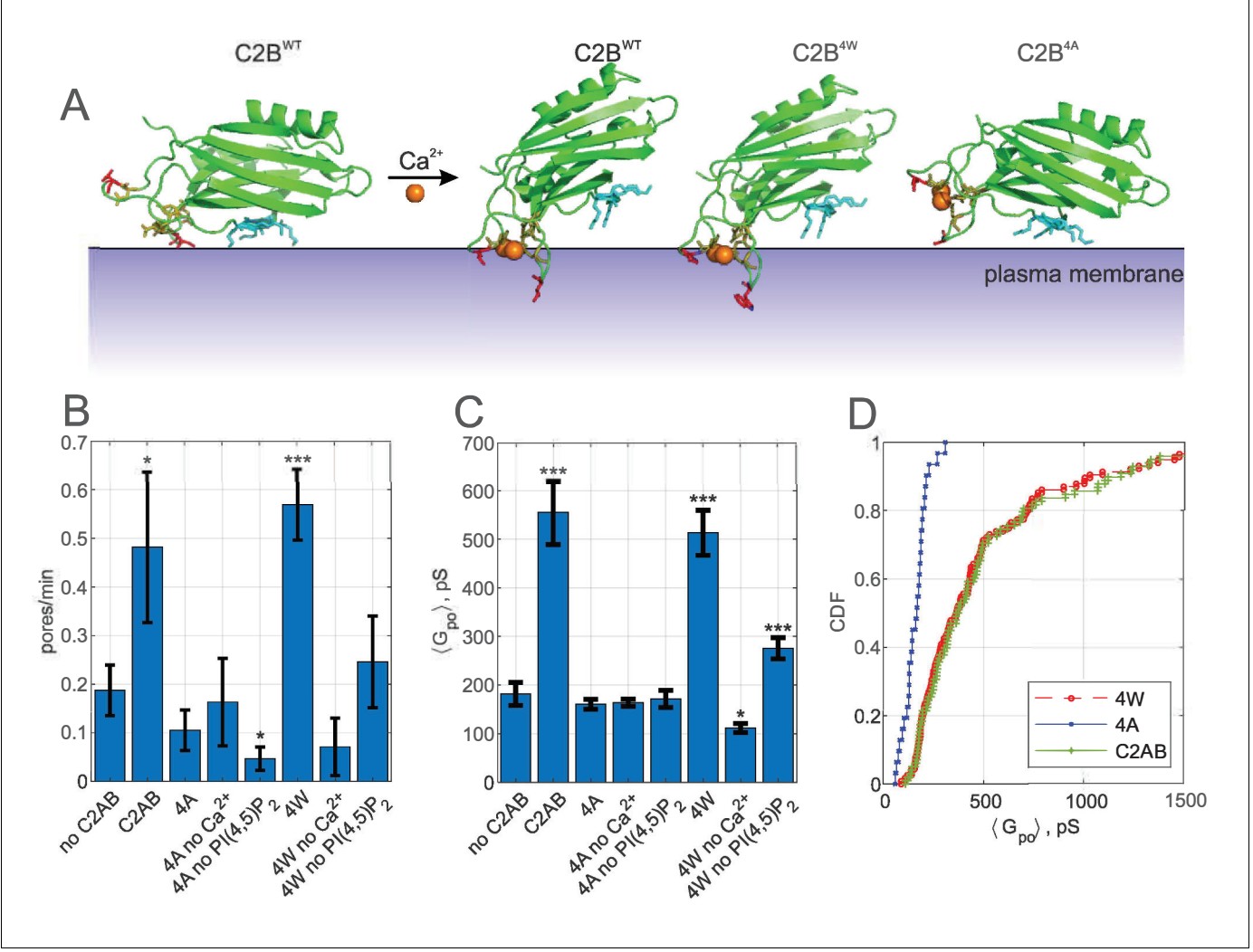

**Figure 5.** Calcium-induced membrane insertion of Syt1 C2AB hydrophobic loops are critical for both pore nucleation and expansion. (A) Schematic depiction of Syt1 C2B domain's calcium-dependent interactions with membranes. Calcium-free C2B interacts with acidic lipids through its poly-lysine motif (highlighted in cyan as in *Figure 1*). Upon binding to calcium, hydrophobic residues (V304 and I367 on C2B) insert into the membrane, causing C2B to reorient (*Chapman, 2008*) and inducing membrane curvature (*Martens et al., 2007*; *Hui et al., 2009*). In the presence of PI(4,5)P$_2$, the calcium-bound C2B assumes a tilted conformation with respect to the membrane (*Kuo et al., 2011*). M173 and F234 on C2A top loops similarly insert into membranes in a calcium-dependent manner, with similar effect on orientation and curvature generation (*Chapman, 2008*) (not shown). A mutant with the membrane-inserting residues replaced with tryptophans (M173W, F234W, V304W, and I367W, '4W') binds membranes more avidly, resulting in more membrane tubulation activity, whereas alanine substitution of the same residues ('4A') abolishes membrane penetration and curvature induction (*Martens et al., 2007*). (B) Pore nucleation rate (mean ± S.E.M) in the presence of wildtype, 4W and 4A mutants. Student's t-test was used to assess significant differences between the 'no C2AB' group and the rest. (C) Mean open-pore conductance (± S.E.M) for the conditions indicated. Two-sample Kolmogorov-Smirnov test was used to assess significant differences between the 'no C2AB' group and the rest. (D) Cumulative density functions for mean open-pore conductances for wild-type Syt1 C2AB, 4W and 4A mutants. In A, calcium-free C2B was rendered from PDB 5w5d (*Zhou et al., 2017*) and calcium-bound C2B was rendered from 5kj7 (*Lyubimov et al., 2016*). *, **, *** indicate p<0.05, 0.01, and 0.001, respectively.

manifested as a small dip in the energy profiles (*Figure 2E*). We do not know the underlying mechanisms, as we have not identified a clear correlation between the peak's amplitude or location and the parameters we varied, such as calcium concentration.

## Pore dilation by Synaptotagmin-1 C2AB requires binding to calcium, PI(4,5)P$_2$, and likely SNAREs

We further tested the requirement for Syt1 C2AB binding to calcium, PI(4,5)P$_2$, and SNAREs for promoting pore dilation, using mutagenesis (*Figure 3*). Binding of calcium to the second C2 domain of

Syt1 is known to be essential for evoked release (*Chapman, 2008*; *Mackler et al., 2002*; *Shin et al., 2009*). When calcium binding to the C2B domain was impaired by mutating a highly conserved aspartate to asparagine (Syt1 C2AB D309N *Nishiki and Augustine, 2004*), mean single pore conductance returned to the value obtained in the presence of SNAREs alone (*Figure 3C*). The rate at which current bursts appeared also returned to the SNARE-alone level (*Figure 3B*). Other pore properties were also indistinguishable from the SNARE-alone case (see *Appendix 1—figure 5*). We conclude that calcium binding to Syt1 C2B is essential for fusion pore dilation, in addition to its well-known role for triggering the opening of a fusion pore (*Wang et al., 2006*).

The C2B domain of Syt1 possesses a polybasic patch (K324-327) that interacts with acidic phospholipids (*Figure 3A*) and is important for synchronous evoked release (*Chang et al., 2018*). Although this interaction occurs in the absence of calcium (*Chapman, 2008*), it contributes to the membrane binding energy of C2AB in the presence of calcium (*Ma et al., 2017*), presumably because multivalent interactions increase the bound lifetime of C2AB. Partially neutralizing the polybasic patch in C2B (K326A, K327A) reduced the fusion rate, and resulted in single pore conductances that were indistinguishable from those for SNARE-alone pores (*Figure 3*). Similarly, the burst lifetime and the flicker rate were comparable to the SNARE-alone level, but conductance fluctuations were reduced, while there was an increase in the pore open probability during a burst, $P_o$ (see *Appendix 1—figure 5*), as would be expected for pores that fluctuate less. Thus, in addition to its established role in evoked release (*Chang et al., 2018*; *Borden et al., 2005*), the polybasic patch in Syt1 C2B is also required for fusion pore dilation.

Two recent crystal structures identified a 'primary' interaction interface between Syt1 C2B and the four-helical SNARE complex (*Zhou et al., 2015*; *Zhou et al., 2017*; *Figure 3A*). Specifically, two arginines (R398 and R399) form salt bridges with glutamates and aspartates in a groove between SNAP25 and Syntaxin-1 (*Zhou et al., 2015*). Mutation of these arginines to glutamines (R398Q, R399Q) was shown to largely abolish evoked release from hippocampal neurons (*Chang et al., 2018*; *Zhou et al., 2015*; *Xue et al., 2008*), possibly by disrupting the interaction of Syt1 C2B with SNAREs (*Chang et al., 2018*; *Zhou et al., 2015*). When we used purified C2AB bearing the same mutations (C2AB$^{R398Q, R399Q}$) both the fusion rate and the mean pore conductance decreased significantly, close to SNARE-alone levels (*Figure 3B,C*). Burst lifetimes, conductance fluctuations, and the pore open probability were not significantly different than for pores induced by SNAREs alone, but the flicker rate was lower (see *Appendix 1—figure 5*).

Together, these results indicate that binding of Syt1 to calcium, PI(4,5)P$_2$, and likely SNAREs, which are all crucial for Syt1's role in evoked neurotransmitter release (*Brunger et al., 2018a*; *Chapman, 2008*), are also essential for its function in dilating SNARE-induced fusion pores.

## Calcium-dependence of pore dilation by Syt1 C2AB

To determine whether pore properties are altered by calcium, we varied the free calcium concentration in the pipette solution and repeated the fusion experiments. Mean open-pore conductance $\langle G_{po} \rangle$ increased with increasing calcium (*Figure 4A*), consistent with a mathematical model (see below). Conductance fluctuations and burst lifetimes also increased, while the flicker rate decreased slightly and the pore open probability during a burst did not change significantly as [Ca$^{2+}$] was increased (see *Appendix 1—figure 6*). That is, pores tended to last longer with higher calcium, and the open state conductance increased. The rate at which pore currents appeared also increased with calcium (*Appendix 1—figure 6F*).

The conductances in the open-state and the corresponding pore radii ($r_{po}$) were broadly distributed at all calcium concentrations tested, but the distributions did not shift uniformly as calcium increased (see *Appendix 1—figure 6*). The apparent free energy profiles, estimated from the pore size distributions, are plotted in *Figure 4B*. With increasing calcium, the well around the most likely radius (~0.5-0.7 nm) became wider, and the slopes of the energy profiles for radii above the well's upper boundary, reflecting the force needed to dilate the pore, decreased as calcium increased. The calcium concentration at which this transition occurs (~20 μM) is consistent with the known calcium binding affinity of Syt1 (*Ma et al., 2017*; *Pérez-Lara et al., 2016*; *Bai et al., 2004b*; *Radhakrishnan et al., 2009*; *Davis et al., 1999*).

We also examined the kinetics of pore dilation as a function of calcium (*Figure 4C,D*). To this end, we averaged pore conductances after aligning them to the initial pore opening, in the presence

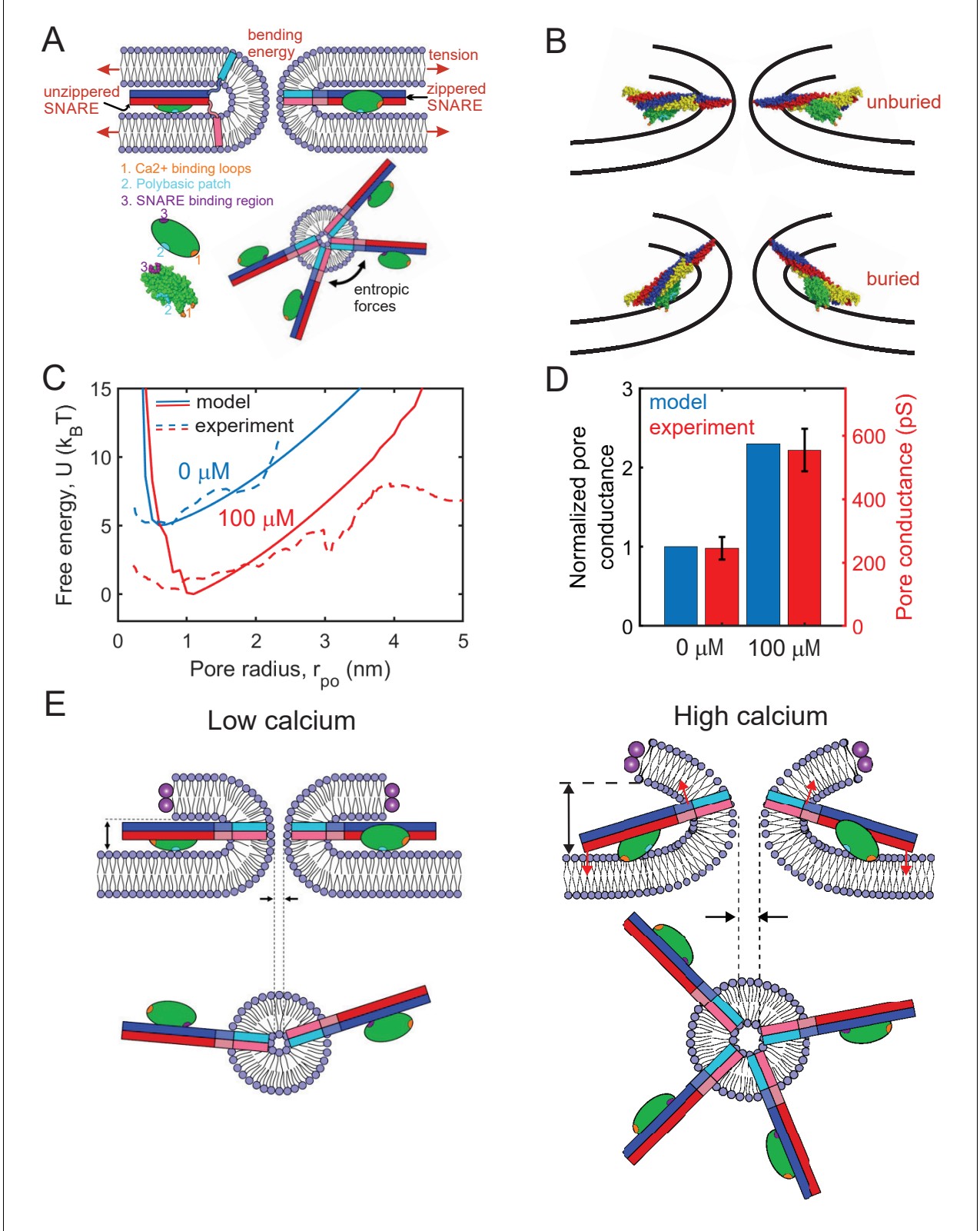

**Figure 6.** Mathematical model of the fusion pore in the presence of Syt1 and SNAREs suggests a mechanical calcium-triggered pore dilation mechanism. (**A**) Schematic of model. The membrane free energy has contributions from membrane tension and bending energy. SNARE complexes may be unzipped and free to roam laterally, or zippered and confined to the pore waist. Crowding among zippered SNARE complexes generates entropic forces that tend to enlarge the pore (top view, shown lower right). The Syt1 C2B domain (green ellipsoid) has a SNARE-binding region, a

*Figure 6 continued on next page*

*Figure 6 continued*

polybasic patch and Ca$^{2+}$-binding loops. (B) Free energy-minimizing fusion pore shapes determined by solving the membrane shape equation in the presence and absence of constraints applied by the SNARE-C2B complex (see Appendix 1). The C2B calcium-binding loops may either be unburied (top panel) or buried (lower panel) in the membrane. In the buried state the SNARE complex tilts upwards, expanding the fusion pore. The membrane shape constraint is evaluated using the SNARE-C2B complex crystal structure in a space filling representation. Both upper and lower panels depict situations in the presence of Ca$^{2+}$. The model predicts the tilted configuration is strongly favored at high [Ca$^{2+}$] following equilibration, while the untilted configuration is relevant to the kinetics that establish this equilibrium, and to experiments using low [Ca$^{2+}$]. VAMP2, syntaxin, SNAP25 and the C2B domain are shown blue, red, yellow, and green, respectively. The C2B hydrophobic membrane-inserting residues (V304, I367), polybasic patch (K326, K327) and SNARE-binding region (R398, R399) are shown orange, cyan, and purple, respectively. The protein structure was generated with PyMOL (*Schrodinger, LLC, 2015*) using the SNARE-C2B crystal structure (PDB ID 5ccg) (*Zhou et al., 2015*). The TMD of the SNARE complex (PDB ID 3hd7) (*Stein et al., 2009*) was incorporated using UCSF chimera software (*Pettersen et al., 2004*). (C) Model-predicted free energy and experimental apparent free energy versus pore radius without calcium and in the presence of excess calcium. (D) Model-predicted normalized conductances shown with experimentally measured values for comparison. Experimental data taken from *Figure 2B* experiments including Ca$^{2+}$ and PI(4,5)P$_2$. (E) Pore dilation mechanism emerging from the model. Under conditions of low calcium concentration, the C2B domain is unburied, the SNARE complex lies parallel to the membrane and the membrane separation is set by the maximum thickness of the SNARE-C2B complex. At high calcium concentrations, the calcium binding loops penetrate the plasma membrane, rotating the C2B domain and the entire SNARE-C2B complex which exerts force (red arrows) on the upper and lower membranes of the fusion pore in a lever-like action. These forces increase the fusion pore height, which is coupled by membrane energetics to fusion pore dilation.

of Syt1 C2AB at different Ca$^{2+}$ levels. The average conductance rapidly increased after initial pore opening for all traces, but reached larger values for larger calcium concentrations (*Figure 4C*). We estimated the pore expansion rate as the 10–90% rise time from the baseline to the level of conductance reached within the first 100 ms after pore opening, divided by the time it took for this rise (*Figure 4D*). With low amounts of calcium (0–30 μM), the expansion rate is ~3–12 nS/s, which increases rapidly to 20–25 nS/s for 40–100 μM Ca$^{2+}$.

Both the increase in mean open-pore conductance (*Figure 4A*) and the pore expansion rate (*Figure 4D*) with increasing free calcium were fit to a Hill equation, using parameters describing cooperative binding and loop-insertion of Syt1 C2AB to lipid bilayers containing PI(4,5)P$_2$ (*Bai et al., 2004b*).

## Calcium-dependent membrane-insertion of Syt1 C2AB is necessary for pore dilation

Calcium binds simultaneously to acidic phospholipids and highly conserved aspartate residues in a pocket formed by loops at the top of the beta-sandwich structure of the Syt1 C2 domains (*Chapman, 2008*; *Shin et al., 2009*; *Martens and McMahon, 2008*). Hydrophobic residues at the tips of the loops flanking the calcium-binding residues in Syt1 C2A (M173 and F234) and C2B (V304 and I367) insert into the membrane as a result of these interactions, strengthening membrane binding of C2 domains (*Chapman, 2008*; *Ma et al., 2017*; *Chapman and Davis, 1998*) while causing a reorientation of the C2 domains (*Kuo et al., 2011*; *Herrick et al., 2006*; *Figure 5A*). The membrane insertion of these hydrophobic residues contributes to the triggering of release (*Lynch et al., 2008*; *Martens et al., 2007*; *Hui et al., 2009*). We wondered whether membrane-insertion of hydrophobic loops also played any role in pore dilation. To test this, we introduced mutations that made the loops insertion-deficient (M173A, F234A, V304A, and I367A, the '4A' mutant [*Lynch et al., 2008*; *Martens et al., 2007*]) or that increased membrane affinity (M173W, F234W, V304W and I367W, the '4W' mutant [*Lynch et al., 2008*; *Martens et al., 2007*]).

In the nanodisc-cell fusion assay, the membrane penetration deficient 4A mutant was non-functional, having no discernible effect on pore dilation or fusion rate when compared to the assay without Syt1, other than a slight reduction in the fusion rate in the absence of PI(4,5)P$_2$ (*Figure 5B–D*). By contrast, the 4W mutant which binds the membrane more avidly essentially behaved like the wild-type C2AB, with the exception that the pore dilation ability of the 4W mutant was less dependent on the presence of PI(4,5)P$_2$ (*Figure 5C* and see *Appendix 1—figure 7*). Thus, calcium-induced membrane penetration of Syt1 C2 domains is required for pore expansion by Syt1.

# Mathematical modeling suggests that Syt1 and SNARE proteins cooperatively dilate fusion pores in a mechanical lever action

How do Syt1 and SNAREs cooperate to expand the pore in the presence of calcium? To help elucidate the mechanism, we developed a detailed mathematical model of the membrane fusion pore and the ApoE scaffold of the NLP in the presence of SNARE proteins and the C2AB domain of Syt1 (see Appendix 1 for model details and parameters). The energetics of the fusion pore membrane are described in the classic Helfrich framework, with contributions from bending energy and membrane tension (*Helfrich, 1973*), while the ApoE scaffold is modelled by adapting the theory of elasticity (*Landau and Lifshitz, 1986a*) (see *Appendix 1—figure 8A*). We obtained the minimum energy shape of the fusion pore with a given height and radius by solving the membrane shape equation (*Zhong-can and Helfrich, 1989*), assuming that the membrane has zero slope where it joins the tCell, taken as a remote location (see Appendix 1). We found there was very little change in the shape of the fusion pore when either this location was changed or freely hinged boundary conditions were used instead at this location, demonstrating that the model is insensitive to these assumptions. To compare directly with the present experiments, we incorporate four SNARE complexes, each of which can either be in the *trans* configuration and free to roam the fusion pore, or else fully zippered in the *cis* configuration near the waist, *Figure 6A* (*Wu et al., 2017b*). The model accounts for the SNARE zippering energy which favors full zippering (*Gao et al., 2012*; *Ma et al., 2015*), and for crowding interactions among zippered SNAREs which favor partial unzipping into the *trans* state, an entropic effect (*Wu et al., 2017b*; *Mostafavi et al., 2017*).

Syt1 C2B domains are assumed bound to each SNARE complex at the so-called primary interface identified in recent crystal structures (*Zhou et al., 2015*; *Zhou et al., 2017*; *Wang et al., 2016*; *Figure 3A*). For simplicity, we first consider only the C2B domain in our model. When Ca$^{2+}$ is bound to the C2B domain loops, the loops may be buried or unburied in the membrane with a relative probability that depends on the calcium concentration according to the Hill equation (*Bai et al., 2004a*; *Radhakrishnan et al., 2009*). We use a Hill coefficient of 2.3, and the measured affinity of calcium for Syt1 in the presence of PI(4, 5)P$_2$-containing membranes (*Bai et al., 2004b*). Without calcium, the loops are assumed unburied.

Thus, in the presence of calcium, the model permits two configurations of the SNARE-C2B complex, implemented according to the crystal structure (PDB ID 5ccg [*Zhou et al., 2015*]), *Figure 6B*. (1) With bound Ca$^{2+}$, the C2B complex can be in the buried state, in which the C2B polybasic patch lies ~ 0.5 nm from the membrane (*Kuo et al., 2011*) and the C2B domain is anchored to the membrane by its calcium-binding loops, reported to penetrate ~ 1 nm deep (*Herrick et al., 2006*). With these constraints, the SNAREpin is forced to tilt its C-terminus 'upwards', see *Figure 6B*; precise implementation of the constraints shows that the C2B anchoring tilts the SNAREpin upwards at ~15° to the plasma membrane, imposing a significant constraint on the shape of the fusion pore. We determined whether a given fusion pore geometry satisfied these constraints by directly comparing the structure of the SNARE-C2B complex with the shape of the fusion pore (see Appendix 1). Only fusion pores satisfying the shape constraints were accepted as possible pores. (2) With no bound calcium, the C2B polybasic patch (*Kuo et al., 2009*) and the SNAREpins orient parallel to the plasma membrane. In this configuration, the SNARE-C2B complex imposes no constraints on the shape of the fusion pore. This unanchored state is also accessible when calcium is bound, with a probability that decreases with increasing calcium concentration.

Given the microscopically long pore lifetimes of seconds, we assumed the fusion pore-SNARE-Syt1 system has sufficient time to equilibrate. For a given pore radius, $r_{\mathrm{po}}$, we calculated the free energy by summing over all allowed SNARE-C2B configurations and all possible numbers of zippered SNAREs. Each state in this sum is weighted by its Boltzmann factor, yielding the free energy $U(r_{\mathrm{po}})$ and pore size distribution $\sim \exp[-U(r_{\mathrm{po}})/k_{\mathrm{B}}T]$. We assumed that the pore height is equal to the value that minimizes the free energy at a given radius $r_{\mathrm{po}}$, since other heights have small probability as the free energy increases rapidly as a function of pore height. The predicted free energy profiles with and without calcium are close to the experimental profiles, as shown in *Figure 6C*. We compared model and experimental free energies up to a maximum pore size of 4 nm, since sampling for larger pores was limited in the experiments. In agreement with experiment, introduction of calcium is predicted to increase the pore size fluctuations, as reflected by the broader distribution. From these pore size statistics, we calculated mean pore sizes and conductances. In the absence of

calcium, the model predicts a mean fusion pore radius ~0.9 nm and a mean height ~9.0 nm, due to entropic crowding effects among *cis* SNARE complexes (*Wu et al., 2017b*), *Appendix 1—figure 8*. These crowding effects expand the pore relative to the SNARE-free case, since a bigger pore increases the entropy of *cis*-SNAREs at the waist by providing more space.

When $Ca^{2+}$ is introduced at high saturating concentrations, the model predicts a ~1.4-fold increase of pore radius to ~1.3 nm, or a ~2.3-fold increase in conductance, close to the experimentally measured ~2.2-fold increase (*Figure 6D*). The pore expansion mechanism is the constraint on the pore shape imposed by the SNARE-C2B complex. At low pore radii, the SNARE-C2B complex acts as a membrane inclusion that increases the height of the fusion pore, forcing the pore to adopt energetically unfavorable shapes, biasing the system toward large pore radii (*Figure 6B,E*, Figure S8C). Due to membrane bending and tension, the fusion pore resists the lever action tending to increase its height and enlarge the pore. However, these resistance forces are insufficient to rotate the SNARE-C2B lever complex and undo its pore-enlarging action, since this would require unanchoring of the Ca-binding loops from the membrane or dissociation of the SNARE-C2B domain binding interface. Both of these are sufficiently energetically unfavorable (*Ma et al., 2017*; *Zhou et al., 2017*) to overcome the fusion pore resistance forces (see Appendix 1 for a detailed discussion). *Figure 6D* shows the predicted increase of normalized pore conductance in elevated $Ca^{2+}$ concentrations, compared with the experimental values. In summary, our model suggests a mechanism in which the SNARE-C2B complex is a calcium-triggered mechanical lever that enlarges the fusion pore in cooperation with entropic forces generated by SNARE complexes (*Figure 6E*). On addition of $Ca^{2+}$, the C2B domain rotates and inserts its calcium binding loops into the membrane, tilting the SNARE complex so that it pushes the membrane surfaces further apart in a lever action. Since this increase in pore height would otherwise increase the net membrane bending energy, the pore diameter increases to offset this penalty (see Appendix 1).

## Discussion

Membrane fusion occurs in stages. First, membranes are brought into close apposition to overcome repulsive hydration forces. Second, a small, nascent fusion pore forms, connecting the fusing membranes. Third, the initial small pore expands to allow passage of cargo molecules (*Karatekin, 2018*; *Sharma and Lindau, 2018*; *Chang et al., 2017*). Among different stages of membrane fusion, pore expansion can be energetically one of the costliest (*Jackson, 2009*; *Cohen and Melikyan, 2004*; *Chizmadzhev et al., 1995*; *Ryham et al., 2013*; *Nanavati et al., 1992*). Consistent with this notion, fusion pores connecting protein-free lipid bilayers fluctuate, flicker open-closed, and eventually reseal unless external energy is supplied in the form of membrane tension (*Chanturiya et al., 1997*), while the initial fusion pore during biological membrane fusion is a metastable structure whose dynamics are regulated by cellular processes (*Sharma and Lindau, 2018*; *Chang et al., 2017*; *Alabi and Tsien, 2013*; *Collins et al., 2016*; *Fulop et al., 2005*; *Staal et al., 2004*; *D'Agostino et al., 2018*; *Doreian et al., 2009*; *Barg et al., 2002*; *Hanna et al., 2009*; *MacDonald et al., 2006*).

Syt1 is involved in both the pore opening and pore expansion stages during calcium-triggered exocytosis. *Before* membrane fusion, Syt1 was proposed to regulate membrane apposition (*Rothman et al., 2017*; *Chang et al., 2018*; *van den Bogaart et al., 2011*; *Seven et al., 2013*; *Lin et al., 2014*), preventing fusion pore opening at low calcium by maintaining the membranes >5–8 nm apart, halting complete SNARE zippering. Upon calcium binding to Syt1, this distance is reduced to <5 nm (*Chang et al., 2018*), sufficient for SNAREs to complete their zippering and initiate fusion. Other mechanisms, such as calcium-dependent release of an inhibition of complete SNARE assembly by Syt1 (*Brunger et al., 2018b*), or concerted action of an oligomeric complex containing Syt1, SNAREs, and additional proteins (*Bello et al., 2018*; *Tagliatti et al., 2020*), have also been proposed for the pore opening stage. It has also been proposed that during this stage, curvature generation by insertion of Syt1's hydrophobic loops into the membranes may contribute to pore opening (*Lynch et al., 2008*; *Martens et al., 2007*; *Hui et al., 2009*).

*After* fusion pore opening, Syt1 contributes to the dilation of the nascent fusion pore (*Wang et al., 2006*; *Lynch et al., 2008*), but the mechanisms for this regulation have remained even less clear. Several Syt1-independent mechanisms regulating fusion pore dynamics have recently emerged. First, membrane tension promotes fusion pore dilation during exocytosis, often through

cytoskeleton-plasma membrane interactions (*Bretou et al., 2014*; *Kozlov and Chernomordik, 2015*; *Wen et al., 2016*). Second, neuronal/exocytic SNARE proteins promote fusion pore dilation by providing entropic forces due to molecular crowding at the pore's waist (*Wu et al., 2017b*), consistent with the observation that increased SNARE availability results in larger, or faster expanding pores (*Bao et al., 2018*; *Wu et al., 2017b*; *Acuna et al., 2014*; *Guček et al., 2016*; *Zhao et al., 2013*). Third, during yeast vacuole-vacuole fusion, increased fusogen volume has been suggested as a mechanism that stabilizes fusion pores (*D'Agostino et al., 2018*; *D'Agostino et al., 2017*). However, these mechanisms cannot explain fusion pore dilation during exocytosis, because none are calcium-dependent, in contrast to exocytic fusion pore expansion (*Wang et al., 2006*; *Hartmann and Lindau, 1995*; *Fernández-Chacón and Alvarez de Toledo, 1995*; *Scepek, 1998*). Previous reconstituted single-pore measurements by *Lai et al., 2013* and *Das et al., 2020* found Syt1 and calcium promoted expansion of SNARE-mediated fusion pores. In the former study, pores were detected indirectly through passage of large probe molecules (*Lai et al., 2013*), while the latter study reported that the larger, stable pores formed in the presence of Syt1, calcium and PI(4,5)P$_2$ could be closed by dissociation of the SNARE complexes by the ATPase NSF, but not by a soluble cytoplasmic fragment of the v-SNARE VAMP2 (*Das et al., 2020*). However, the mechanism of fusion pore dilation remained unclear.

Here, we found that Syt1 has roles in both fusion pore formation and dilation, consistent with studies in secretory cells (*Wang et al., 2006*; *Lynch et al., 2008*) and in previous reconstitutions (*Lai et al., 2013*; *Das et al., 2020*), and we focused on pore dilation mechanisms. Syt1 promotes expansion of SNARE-induced fusion pores in a calcium- and acidic lipid-dependent manner. When PI(4,5)P$_2$ is present, increasing free Ca$^{2+}$ leads to pores with larger mean open-pore conductance. Fusion pore expansion by Syt1 also likely relies on Syt1's interactions with the neuronal SNARE complex, because when we used C2AB domains with mutations (R398Q,R399Q) designed to disrupt the 'primary' interaction interface with the SNARE complex (*Zhou et al., 2015*; *Zhou et al., 2017*), the pore dilation function of Syt1 C2AB was largely reduced (*Figure 3*). The same mutations were previously shown to greatly reduce evoked release from hippocampal neurons (*Chang et al., 2018*; *Zhou et al., 2015*; *Xue et al., 2008*), possibly by disrupting the interaction of Syt1 C2B with SNAREs (*Chang et al., 2018*; *Zhou et al., 2015*). The most relevant interactions in which these residues engage is however not completely resolved, so results of mutagenesis of these residues must be interpreted with caution. For example, this mutation did not have a significant effect in the co-IP experiments of Syt1 with SNAREs (*Zhou et al., 2015*), but it did have substantial effects on the ability of Syt1 C2B to bridge two membranes (*Xue et al., 2008*). In addition, in the presence of polyvalent ions such as Mg$^{2+}$ and ATP, Syt1 was found not to bind to SNAREs (*Park et al., 2015*), but ATP did not have any effect in a tethered-liposome fusion assay (89). Later work by *Wang et al., 2016* examined these interactions in the presence of membranes and SNARE complexes, and suggested that the C2B (R398 R399)–SNARE complex interaction is Ca$^{2+}$ independent ($K_d$<1 μM in the presence of PI(4,5)P$_2$ in the membranes), stronger than the C2B (R398 R399)–acidic lipid interactions, persists during insertion of the Ca$^{2+}$-binding loops into the membrane, and occurs simultaneously with the calcium-independent interactions of the C2B polybasic patch with PI(4,5)P$_2$ containing membranes. Wang et al. showed ATP/Mg$^{2+}$ does not disrupt Syt1-SNARE complex interactions in the absence of Ca$^{2+}$, but the effect was not tested in the presence of Ca$^{2+}$ (*Wang et al., 2016*). Thus, although the most likely interpretation is that mutation of R398,R399 disrupts Syt1 C2B-SNARE complex binding through the primary interface, other possibilities cannot be excluded.

A mathematical model suggests the major contribution of Syt1 to pore dilation is through its mechanical modulation of the fusion pore shape. Syt-SNARE complexes introduce non-local constraints on the fusion pore shape, making larger pores more energetically favorable. How does the non-local constraint come about? Previous work showed calcium binding to isolated Syt1 C2 domains leads to insertion of the hydrophobic residues at the tips of both of the the calcium-binding loops into the membrane (*Chapman, 2008*; *Herrick et al., 2006*; *Kuo et al., 2009*; *Bradberry et al., 2019*) (however, see *Bykhovskaia, 2021*). In the presence of PI(4,5)P$_2$, calcium-bound C2B assumes a conformation in which its long axis is tilted with respect to the membrane normal, as it interacts with the membrane simultaneously through its calcium binding loops and the polybasic patch (K324-327) bound to PI(4,5)P$_2$ (*Kuo et al., 2011*; *Pérez-Lara et al., 2016*). When present, C2B also binds the t-SNAREs Stx1 and SNAP25, with its long axis parallel to the SNARE bundle, in a calcium-independent manner (*Zhou et al., 2015*; *Wang et al., 2016*). In this orientation,

the polybasic patch on C2B (K324-327) is free to interact with acidic lipids on the target membrane (*Zhou et al., 2015*). At low, resting amounts of calcium, the calcium-free SNARE-C2B complex is therefore expected to lie parallel to the membrane, with the C2B domain simultaneously interacting with target membrane acidic lipids and the SNARE complex (*Zhou et al., 2015*; *Figure 6*). By contrast, in the presence of high calcium, the calcium-bound C2B domain will tend to reorient such that its hydrophobic top loops insert into the target membrane, resulting in a tilting of the SNARE complex of ~15 degrees, which alters the pore shape (*Figure 6*). The resultant pore size increase quantitatively accounts for the conductance increase in the presence of Syt1, and its requirements for intact calcium- and SNARE-binding regions on C2B. At intermediate calcium levels, the mean pore radius is expected to have an intermediate value, as the Syt1 molecules would be activated by calcium for a fraction of the time only. Thus, our results may explain why initial fusion pore size and its expansion rate increase as intracellular calcium increases (*Wang et al., 2006*; *Lynch et al., 2008*; *Hartmann and Lindau, 1995*; *Fernández-Chacón and Alvarez de Toledo, 1995*; *Scepek, 1998*). In addition, regulation of the fusion pore shape including interbilayer distance may be a general mechanism to stabilize fusion pores against re-closure, as a similar mechanism was observed during yeast vacuole-vacuole fusion (*D'Agostino et al., 2018*; *D'Agostino et al., 2017*).

Mutations of the hydrophobic residues at the tips of the calcium-binding loops of the C2 domains (M173, F234, V304, and I367) designed to increase or decrease the affinity of Syt1 for calcium-induced membrane binding were previously interpreted largely in terms of the ability of these mutants to generate membrane curvature. Indeed, the rates of fusion between liposomes (*Martens et al., 2007*; *Hui et al., 2009*) and exocytosis (*Lynch et al., 2008*; *Rhee et al., 2005*) correlate well with the curvature-generation ability of the Syt1 mutants. By contrast, here the correlation between the curvature-generation ability of the mutants and pore expansion was not strong, with the 4W mutant with enhanced membrane tubulation activity (*Lynch et al., 2008*; *Martens et al., 2007*) having a similar effect as wild-type C2AB. Modeling supported the idea that curvature-generation by Syt1 membrane penetration is not needed to explain how Syt1 promotes pore expansion.

We also explored how Syt1 affects pore dilation kinetics as a function of calcium. We found pore expansion rate increases with increasing $[Ca^{2+}]_{free}$, with a similar dependence on calcium as the mean open-pore conductance (*Figure 4A,D*), from ~3–12 nS/s at low calcium (0–30 μM), to 20–25 nS/s at high calcium (40–100 μM). Modeling suggests the C2A domain of Syt1 is critical for rapid expansion of the fusion pore, by contributing to the total binding energy of Syt1 C2 domains to acidic membranes. By comparison, in secretory cells the pore opens suddenly (*Breckenridge and Almers, 1987*) before continuing to expand at a slower rate. In horse eosinophils stimulated by intracellular application of GTP-γ-S, pores were found to expand, on average, at 19 nS/s, 40 nS/s, and 89 nS/s at low (<10 nM), 1.5 μM, and 10 μM $Ca^{2+}$, respectively (*Hartmann and Lindau, 1995*), consistent with a later study (*Scepek, 1998*). Pore expansion rates were 5–10 nS/s for rat mast cells, with higher rates at high calcium (*Fernández-Chacón and Alvarez de Toledo, 1995*), and varied from 15 to 50 nS/s for bovine chromaffin cells (*Fang et al., 2008*; *Berberian et al., 2009*; *Dernick et al., 2005*). Lower rates (~7 nS/s) were observed in excised patch recordings (*Dernick et al., 2005*). A rate of ~98 nS/s was reported for rat chromaffin cells overexpressing myosinII (*Neco et al., 2008*). These pore expansion rates, and the increasing rates with increasing calcium, are remarkably consistent with our findings.

Our findings also recapitulate the observation that during exocytosis, fusion pore fluctuations increase with intracellular calcium (*Zhou et al., 1996*). A mathematical model suggests that this originates in the cooperative mechanical effects of Syt1 and SNAREs which exert outward expansive forces on the fusion pore. These forces oppose the inward force that results from the intrinsic tendency of the protein-free fusion pore to close down due to membrane bending and tension effects (*Wu et al., 2017b*). The net inward force is thus lowered, leading to a broader distribution of pore sizes and bigger fluctuations.

In several neuronal preparations, the maximal rate of secretion scales as $[Ca^{2+}]_i^n$ with $n \approx 4$ (*Schneggenburger and Neher, 2000*; *Schneggenburger and Neher, 2005*; *Dodge and Rahamimoff, 1967*; *Sun et al., 2007*; *Kochubey et al., 2011*; *Heidelberger et al., 1994*), while in our system the mean open pore conductance or the rate of fusion pore expansion (*Figure 4A,D*) are consistent with a Hill relationship with cooperativity ~2 and calcium affinity ~20 μM, taken from studies of purified recombinant Syt1 C2AB binding to lipid bilayers (*Bai et al., 2004b*). There are several reasons for these differences. Most importantly, the maximal rates of secretion measured in neurons or

neuroendocrine chromaffin cells is due to the rapid fusion of a pool of docked and primed vesicles called the readily releasable pool (RRP) (*Kaeser and Regehr, 2017*; *Sørensen, 2004*; *Rizzoli and Betz, 2005*). Vesicles acquire fusion-competence at low, resting calcium levels ($\leq$0.1 µM). When the calcium concentration near release sites increases rapidly in response to stimulation, fusion from the RRP ensues within milliseconds. Docking (~30 s) and priming (~10 s) are much slower events (*Kaeser and Regehr, 2017*; *Sørensen, 2004*) and require tethering and priming factors such as Munc13 and Munc18 (*Brunger et al., 2018a*; *Rizo, 2018*). There is no RRP or its equivalent in our assay: nanodiscs dock and fuse with the target cell membrane under a constant calcium level throughout the measurement and key components of the docking and priming machinery such as Munc13 and Munc18 are absent in our minimalistic reconstitution. Thus, the steep calcium-dependence of the maximal rate of RRP secretion observed in neurons is not directly comparable to the fusion pore opening or expansion kinetics in our assay in which discs fuse with the target membrane under conditions of constant calcium levels, very low fusion rates, and absence of docking and priming factors.

The nanodisc-cell fusion assay is tuned for sensitivity to post-fusion stages. Unfortunately, like other electrical or electrochemical methods that generate a signal only after fusion pore opening, our assay cannot directly detect pre-fusion stages. In particular, the delay between docking and fusion of nanodiscs, and the molecular configurations leading to the opening of the initial fusion pore are currently not known. Until a better understanding of such pre-fusion stages is achieved, our post-fusion studies should be interpreted with care. Another, possibly related, limitation is that due to the small numbers of proteins that can be incorporated into nanodiscs, large fluctuations are expected in the actual copy numbers from disc-to-disc. Such fluctuations likely contribute to the variability observed in our single-pore measurements, for example, of mean conductance values. A detailed discussion of the relevance and limitations of nanodisc-based single pore measurements in relation to exocytotic fusion pores monitored in secretory cells can be found in *Karatekin, 2018*.

In neurons and many neuroendocrine cells, fusion is triggered by a brief calcium transient. The finding that fusion pore dilation is calcium sensitive suggests that the pore size, expansion rate, and duration can be modulated by calcium dynamics. Thus, weak stimulations that result in brief calcium transients would be more likely to lead to small fusion pores and slow release, and strong stimulations would conversely result in larger and faster dilating pores. This behavior is indeed observed in neurons (*Pawlu et al., 2004*), and in neuroendocrine cells (*Fulop et al., 2005*; *Cárdenas and Marengo, 2016*). In this framework, different Syt isoforms would affect fusion pore dynamics differently, depending on their ability to reorient with respect to the membranes, their interactions with the SNAREs, and their calcium affinities.

## Materials and methods

### Recombinant protein expression and purification
Expression and purification of the constructs used are described in Appendix 1, Supplementary Materials and methods.

### Reconstitution of synaptotagmin-1 and VAMP2 into nanodiscs
Eight copies of VAMP2 (~four per face) were incorporated into nanolipoprotein particles (vNLP8) as previously described (*Wu et al., 2016*; *Wu et al., 2017b*; *Bello et al., 2016*). The protocol was modified to produce nanolipoprotein particles co-reconstituted with full-length Syt1 and VAMP2 (vsNLP), as detailed in Appendix 1, Supplementary Materials and methods.

### Stable flipped SNARE cell lines
Stable 'tCell' HeLa cell lines expressing flipped t-SNAREs (rat Syntaxin-1, residues 186–288, and rat SNAP-25, residues 2–206) and the nuclear marker CFP-nls (cyan fluorescent protein fused to nuclear localization signal) were a generous gift from the Rothman laboratory (*Giraudo et al., 2006*) and cultured as previously reported (*Wu et al., 2016*; *Wu et al., 2017b*). Details are given in Appendix 1, Supplementary Materials and methods.

## Single fusion pore conductance assay

All recordings were done as previously described (*Wu et al., 2016*; *Wu et al., 2017b*), and detailed in Appendix 1, Supplementary Materials and methods. Estimations of fusion rates and pore properties are explained in Appendix 1, Supplementary Materials and methods, along with evidence that ATP-dependent channel activity is absent and that cell membrane potential changes are negligible during recordings.

## Statistical analysis

Details are given in Appendix 1, Supplementary Materials and methods, and in figure legends.

## Acknowledgements

We thank Ekaterina Stroeva and Shyam Krishnakumar (Rothman Laboratory, Yale University) for help with reconstitution of full-length Syt1 into nanodiscs. This work was supported by National Institute of General Medical Sciences, National Institute of Neurological Disorders and Stroke, and the National Eye Institute of the National Institutes of Health under award numbers R01NS113236 and R01EY010542 (to EK) and R01GM117046 (to BOS). The content is solely the responsibility of the authors and does not necessarily represent the official views of the National Institutes of Health. We acknowledge computing resources from Columbia University's Shared Research Computing Facility project. We thank Rui Su and members of the Karatekin lab for helpful discussions.

## Additional information

### Funding

| Funder | Grant reference number | Author |
| --- | --- | --- |
| National Institute of Neurological Disorders and Stroke | R01NS113236 | Erdem Karatekin |
| National Eye Institute | R01EY010542 | Erdem Karatekin |
| National Institute of General Medical Sciences | R01GM117046 | Ben O'Shaughnessy |
| Columbia University | Shared Research Computing Facility | Ben O'Shaughnessy |

The funders had no role in study design, data collection and interpretation, or the decision to submit the work for publication.

### Author contributions

Zhenyong Wu, Conceptualization, Data curation, Formal analysis, Investigation, Writing - original draft, Writing - review and editing; Nadiv Dharan, Zachary A McDargh, Software, Formal analysis, Validation, Investigation, Visualization, Writing - review and editing; Sathish Thiyagarajan, Conceptualization, Software, Formal analysis, Validation, Investigation, Visualization, Writing - review and editing; Ben O'Shaughnessy, Conceptualization, Formal analysis, Supervision, Funding acquisition, Validation, Writing - original draft, Project administration, Writing - review and editing; Erdem Karatekin, Conceptualization, Resources, Software, Formal analysis, Supervision, Funding acquisition, Validation, Investigation, Writing - original draft, Project administration, Writing - review and editing

### Author ORCIDs

Zachary A McDargh (iD) http://orcid.org/0000-0001-9022-5593
Erdem Karatekin (iD) https://orcid.org/0000-0002-5934-8728

### Decision letter and Author response

Decision letter https://doi.org/10.7554/eLife.68215.sa1
Author response https://doi.org/10.7554/eLife.68215.sa2

## Additional files

### Supplementary files
- Transparent reporting form

### Data availability

All data associated with the plots shown in this study are included in the manuscript and supporting files. Source data files have been provided for all figures, in the form of a. zip file containing mostly matlab .fig and/or .mat files corresponding to the data presented in the manuscript and the Appendix. The raw data can be extracted for every plot from the .fig file. In a few cases, we included Excel or Igor Pro files.

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

# Appendix 1

## Recombinant protein expression and purification

All SNARE and Synaptotagmin-1 constructs used were generous gifts from James E. Rothman, unless noted otherwise. Plasmid pET32a-Trx-His6X-ApoE422K, used to express the N-terminal 22 kDa fragment of apolipoprotein E4 (residues 1–199, ApoE422K), was kindly provided by Dr. Nicholas Fischer, Lawrence Livermore National Laboratory, CA (*Morrow et al., 1999*; *Blanchette et al., 2008*). Full-length VAMP2 (residues x1-116 in plasmid pET-SUMO-VAMP2) and ApoE422K were expressed and purified as previously described (*Wu et al., 2016*; *Wu et al., 2017b*). Rat Syt1 residues 96–421 corresponding to cytoplasmic C2AB domains were expressed from a pET28a-SUMO-synaptotagmin1 vector. C2AB$^{R398,399Q}$, C2AB$^{D309N}$ and C2AB$^{K326,327A}$ were generated from the wild-type sequence using the QuickChange site-directed mutagenesis kit (Stratagene, La Jolla, CA). C2AB$^{4W}$ and C2AB$^{4A}$ were prepared using QuikChange Multi Site-Directed Mutagenesis Kit (Stratagene, La Jolla, CA). Wild-type C2AB and all mutated versions of C2AB were expressed in BL21 (DE3) and purified as previously reported (*Ma et al., 2017*). Full-length Syt1 (pET28a-SUMO-synaptotagmin 1, residues 57–421) was expressed in BL2 (DE3) at 37°C to optical density 0.8 (at 600 nm) and induced with 1 mM isopropyl β-D-thiogalactoside (IPTG) for 4 hr. Cells were then lysed by a cell disruptor (Avestin, Ottawa, CA) and lysates were clarified by centrifugation (35,000 rpm at 4°C for 30 min using a Beckman-Coulter Ti45 rotor and 70 ml polycarbonate tubes, corresponding to 142,160 × g). The supernatant was incubated with Ni-NTA agarose (Qiagen, Valencia, CA) overnight at 4°C. Protein-bound beads were washed by buffer A (25 mM HEPES, pH 7.4, 400 mM KCl, 0.5 mM tris-2-carboxyethyl phosphine [TCEP]) supplemented with 50 mM imidazole and 1% Octylglucoside (OG). The protein was first separated from beads using buffer A supplemented with 400 mM Imidazole and 4% OG. Then the His-SUMO tag was cleaved by SUMO proteinase at 4°C for 2 hr. The protein was diluted four times by dilution buffer (25 mM HEPES, 0.5 mM TCEP and 4% OG) and then immediately loaded into Mono S5/50G column (GE Healthcare Bio-Sciences, Pittsburgh, PA). The full-length Syt1 was washed out by high-salt buffer (25 mM HEPES, 1 M KCl, 0.5 mM TCEP and 1% OG). After concentration determination using the Bradford assay (Bio Rad, Hercules, CA), the samples were aliquoted, flash frozen by plunging into liquid nitrogen, and stored at −80°C for future use.

## Co-reconstitution of Synaptotagmin-1 and VAMP2 into nanolipoprotein particles (NLPs)

Eight copies each of VAMP2 and full-length Synaptotagmin-1 (Syt1) (~four per face each) were incorporated into nanolipoprotein particles (vsNLP8) following previous protocols for reconstitution of VAMP2 alone (*Wu et al., 2016*; *Wu et al., 2017b*; *Bello et al., 2016*). A mixture of palmitoyl-2-oleoylphosphatidylcholine (POPC) and 1,2-dioleoyl phosphatidylserine (DOPS) (85:15 molar ratio) dissolved in a chloroform-methanol mixture (2:1 by volume) was dried under nitrogen flow, then placed under vacuum for 2 hr. All lipids were purchased from Avanti Polar Lipids (Alabaster, AL). The lipid film was re-suspended in 25 mM HEPES, pH 7.4, 140 mM KCl, 1 mM TCEP buffer with 1% OG supplemented by the desired amount of full length syt1 and VAMP2. The mixture was vortexed for 1 hr at room temperature followed by the addition of ApoE422K and vortexed for another half hour at room temperature and then 3 hr at cold room. The ApoE422K:VAMP2: syt1: lipid ratio for vsNLPs was 1:2:2:180. Excess detergent was removed using SM-2 bio-beads (Bio-Rad) overnight at 4°C with gentle shaking. The assembled vsNLPs were purified using size-exclusion chromatography using a Superose 6, 10/300 GL column (GE Healthcare Bio-Sciences, Pittsburgh, PA). Collected samples were concentrated using Amicon Ultra (30 KDa cutoff) centrifugal filter units, and analyzed by SDS-PAGE with Coomassie staining. The size distribution of the NLPs was determined for every batch of production using transmission electron microscopy (JEM-1400, JEOL, MA, USA). This allowed estimating the average number of ApoE copies per disc as before (*Wu et al., 2017b*; *Bello et al., 2016*), using previously published information about the number of ApoE copies as a function of disc size (*Blanchette et al., 2008*). The copy numbers of Syt1 and VAMP2 per disc were then estimated from the quantification of Syt1- or VAMP2-to-ApoE ratio using densitometry (ImageJ, NIH).

## Stable flipped SNARE cell lines

Stable 'tCell' HeLa cell lines expressing flipped t-SNAREs (rat Syntaxin-1, residues 186–288, and rat SNAP-25, residues 2–206) and the nuclear marker CFP-nls (cyan fluorescent protein fused to nuclear localization signal) were a generous gift from the Rothman laboratory (*Giraudo et al., 2006*) and cultured as previously reported (*Wu et al., 2016*; *Wu et al., 2017b*). Mycoplasma contamination was reported and tested not to affect the results in *Wu et al., 2017b*. Note that the as long as the cells express flipped t-SNAREs on their surfaces (which was quantified in the references above), they fuse with membranes harboring the cognate v-SNAREs. In fact, fusion was reported with other cell lines (e.g. CHO or HEK under transient expression of flipped SNARE constructs. e.g. see *Hu, 2003*). The flipped SNARE constructs used in the generation of these lines, pBI-flipped Syntaxin-1 (186–288)-flipped SNAP-25-IRES-CFP-nls, are schematically shown in *Appendix 1—figure 1I* (*Giraudo et al., 2006*; *Giraudo et al., 2005*). The pBI expression vector is a bidirectional mammalian expression vector of the Tet-Off gene expression system that allows co-regulation of the synthesis of two gene products in stoichiometric amounts (*Baron et al., 1995*). The cells were cultured in DMEM (4500 mg/L glucose, L-glutamine, sodium pyruvate, and sodium bicarbonate) and 10% (v/v) fetal calf serum at 37°C.

## PI(4,5)P$_2$ incorporation and immunostaining

Where indicated, short-chain diC8-PI(4,5)P$_2$ (Echelon Biosciences Inc, Salt Lake City, UT) (1 mM stock solution, dissolved in water), was added to the cell culture medium to a final concentration of 20 μM and incubated 20 min at 37°C. Cells were then washed three times using extracellular buffer (ECS: 125 mM NaCl, 4 mM KCl, 2 mM CaCl$_2$, 1 mM MgCl$_2$, and 10 mM HEPES, pH adjusted to 7.2 with NaOH and 10 mM glucose added freshly).

For assessing diC8-PI(4,5)P$_2$ incorporation into the outer leaflet of the plasma membrane and lifetime, after 20 min incubation with the lipid, cells were rinsed thoroughly with phosphate buffered saline (PBS) supplemented with 10% goat serum, and kept at 37 °C with the same solution for different durations. Mouse monoclonal anti-PI(4,5)P$_2$ primary antibodies (Echelon Biosciences Inc, Utah) were added to the cells at time points of 0, 40, and 80 min and incubated 1 hr at 37°C. Then cells were fixed with 4% paraformaldehyde (Electron Microscopy Sciences, PA) for 20 min at room temperature before addition of goat anti-mouse IgM heavy chain secondary antibody conjugated with Alexa Fluor 647. Control cells that were not incubated with diC8-PI(4,5)P$_2$ were treated similarly and fixed by 4% paraformaldehyde. Some cells were then permeabilized by 0.5% saponin (Sigma, MO) to allow access of the antibody to the inner leaflet of the plasma membrane where endogenous PI(4,5)P$_2$ resides. Cells were blocked for 30 min with PBS supplemented with 10% goat serum, followed by incubation with anti-PI(4,5)P$_2$ primary antibody for 1 hr at 37°C. After three successive washes in PBS, cells were incubated with the secondary antibody as above. All groups of cells were washed three times with PBS and mounted on a glass slide with mounting medium (ProLong Gold Antifade Mountant with DAPI, Molecular Probes, OR). Fluorescence images were collected using a spinning disk confocal microscope (model TiE, Nikon, Japan, equipped with a Yokogawa CSU-W1 spinning disc head and CFI Plan Apochromat Lambda 60x/1.4 oil immersion objective). Images were analyzed using ImageJ software. We drew a region of interest (ROI) around cells using the freehand ROI tool and measured the mean pixel intensity in the ROI. We then subtracted the intensity from a nearby region not containing any cells to define the background subtracted pixel intensity to define $\Delta F$ in *Figure 1B*. For each condition, 10 regions of interest encompassing cells were analyzed from three to six independent preparations.

## Whole-cell conductance of flipped t-SNARE cells

We measured whole-cell current responses to step changes in membrane potential under voltage-clamp, from HeLa cells stably expressing flipped t-SNAREs (Figure S11A). Currents were averaged for 27 cells and plotted against voltage (Figure S11B). Pipettes were filled with intracellular solution (in mM): 134 KCl, 2 MgCl$_2$, 1 CaCl$_2$, 10 HEPES and 10 EGTA (pH is adjusted to 7.2 by KOH).

## Single fusion pore conductance assay

All recordings were done as previously described (*Wu et al., 2016*; *Wu et al., 2017b*). Briefly, a dish with cultured tCells was rinsed using ECS, then mounted on a Thermo-Plate (Tokai Hit, Shizuoka-ken, Japan) pre-set to 37°C. tCells were visualized with an inverted Olympus IX71 microscope (Olympus Corp., Waltham, MA) using a ThorLabs USB3.0 digital camera (UI-3240CP-NIR-GL-TI) controlled by ThorCam software (ThorLabs, Newton, NJ). Recording pipettes (borosilicate glass, BF 150-86-10, Sutter Instruments, Novato, CA) were pulled using a model P-1000 pipette puller (Sutter Instruments, Novato, CA) and polished using a micro-forge (MF-830, Narishige, Tokyo, Japan). The pipette solution (PipSol) contained: 125 mM NaCl, 4 mM KCl, 1 mM MgCl2, 10 mM HEPES, 26 mM TEA-Cl, 2 mM ATP (freshly added), 0.5 mM EGTA, pH adjusted to 7.2 by NaOH and the indicated free calcium (0–500 µM) was adjusted by 0.1 M Calcium Standard Solutions (Thermo Fisher Scientific, Waltham, MA). Free calcium was calculated using MaxChelator (https://somapp.ucdmc.ucdavis.edu/pharmacology/bers/maxchelator/CaMgATPEGTA-TS.htm) taking into account ATP, $Mg^{2+}$, ionic strength, temperature, and pH. The pipette was pre-filled by PipSol and then back filled with PipSol supplemented with nanodiscs with or without additional C2AB. All voltage-clamp recordings were made using a HEKA EPC10 Double USB amplifier (HEKA Elektronik Dr. Schulze GmbH, Lambrecht/Pfalz, Germany), controlled by Patchmaster software (HEKA). Current signals were digitized at 20 kHz and filtered at 3 kHz. The recording traces were exported to MatLab (MathWorks, Natick, MA) and analyzed as previously described in detail (*Wu et al., 2016*; *Wu et al., 2017b*).

## Detection of fusion pore currents

As described previously (*Wu et al., 2016*; *Wu et al., 2017b*; *Dudzinski et al., 2019*), the pipette tip was initially filled with ~1 µl of disc-free buffer and back-filled with NLPs suspended in the same buffer (final [NLP] ≈ 100 nM, 120 µM lipids). This allowed establishing a tight seal ($R_{seal}$>10 GOhm) with high success rate and recording a stable baseline before the NLPs diffused to the membrane patch and started fusing with it a few to several min later. All cell-attached recordings were performed using a holding potential of $V_p = -40$ mV relative to bath. With a cell resting membrane potential of 56±7 mV (mean ± S.D., n=36 *Wu et al., 2016*), this provided 16 mV driving force across the patch membrane. The pipette solution had resistivity 0.60 Ohm.m, measured using a conductivity cell (DuraProbe, Orion Versa Star, Thermo Scientific).

After a good seal was established on a cell, currents were recorded under voltage-clamp for 800 s, in 40 s sweeps, with a sampling rate of 20 kHz using a HEKA EPC10 Double USB amplifier (HEKA Elektronik), controlled by Patchmaster software (HEKA). The analysis pipeline started with initial offline inspection of the traces in PatchMaster. Traces with activity were exported to Matlab (Mathworks, Natick, MA) where they were analyzed in more detail using an interactive graphical user interface we developed to help identify, crop and process single fusion pore currents (*Wu et al., 2016*; *Wu et al., 2017b*; *Dudzinski et al., 2019*). Traces with excessive noise or unstable baseline were excluded from analysis. Exported traces were low-pass filtered (280 Hz cutoff) and frequencies due to line voltage were removed using notch filtering. Zero phase shift digital filtering algorithms (Matlab Signal Processing Toolbox function filtfilt) were employed to prevent signal distortion. Filtered traces were averaged in blocks of 80 points (125 Hz final bandwidth) to achieve rms baseline noise ≤ 0.2 pA. Currents $I$ for which $|I| > 2.0$ pA for at least 250 ms were accepted as fusion pore current bursts. During a burst, rapidly fluctuating currents often returned to baseline multiple times, i.e. pores flickered. To quantify pore flickering, we defined currents ≤0.2 pA and lasting ≥60 ms (15 points) as open pores and currents not meeting these criteria as closed. For a given burst, the number of open periods was equal to the number of flickers, $N_{flickers}$. The burst lifetime is defined as the time from the initial to the final point detected using the criteria above. Current bursts spaced >5 s apart by a quiet baseline were assigned to separate bursts, since the typical lifetime of well-isolated bursts is 5-10 s. An example of a current burst is shown in *Appendix 1—figure 1J* with the threshold current, detected open sub-periods, and the burst lifetime indicated. Examples of entire 800 s recordings are shown in Fig. S2. The MatLab programs used in analysis and the data are available upon request.

## Estimation of fusion rate

To estimate the fusion rate for each recording (i.e. the rate at which current bursts appeared), we counted the number of current bursts that fit the set criteria (current amplitude >2 pA for at least 250 ms) and divided this number by the duration of the recording. Examples are shown in *Appendix 1—figure 2A, B*. These per-cell rates were averaged over all cells to estimate the average rate of fusion ('pores/min') and its standard deviation for a given condition. Standard error of the mean was calculated as the standard deviation divided by the square root of the number of cells. Periods during which the baseline was not stable were excluded from this analysis. For individual cells, the number of well-isolated pores varied from 0 to 22. Many recordings ended with what seemed to be currents from overlapping fusion pores (*Appendix 1—figure 2B*). Such end-of-record currents were also excluded, since they could also be attributed to a loose seal. Thus, the fusion rates we report may underestimate the true rates, especially for conditions where fusion activity was high.

We checked that increasing or decreasing the concentration of v-SNARE NLPs in the pipette solution increased or decreased the fusion rate, respectively. Indeed, we found there is good linear correlation between the v-SNARE NLP concentration and the fusion rate, as shown in *Appendix 1—figure 2C*.

As an alternative estimate of the fusion rate, for every condition, we summed all detected pores, $N_{tot}$, and the analysis time $\tau_{tot}$ over all cells (excluding portions with noisy/unstable baseline), and calculated the total number of pores divided by the total analysis time, $F_{tot} = N_{tot}/\tau_{tot}$. The results of this estimate were close to the ones described above, as shown in *Appendix 1—figure 2D*.

## Estimation of fusion pore parameters

The number of flickers, $N_{flickers}$, and the burst lifetime, $T_o$, were defined as explained above. The flicker rate was defined as the number of flickers divided by the burst lifetime for individual pores. The pore open probability, $P_o$, is defined as the total time the pore was in the 'open' state divided by the burst lifetime, $T_o$, for individual pores. We converted current to conductance by dividing every point in a current trace by the transmembrane voltage $V_m = V_{cell} - V_p = -16$ mV, where $V_p$ is the pipette potential (-40 mV) and $V_{cell} = -56$ mV as indicated above. To calculate the open-pore conductance, $G_{po}$, and its statistics, we used pore open-state values, denoted by the subscript 'po'. Similarly, we used pore open-state values to calculate the distributions of open-pore conductance values and radii. For the distributions in *Figure 2C,D*, S4, S6, and S7, we first computed the probability density functions (PDFs) for individual pores using a fixed bin width for all, then averaged these to give equal weight to all pores. All distribution fits (e.g. Figure S3E, F) were performed using Matlab Statistics Toolbox functions fitdist or mle, using maximum likelihood estimation. Open-pore conductance values were used point-by-point to estimate the open-pore radii, by approximating the pore as a cylinder and using the expression (*Hille, 2001*) $r_{po} = \left(\rho\lambda G_{po}/\pi\right)^{1/2}$, where $\rho$ is the resistivity of the solution, $\lambda = 15$ nm is the length of the cylinder, and $G_{po}$ is the open-pore conductance.

For assessing statistical significance when comparing sample means, we used the student's t-test when the parameters were normally distributed, or the nonparametric two-sample Kolmogorov-Smirnov test otherwise (ttest2 or kstest2, Matlab Statistics Toolbox), as indicated in figure legends. We considered each single-pore measurement a biological replicate.

## Estimation of fusion pore expansion rates

For aligning and averaging conductance traces in *Figure 4C*, we shifted the time axis such that $t = 0$ corresponded to the first data point in a burst. We estimated pore expansion rate as the 10-90% rise time from the baseline to the level of conductance reached within the first 100 ms after pore opening, divided by the time it took for this rise using the Matlab function 'slewrate'.

As an alternative, we also fit a straight line to each of the aligned and averaged conductance rise, for the initial 16 ms of the rise, and used the slope of the line as an estimate of the pore expansion rate. Pore expansion rates estimated from these slopes as a function of [Ca²⁺] resulted is a plot very similar to the one in *Figure 4D* obtained using the slew rate estimate above. The differences in the slopes can be largely explained by differences in the conductance level reached within ~16 ms. After filtering and block averaging, the spacing between successive points is 4 ms in individual traces,

corresponding to a Nyquist frequency of 125 Hz. That is, we should be able to faithfully reconstruct signals varying on a time scale of 8 ms or slower. However, slopes calculated over a 16 ms span are still likely to be limited by our resolution to some degree, because we cannot detect finer kinetic details during this period. Thus, the pore expansion rates we averaged over 16 ms may be underestimates of the true rates and finer details of the kinetics cannot be resolved.

## No evidence for ATP-dependent channel activity in flipped t-SNARE cells

For cell-attached single-pore measurements, ATP was included in the pipette solutions. HeLa cells were reported to express ATP-dependent P2 receptors (*Welter-Stahl et al., 2009*). To test whether ATP-dependent channel activation is present in the flipped t-SNARE cells, we recorded currents from cell-attached, voltage-clamped patches from these cells in the absence and presence of ATP (nanodiscs were absent). Pipette solutions were the same as for single fusion pore measurements with 100 μM free calcium, except for ATP as noted. Both in the absence and presence of ATP (2 mM), we occasionally had patches that displayed channel-like activity (Figure S9). We conclude that the activity of these channels is not regulated by ATP, consistent with an earlier report (*Welter-Stahl et al., 2009*).

In addition, we note that the vast majority of channel-like currents as in Figure S9 are excluded from our analysis of fusion pore currents, because their lifetime is too short (<250 ms), their amplitude is too low (<-2 pA), or both, and therefore do not significantly affect our results.

## Cell membrane potential changes do not significantly distort cell-attached fusion pore recordings

It has been reported that cell membrane potential may change under some conditions during cell-attached recordings (e.g. see *Fenwick et al., 1982*). In such recordings, the single-channel conductance $g$ is underestimated (compared to its true value $G$), unless $G_{cell} \gg G_{patch}$, where $G_{cell}$ is the cell membrane conductance, and $G_{patch}$ is the patch conductance (*Fischmeister et al., 1986*). Given that the ratio of the cell area to patch area is typically $A_{cell}/A_{patch} > 100$ or 1000, and that membrane capacitance is proportional to membrane area, one would expect the requirement for $G_{cell} \gg G_{patch}$ is easily satisfied. However, for some small cells, sometimes it is found that ions passing through single channels can change the cell membrane potential, hence the potential across the patch (*Fenwick et al., 1982*; *Hamill, 1983*). The effect was found only occasionally for some cells from the same preparation, and for small cells. *Fenwick et al., 1982* suggested that some local damage to the membrane patch during the formation of the gigaseal may occur in some cases.

Several lines of evidence suggest cell membrane potential changes do not significantly distort our cell-attached recordings:

1. If the cell membrane potential changed due to currents passing through fusion pores, such currents would depolarize the cell membrane and reduce the transmembrane voltage across the patch ($V_m = V_{cell} - V_p$). Indeed a 15–20 mV depolarization of the cell membrane from its starting value of -56 mV would bring it close to $V_p$ and largely abolish the driving force $V_m$ for current flow across the patch. This would result in larger currents at the beginning of a pore event compared to its end, and this effect would be strongest for the condition producing the largest pores, that is, in the presence of full-length Syt1 (with calcium and PI(4,5)P$_2$). To test this idea, we aligned pore currents to the beginning or end of events, and averaged them, as shown in Fig. S10. We do not find large differences between averaged traces aligned either way.

2. The hallmark of cell membrane potential changes in single-channel recordings is 'relaxation' of single-channel currents when the channels open and close (*Fenwick et al., 1982*). In our recordings, we do not see such relaxation, even after aligning pore current to their moment of closure and averaging them as shown in Figure S10.

3. In whole-cell voltage-clamp recordings, we found $G_{cell} = 5 - 6$ nS for flipped t-SNARE cells (Fig. S11). Thus, the condition $G_{cell} \gg G_{patch}$ is satisfied in most of our recordings. Even in the presence of Syt1, $G_m$ is nearly 10 times larger than the average conductance ($G_{patch} \approx 600$ pS). The range of mean open-pore currents and transmembrane voltages comprising 95% of the

data values for C2AB in the presence of PI(4,5)P$_2$ and 100 $\mu$M calcium are indicated as a red-colored box on Figure S11B.

## Statistical analysis

For fusion rates and other parameters that were expected to follow a normal distribution, the two-sample t-test was used. For open-pore conductance, or other parameter distributions which do not follow a normal distribution, the two-sample Kolmogorov-Smirnov test was used for pair-wise comparisons. In *Figure 1B* we used one-way ANOVA, followed by a multiple comparison test (using the Tukey-Kramer criterion). For all statistical analyses, we used Matlab Statistics and Machine Learning Toolbox (MathWorks). Details are provided in figure legends.

## Mathematical model of the fusion pore with snares and synaptotagmin-1

The shape of the fusion pore between the nanolipoprotein particle (NLP) and the tCell membrane is determined by minimizing the Helfrich energy (*Helfrich, 1973*); this is achieved by numerically solving the membrane shape equation with constraints fixing the pore radius $r_{\mathrm{po}}$ and height $h$, defined to be the separation between the NLP and tCell membrane (see subsection Numerical method for solving the membrane shape equation). We assume that each side of the NLP contains $N$ v-SNAREs and that all are available to associate with the t-SNAREs in the tCell membrane and contribute to pore expansion. Out of $N$ SNAREs, $N_{\mathrm{Z}}$ denotes the number of fully zippered SNAREs. For a given set of values $(r_{\mathrm{po}}, h, N, N_{\mathrm{Z}})$ the total free energy of the fusion pore is

$$U_{\mathrm{tot}}(r_{\mathrm{po}}, h, N, N_{\mathrm{Z}}) = U_{\mathrm{mb}} + U_{\mathrm{hyd}} + U_{\mathrm{SNARE}} + U_{\mathrm{scaffold}}, \tag{1}$$

where $U_{\mathrm{mb}}$, $U_{\mathrm{hyd}}$, $U_{\mathrm{SNARE}}$, and $U_{\mathrm{scaffold}}$ stand for the membrane energy of the pore, the energy due to hydration forces between the NLP and tCell membranes, the free energy associated with the SNAREpins, and the free energy of the deformed NLP scaffold, respectively.

Each SNARE is bound to a Syt1 C2AB domain at the primary interface between the SNARE and the C2B domain (*Zhou et al., 2015*). The C2AB domain calcium binding loops can be unburied or buried in the membrane with a probability that depends on calcium concentration (see 'Calcium dependent pore conductance' below). In the unburied state, the C2B polybasic patch is facing the tCell membrane and parallel to it (*Kuo et al., 2009*). In this orientation the C2B-attached SNARE is also roughly parallel to the membrane. In the buried state, the C2B domain anchors to the membrane by insertion of its calcium binding loops $\sim 1\mathrm{nm}$ into the membrane (*Herrick et al., 2006*), and the polybasic patch is distanced $\sim 0.5\mathrm{nm}$ from the membrane (*Kuo et al., 2011*). With respect to the unburied state, this configuration has a rotated C2B domain, which is attached to the SNARE complex at the primary interface, such that the SNARE complex is somewhat raised above the membrane and is concomitantly tilted by ~15° with respect to the membrane plane (*Figure 6B*, main text). This tilt angle is measured by taking the inverse sine of the ratio between the SNARE motif length projected on the vertical axis of the pore and the length of the SNARE motif. Thus, the C2B domain acts as a fulcrum about which the SNARE lever pivots. This configuration imposes a geometric constraint on the pore, leading to increased pore radius and height.

We calculate the pore conductance in the absence of calcium where C2B domains are unburied and in saturating calcium levels when all C2B domain are buried, and use these values to predict the mean pore conductance as described in the subsubsection *Model- predicted pore conductance* below. The expressions for the different energy terms in *Equation (1)* are described below.

## Membrane free energy

The NLP and tCell membranes are modelled as a planar bilayer with diameter $D$ and an infinite planar bilayer, respectively, and both are at a constant surface tension. The membrane free energy of the fusion pore is given by *Helfrich, 1973*,

$$U_{\mathrm{mb}}(r_{\mathrm{po}}, h) = \min\left\{\int_{A_{mb}}\left[\frac{\kappa}{2}(2C)^2 + \gamma\right]dA\right\} \tag{2}$$

where $\kappa$ is the membrane's bending modulus, $\gamma$ is the membrane tension, $C$ is the local mean curvature, and the integration is taken over the area of the membrane mid plane of the pore, $A_{\mathrm{mb}}$. The shape of the fusion pore is determined by solving a set of differential equations whose solutions minimize the membrane energy subject to the constraints of a fixed pore height $h$ and pore radius $r_{\mathrm{po}}$ (see Appendix 1 subsection 'Numerical method of solving the membrane shape equation'). A term associated with the Gaussian curvature is omitted throughout our analysis because it depends only on the membrane topology. We set the bending modulus to a typical value of $\kappa = 20 k_B T$. We set the value of $\gamma$ to $0.1\,\mathrm{pNnm^{-1}}$, which was obtained as a best fit parameter by comparing model-predicted pore energies with results from a similar experimental setup where fusion between the tCell and the NLP was induced by SNAREs alone (**4**), *Appendix 1—figure 8B*.

## Numerical method of solving the membrane shape equation

We used the MATLAB differential equation solver 'bvp4c' with an absolute tolerance of $10^{-6}$ and a relative tolerance of $10^{-4}$ to solve the membrane shape equation (MathWorks, Natick, MA). This method requires that the equation be rendered as a set of first order ordinary differential equations. The process by which we determined these differential equations is described here.

For simplicity, the fusion pore is assumed to be axisymmetric in our calculations, that is, symmetric under rotations about the $z$-axis. Given this assumption, the membrane energy can be written

$$U_{\mathrm{mb}} = \int_0^{2\pi} \mathrm{d}\theta \int_0^L \mathrm{d}s\, r(s) \left[\frac{\kappa}{2}(2C)^2 + \gamma\right] = 2\pi \int_0^L \mathrm{d}s\, r(s) \left[\frac{\kappa}{2}(2C)^2 + \gamma\right],$$

(3)

where $\theta$ is the azimuthal angle, $s$ measures the arclength along a meridian of the fusion pore (i.e., a curve of constant $\theta$), $r(s)$ is the distance of a given point on the fusion pore from the z-axis, and $L$ is the total arclength of the meridian. We add to this expression two Lagrange multiplier terms which fix the definition of $s$ as the arclength,

$$U_{\mathrm{mb}} = 2\pi \int_0^L \mathrm{d}s\ \left\{ r(s)\left[\frac{\kappa}{2}(2C)^2 + \gamma\right] + f_r(s)(r'(s) - \cos(\phi(s))) \right.$$
$$\left. + f_z(s)(z'(s) - \sin(\phi(s))) \right\}$$

(4)

where $\phi(s)$ gives the angle between the local tangent vector to the meridian and the radially outward direction. The Lagrange multipliers $f_r$ and $f_z$ can be interpreted as the radial and vertical components, respectively, of the force exerted on a curve of constant $z$ due to membrane stress. With this parametrization, the mean curvature can be written

$$C = \frac{1}{2}\left(\phi'(s) + \frac{\sin(\phi(s))}{r(s)}\right).$$

(5)

Inserting this expression for the mean curvature into the membrane energy and taking the functional derivative of the membrane energy with respect to $r(s), z(s), \phi(s), f_r(s)$, and $f_z(s)$, we find a set of differential equations characterizing fusion pore shapes that minimize the membrane energy,

$$r'(s) = \cos(\phi(s))$$
$$z'(s) = \sin(\phi(s))$$
$$\phi'(s) = 2C - \frac{\sin(\phi(s))}{r(s)}$$
$$C'(s) = \frac{f_r(s)\sin(\phi(s)) - f_z(s)\cos(\phi(s))}{2\kappa r(s)}$$
$$f_r'(s) = \gamma + 2\kappa C\left(C - \frac{\sin(\phi(s))}{r(s)}\right)$$
$$f_z'(s) = 0.$$

(6)

These are the Hamilton's equations corresponding to the Lagrangian given by $U_{\mathrm{mb}}$, and are equivalent to the membrane shape equation. The last equation indicates that $f_z$ is a constant; this is associated with the assumed symmetry of the fusion pore. We therefore have five first order differential equations, and two unknown parameters ($f_z$ and $L$), requiring seven boundary conditions.

At the end of the meridian corresponding to the perimeter of the NLP (defined as $s = 0$), the boundary conditions are

$$r(0) = R_{\text{NLP}},$$
$$2\kappa C(0) = k \sin(\phi(s)) \tag{7}$$

where $R_{\text{NLP}}$ is the radius of the NLP, and $k$ is the twisting stiffness of the ApoE scaffold. The latter equation guarantees torque equilibrium between the membrane and the NLP scaffold, described in the Appendix 1 'Mathematical Model of the ApoE scaffold'. At the opposite end of the meridian, where the fusion pore meets the tCell membrane, the boundary conditions are

$$r(L) = R_\infty,$$
$$z(L) = 0, \tag{8}$$
$$\phi(L) = 0.$$

where $R_\infty = 30\,\text{nm}$ is chosen to be much larger than the length scale of the fusion pore. We found that the shape of the fusion pore was not significantly changed by increasing $R_\infty$ to 50 nm or by imposing freely hinged boundary conditions at the tCell interface (i.e. taking $C(L) = 0$), showing that the model is insensitive to these boundary conditions, Figure S8E. The choice $z(L) = 0$ is arbitrary, and is only used to set a reference point for the $z$ coordinate.

We fix the vertical force on the fusion pore $f_z$ in order to vary the height of the pore. The membrane energy is minimized when $f_z = 0$. However, forces created by the Syt-SNARE complex can alter the height of the fusion pore. We therefore scan the parameter $f_z$ from $-6\text{pN}$ to $+6\text{pN}$ to find fusion pores that satisfy the geometric constraint imposed by the Syt-SNARE complex. We assume that the selected height is that which minimizes the free energy subject to the constraints of the Syt-SNARE complex.

Lastly, the contour length of the meridian $L$ is unknown. The selected contour length is the one that minimizes the membrane energy. The condition that $L$ minimizes the membrane energy is given by *Jülicher and Seifert, 1994*:

$$0 = \cos(\phi(L)) + \sin(\phi(L))f_z + r(L)\left(2\kappa C(L)\left[C(L) - \frac{\sin(\phi(L))}{r(L)}\right] - \gamma\right). \tag{9}$$

In order to find the shape of a fusion pore with a given radius $R_{\text{pore}}$, we additionally impose boundary conditions

$$r(x) = R_{\text{pore}}$$
$$\phi(x) = \frac{\pi}{2} \tag{10}$$

at the waist of the fusion pore, between the NLP and the tCell membrane. Note that this is measured from the membrane midplane, and related to the pore radius by $R_{\text{pore}} = r_{\text{po}} + \delta/2$, where $\delta$ is the membrane thickness. Since the location of the waist is not known in advance, this procedure is mathematically equivalent to solving the shape equation twice, once from the NLP to the waist, and once from the waist to the point where the fusion pore joins the tCell membrane.

## Mathematical Model of the ApoE scaffold

We modelled the ApoE scaffold of the NLPs as an elastic rod with rectangular cross-section, a simple representation of the two parallel alpha helices comprising the scaffold. The elastic bending energy of the rod is

$$U_{\text{scaffold}} = \int dL\left(\frac{K_{\text{soft}}}{2}C_{\text{soft}}^2 + \frac{K_{\text{hard}}}{2}C_{\text{hard}}^2\right), \tag{11}$$

where $K_{\text{soft}}$ and $K_{\text{hard}}$ are the bending moduli of the rod in the material directions across its narrow and wide faces, respectively, $C_{\text{soft}}$ and $C_{\text{hard}}$ are the respective material curvatures, and $L$ measures the arclength around the scaffold. The material curvatures represent the curvature of the scaffold projected onto the basis vectors of the material cross section. Because of the rectangular shape of the cross section, the rod is more difficult to bend across its wide face than across its narrower face.

For a homogeneous scaffold with a rectangular cross-section of width $w$ (across the wider face) and thickness $t$ (across the narrower face), this is quantified by a classical result from elasticity theory, which states $K_{\text{soft}} = Et^3w/12, K_{\text{hard}} = Etw^3/12$, where $E$ is the Young's modulus of the scaffold (*Landau and Lifshitz, 1986b*). From these scaling relations, we infer that the scaffold should have twice the bending modulus of a single alpha helix in the soft direction, and 8 times the bending modulus of a single alpha helix in the hard direction, as the scaffold comprises two parallel alpha helices. Given the typical persistence length of an alpha helix of ~100nm and the well-known relation between bending modulus and persistence length $K = k_{\text{B}}TL_{\text{p}}$ (*Choe and Sun, 2005*), we conclude $K_{\text{soft}} \approx 200kT\text{nm}$ and $K_{\text{hard}} \approx 800kT\text{nm}$.

Suppose the cross section of the scaffold is rotated such that the long axis of the cross section makes an angle $\phi$ with the vertical direction (as in Figure S8A), while maintaining the shape of the scaffold as a ring of radius $R_{\text{NLP}} = 12\text{nm}$. Then, the material curvatures are

$$C_1 = \frac{\cos\phi}{R_{\text{NLP}}} \quad C_2 = \frac{\sin\phi}{R_{\text{NLP}}}, \tag{12}$$

This gives an elastic bending energy

$$U_{\text{scaffold}} = 2\pi R_{\text{NLP}} \left( \frac{K_{\text{soft}}}{2} \frac{\cos^2\phi}{R_{\text{NLP}}^2} + \frac{K_{\text{hard}}}{2} \frac{\sin^2\phi}{R_{\text{NLP}}^2} \right). \tag{13}$$

The torque per unit length to twist the scaffold through an angle $\phi$ is thus

$$\tau = \frac{1}{2\pi R_{\text{NLP}}} \frac{\partial U_{\text{scaffold}}}{\partial \phi} = \frac{K_{\text{hard}} - K_{\text{soft}}}{2R_{\text{NLP}}^2} \sin 2\phi = k \sin 2\phi, \tag{14}$$

where $k$ is the apparent twisting rigidity of the scaffold. Using the parameter values above, we find $k = 2.1 k_B T/\text{nm}$. The model predicts that a torque per unit length $\sim 2kT/\text{nm}$ twists the rod ~30 degrees.

We incorporated these effects in our calculation of the fusion pore shape by imposing a torque equilibrium condition at the NLP edge, so that membrane torque is resisted by the ND scaffold. The membrane bending torque per unit length about the local tangent vector to the NLP boundary is given by $\tau = 2\kappa C$, where $C$ is the local mean curvature, neglecting terms associated with the Gaussian curvature modulus (*Deserno, 2015*). Thus, equilibrium is attained when

$$2\kappa C = k \sin 2\phi \tag{15}$$

at the boundary of the NLP.

## Geometric constraints imposed by the Syt-SNARE complex

In order to determine whether a fusion pore determined by solving the membrane shape equation satisfied the constraints imposed by the Syt-SNARE complex, we directly compared the geometry of the fusion pore with that of the Syt-SNARE complex. First, we measured the dimensions of the Syt-SNARE complex and the orientation of the SNARE complex relative to the membrane using PyMol. The long axis of the SNARE formed a ~15° angle with the membrane when docked via the primary interface to the Syt C2B domain in the orientation measured using electron paramagnetic resonance (*Zhou et al., 2015*; *Kuo et al., 2011*). The point where the SNARE complex would contact the membrane (determined by the Syntaxin and VAMP TMDs) was located $d = 8.0\text{nm}$ from the point where the C2B domain inserted into the PM, and the line connecting the two contact points made a $\psi = 39°$ angle with the PM.

To determine if the fusion pore satisfied the relevant constraints, we represented the Syt-SNARE complex as a wedge, with one segment of length 10 nm approximately perpendicular to the membrane, representing the long axis of the SNARE complex. Another segment of length 8 nm at a 17.7° angle to the first represented the line between the two points where the Syt-SNARE complex inserts into the membrane. We scanned this wedge along the meridian of the fusion pore seeking two points $r_1$ and $r_2$ representing the membrane contact points of the Syt-SNARE complex satisfying the following criteria:

1. The distance between $r_1$ and $r_2$ is within 10% of the measured distance between the membrane contact points, $\frac{|r_1 - r_2|}{d} < 0.1$. This 10% error range is roughly comparable with the depth of insertion of the C2B domain Ca-binding loops beyond the phosphate plane (**Kuo et al., 2009**; **Kuo et al., 2011**; **Pérez-Lara et al., 2016**).

2. The line connecting $r_1$ and $r_2$ makes an angle with the membrane tangent vector at $r_2$ within 0.1 radians of the measured value, $\psi - 0.1 < \arccos\left(\frac{r_1 - r_2}{|r_1 - r_2|} \cdot t_2\right) < \psi + 0.1$, where $t_2$ is the membrane tangent vector (pointing along the meridian) at $r_2$, representing the point where Syt contacts the membrane.

3. The long axis of the SNARE complex makes an angle with the membrane normal vector at $r_1$ of less than 0.1 radians.

If all three conditions were satisfied, the pore was assumed to satisfy the constraint imposed by the Syt-SNARE complex.

## Short-ranged steric hydration free energy

The pressure due to short-ranged hydration forces between membranes with separation $d$ follows the form $P_0 \exp(-d/\lambda)$, where $\lambda$ is the characteristic length scale over which the hydration forces decay and $P_0$ is a pressure pre-factor (**Rand and Parsegian, 1989**). The steric hydration free energy was evaluated in our previous work by calculating the work done by the hydration pressure to increase the pore size of a toroidal pore (**Wu et al., 2017b**). The expression for the hydration free energy is given by

$$U_{\text{hyd}} = P_0 \lambda \pi l \exp\left(-\frac{2r_{po}}{\gamma}\right)\left(r_{\text{po}} + \frac{\gamma}{2}\right), \tag{16}$$

where $l = \sqrt{2\lambda(h + 2\delta)}$ is the effective pore height that substantially contributes to the steric hydration interaction. For purposes of determining the hydration energy, we used this expression, approximating the fusion pore as a toroid. Another term giving the work done by the hydration forces to bring two distant planar membranes to a separation $h$ was omitted because it contributed negligibly. The second term is the work done to separate the membranes to form a pore of radius $r_{\text{po}}$. Values for $P_0$ and $\lambda$ are obtained from previous studies and are set to $P_0 = 5 \times 10^{11} \text{dyn/cm}^2$ and $\lambda = 0.1\text{nm}$ (see **Appendix 1—table 1**).

## Free energy of SNAREs

We assume that each side of the NLP contains $N$ v-SNAREs that are all available to associate with the t-SNAREs in the tCell membrane and contribute to pore expansion. SNAREs can be fully zippered, where their TMDs are circularly arranged near the fusion pore waist. Alternatively, they can adopt a partially zippered configuration, where the TMD and linker domain are unzipped, and the v-SNARE and t-SNARE TMDs are located on the NLP and tCell membranes, respectively, but on the same side of the fusion pore. We denote the number of fully and partially zippered as $N_{\text{Z}}$ and $N_{\text{UZ}}$, respectively. The SNARE free energy in the fully zippered state reads

$$U_{\text{Z}}\left(r_{\text{po}}, N_{\text{Z}}\right) = -N_{\text{Z}}k_{\text{B}}T\left[\ln\frac{2\pi r_{\text{po}} - N_{\text{Z}}b}{N_{\text{Z}}b} + 1\right] - N_{\text{Z}}k_{\text{B}}T \ln \Omega_{\text{Z}} - N_{\text{Z}}\epsilon_{\text{Z}}, \tag{17}$$

where $k_{\text{B}}$ is the Boltzmann constant, $T$ is the temperature and $b = 2\text{nm}$ is the thickness of a single SNARE (**Wu et al., 2017b**). The first term in **Equation (17)** is the positional entropy of the zippered SNAREs TMDs. The second term is the orientational entropy associated with the zippered SNAREs. We assume that these are stiff rods that can explore a small solid angle of $\Omega_{\text{Z}} = 0.05\text{sr}$, based on molecular dynamics studies of t-SNARE TMDs showing that these domains explore angles of 10° around their equilibrium position (**Knecht and Grubmüller, 2003**). We assume that for the zippered SNAREs the equilibrium orientation is the local normal to the fusion pore membrane.

The last term is the total energy released when the TMDs and the adjacent linker regions of $N_{\text{Z}}$ SNAREs are fully zippered, where $\epsilon_{\text{Z}}$ is the zippering energy per SNARE. This zippering energy was obtained as a best-fit parameter in a previous study where fusion was induced between NLPs and tCells with only SNAREs (**Fulop et al., 2005**). The best-fit value of 9.6 $k_{\text{B}}T$ (**Appendix 1—table 1**) is

higher than the ~5 $k_B T$ that we estimated from a previous study as the zippering energy of the linker domains (*Gao et al., 2012*), as explained in the following paragraph.

The linker domain (LD) has ~10 residues (*Stein et al., 2009*), and is thus ~ 3 nm in length, assuming an unfolded contour length 0.3 nm per residue. Previous measurements show the free energy to unzip the SNAREs has slope ~1.5 $k_B$T per nm when the LDs are being unzipped (*Gao et al., 2012*). Thus, we estimate the LD unzippering energy from Gao et al. is ~ 5 $k_B$T.

The SNARE free energy in the partially zippered state reads

$$U_{UZ}(r_{po}, N_{UZ}) = -N_{UZ} k_B T \ln \frac{2\pi D}{b} - N_{UZ} k_B T \ln \Omega_{UZ}.$$

(18)

The first term in *Equation (18)* is the positional entropy of the TMDs, while the second term is the orientational entropy associated with a solid angle $\Omega_{UZ}$ explored by the SNAREs. In the partially zippered state the SNARE linker domains are assumed to be unstructured, which allows them to adopt all orientations where they are not intersecting with the membranes. Since this orientational freedom is available when the SNAREs are away from the fusion pore, we restrict their position to the edge of the pore and set $\Omega = \Omega_{UZ} = \pi$.

We assume that, in elevated Ca concentrations, the C2B domains will bind the membrane via their Ca-binding loops. This lifts the C-terminal end of the SNARE complex ~5 nm above the tCell membrane; this has been proposed to drive dissociation of SNARE complexes from Syt in the presence of Ca (*Voleti et al., 2020*; *Grushin et al., 2019*). As this dissociation would cost $10 - 12 k_B T$ (see subsection Pushing forces from the membranes are insufficient to disrupt the SNARE-Syt primary-interface interaction), we omit this possibility and assume that SNAREs are unable to explore the fusion pore in elevated Ca concentrations. In this case, the positional entropy of SNAREs is unaltered by unzipping, and the free energy difference between the zippered and unzipped state is therefore given by $\epsilon_Z$ per zippered SNARE.

## Total free energy as a function of pore size, minimum membrane separation, and total number of SNAREs

To obtain the free energy of a fusion pore with a radius $r_{po}$, we numerically summed all the Boltzmann factors of all possible states according to

$$\exp\left(-\frac{U_{tot}(r_{po}, h, N)}{k_B T}\right) = \sum_{N_Z = 0}^{N} \exp\left(-\frac{U_{tot}(r_{po}, h, N, N_Z)}{k_B T}\right)$$

$$U_{tot}(r_{po}, N) = \min_h \left\{ U_{tot}(r_{po}, h, N) \right\}$$

(19)

where we set the number of SNAREs to $N = 4$ to match experiment.

### Model-predicted pore conductance

Consider an axially symmetric fusion pore whose inner surface is described by a function $r(z)$ which gives the distance from the axis to the luminal surface of the membrane. The resistance of the pore is given by

$$R_{po} = \rho \int_0^L \frac{dz}{\pi r(z)^2}$$

(20)

where $\rho$ is the resistivity of the solution in the pore lumen, and the height of the pore lumen $L = h + 2\delta$, where $\delta$ is the thickness of the bilayer (*Nanavati et al., 1992*). We numerically evaluated the above integral for fusion pores determined by solving the membrane shape equation. The total resistance of the pore also has a contribution from access resistance given by *Nanavati et al., 1992*,

$$R_{acc} = \frac{\rho}{2r_{po} + L}.$$

(21)

The pore conductance is then given by $G_{po} = (R_{po} + R_{acc})^{-1}$.

## Pushing forces from the membranes are insufficient to disrupt the SNARE-Syt primary-interface interaction

We show here that forces needed to separate the membranes and expand the fusion pore are not large enough to disrupt the SNARE-Syt primary interface.

To calculate the pushing forces shouldered by the SNARE-Syt complexes, we calculated increase in the pore free energy $U_{\mathrm{mb}}$ caused by the expansion driven by the SNARE-Syt complexes as a function of pore radius (Fig. S8D). From Fig. S8D, we see that change in height costs ~0-10 $k_B$T. The forces increasing the height of the pore are $\partial U_{\mathrm{mb}}/\partial h \approx 10\mathrm{pN}$. This force is shared across 4 SNARE-Syt complexes. Thus, an estimate for the force shouldered per complex during pore expansion is ~2.5 pN.

We estimate that 2.5 pN is far less than the force needed to break the primary C2B-SNARE complex interface. Reported dissociation constants for the SNARE-C2B complex are 0.86 μM (**Wang et al., 2003b**) and 14 μM (**Wang et al., 2001**). These correspond to binding energies ΔG of 12 $k_B$T and 10 $k_B$T, respectively, after estimating a microscopic capture radius of 2 nm (equivalently a reference concentration 0.21M) and using ΔG = -k_BTlnKd/0.21M. Even if we conservatively use a large 'unbinding distance' $d$ ~ 3 nm, breaking the interface thus requires a force of order ΔG/$d$ ~14–16 pN, much larger than the ~4 pN force exerted on a lever complex by the fusion pore. Thus, we expect the SNARE-Syt complexes will remain intact.

**Appendix 1—table 1.** Parameters used in the mathematical model with coarse-grained membranes, SNAREs and Synaptotagmin-1 C2B domains.

(A) Measured in **Mitra et al., 2004**. (B) Estimated previously as a best fit model parameter to experiments where NLP-tCell fusion pore was induced only by SNAREs (**Wu et al., 2017b**). (C) Consistent with NLP diameter measurements in this study. (D) Calculated as the weighted average of the hydration pressures of palmitoyl-2-oleoyl phosphatidylcholine (POPC) and 1,2-dioleoyl phosphatidylserine (DOPS) adopted from **Rand and Parsegian, 1989**, using a (85:15) molar ratio of POPC:DOPS as present in the NLP in the current study. (E) Values for the bending modulus range between $10 - 50k_BT$ (**Brochard and Lennon, 1975**; **Khelashvili et al., 2013**; **Marsh, 2006**; **Cohen and Melikyan, 2004**). We used a value of $\kappa = 20k_BT$. (F) Obtained by fitting the membrane energy as a function of pore radius to measurements from a previous study using a similar method to measure the pore free energy in the absence of SNAREs, see Figure S8B (**Wu et al., 2017b**). (G) Calculated based on a 10° angle explored by t-SNARE TMDs around the equilibrium configuration, as measured in molecular dynamics simulations (**Knecht and Grubmüller, 2003**). (H) Estimated here from the ~100nm persistence length of typical alpha helices and cross-sectional dimensions of the ApoE scaffold (see Appendix 1 subsection 'Mathematical Model of the ApoE scaffold'). (I) Measured in Syt1-liposome binding assays (**Bai et al., 2004a**; **Bai et al., 2004b**)

| Symbol | Meaning | Value | Legend |
|---|---|---|---|
| $\delta$ | Membrane thickness | 5nm | (A) |
| $\epsilon_{\mathbf{Z}}$ | Zippering energy of SNARE's linker and TMD domains | $9.6k_BT$ | (B) |
| $D$ | NLP diameter | 24nm | (C) |
| $\lambda$ | Hydration interactions decay length | 0.1nm | (B) |
| $P_0$ | Pressure pre-factor for steric hydration interaction | $5 \times 10^{11}\mathrm{dyn/cm^2}$ | (D) |
| $\kappa$ | Membrane bending modulus | $20k_BT$ | (E) |
| $\gamma$ | Membrane tension | $0.1\mathrm{pN \cdot nm^{-1}}$ | (F) |
| $\Omega_{\mathbf{Z}}$ | Solid angle explored by fully zippered SNAREs | 0.05sr | (G) |
| $k$ | Twisting rigidity of ApoE scaffold | $600k_BT \cdot \mathrm{nm}$ | (H) |
| $[\mathbf{Ca^{2+}}]_{1/2}$ | Apparent affinity of Syt1 to calcium in the presence of PIP2 containing membranes | $23\mu$M | (I) |
| $n$ | Hill coefficient | 2.3 | (I) |

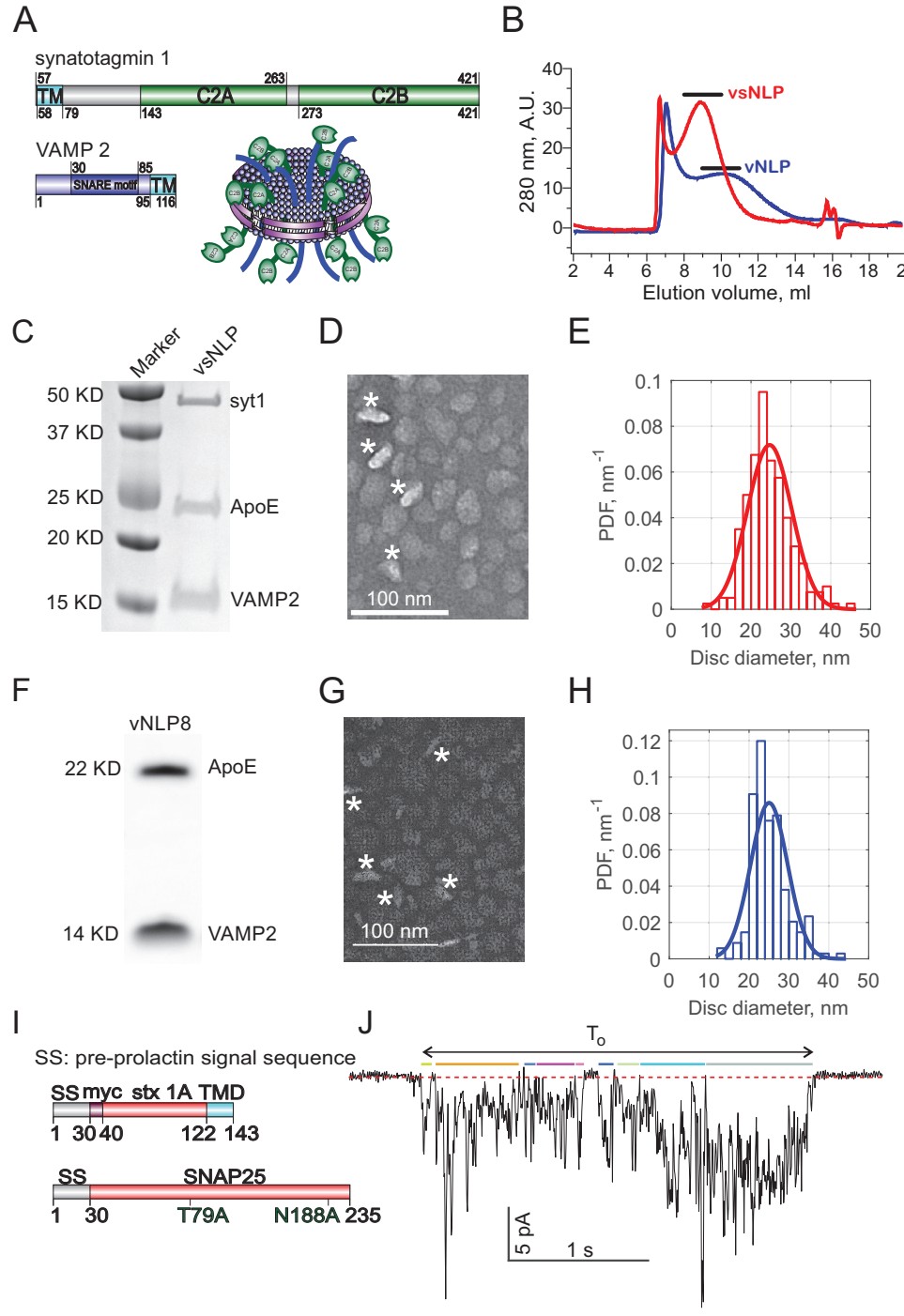

**Appendix 1—figure 1.** Co-reconstitution of Synaptotagmin-1 and VAMP2 into nanolipoprotein particles (vsNLPs). (**A**) Schematic of an NLP reconstituted with 4 copies per face each with Syt1 and VAMP2. Syt1 C2AB domains are shown in green; VAMP2 is shown in blue and the scaffold protein ApoE protein is shown in cyan. Domain structures of Syt1 and VAMP2 are indicated. (**B**) Typical size-exclusion chromatography elution profiles of vNLP (NLPs loaded with VAMP2 alone) and vsNLP (NLPs loaded with VAMP2 and Syt1) samples using a Superose 6 Increase 10/300 GL column. Proteins were detected using absorbance at 280 nm. Collected eluted volumes are indicated by horizontal bars above each profile. (**C**) SDS-PAGE stained with Coomassie Brilliant Blue shows the purified NLPs carried Syt1 and VAMP2 proteins. (**D**) A representative transmission electron

*Appendix 1—figure 1 continued on next page*

*Appendix 1—figure 1 continued*

microscopy (TEM) image of a vsNLP sample after purification. Nanodiscs indicated by white stars have their lipid bilayer plane positioned perpendicularly to the imaging plane. (**E**) Distribution of vsNLP sizes from TEM images. More dilute samples (5-10x) than the example shown in D were used for size quantification, such that most NLPs were lying flat on the grid. A Gaussian fit to the distribution is shown as the red solid line (fitted mean diameter = 25±5.6 nm (± SD), n = 200 NLPs). (**F–H**) Characterization of nanolipoprotein particles reconstituted with VAMP2 alone. (**F**) SDS-PAGE of purified vNLPs stained with Coomassie Brilliant Blue indicting the NLPs incorporated ApoE and VAMP2 proteins. (**G**) Negative stain transmission electron microscopy image of a representative vNLP sample. Nanodiscs indicated by white stars are oriented with their disc plane perpendicular to the imaging plane. (**H**) Size distribution of vNLPs reconstituted with a total of 8 copies of VAMP2 per disc. A Gaussian fit is shown as the red solid line (best fit diameter = 25 ± 4.6 nm (mean ± SD), 171 discs were analyzed). (**I**) Domain structure of flipped t-SNARE constructs used to generate HeLa cells stably expressing flipped t-SNAREs (*Giraudo et al., 2006*; *Giraudo et al., 2005*). (**J**) Example of a fusion pore current burst and definition of analysis parameters. To be included in the analysis, a current burst must have amplitude >2 pA and last at least 250 ms. Open sub-periods during a burst cross a threshold (-0.25 pA, red dotted line) for at least 60 ms (indicated as the thick colored bars above the current trace). The number of open sub-periods during a burst is equal to the number of pore flickers, $N_{flickers}$. The duration of the burst is the time from the first detected open-pore point until the last one and is denoted $T_o$. The flicker rate is $N_{flickers}/T_o$. The pore open probability is the sum of the open-pore sub-periods (the colored bars) divided by the burst lifetime, $T_o$.

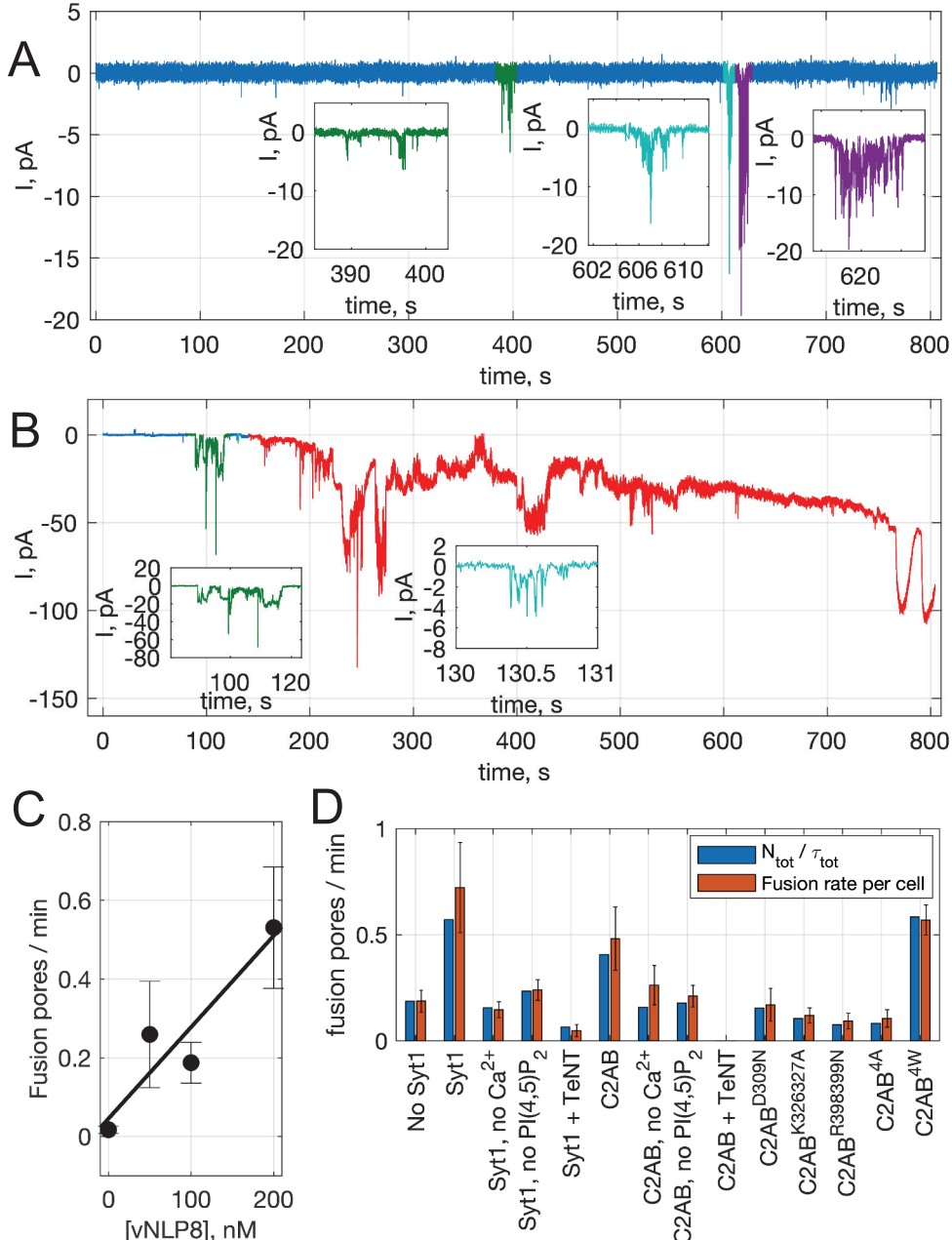

**Appendix 1—figure 2.** Fusion rates. (**A**) An example of an entire 800 s recording that started shortly after establishing a tight seal. The colored regions correspond to currents that are counted as fusion pore currents because they fit the criteria explained in Appendix 1 *Detection of pore currents* and *Estimation of fusion rate* (also see Fig. S1J). These regions are shown with expanded axes in insets. In this example, 3 pores were counted in 800 s of recording. (**B**) Another example of an 800 s recording. In this case, large currents appeared starting ~140 s. The baseline did not recover before the end of the recording and the red-colored portion was excluded from analysis. Thus, two current bursts contributed to the fusion rate from this trace (starting ~90 and 130 s, colored in green and teal), from 140 s of recording. (**C**) Fusion rate increases with increasing NLP concentration. Fusion rates were calculated as described in Appendix 1 *Estimation of fusion rate* and plotted against the concentration of NLPs reconstituted with v-SNAREs (8 copies total). Pores from 12-37 cells were recorded for every condition. Error bars indicate S.E.M. The best fit straight line is

*Appendix 1—figure 2 continued on next page*

*Appendix 1—figure 2 continued*

shown (slope = $2.3 \times 10^{-3}$ pores/(min•nM), $R^2 = 0.86$). (**D**) Comparison of per cell and overall fusion rates. As an alternative estimate of the fusion rate, we summed all detected pores, $N_{tot}$, and the analysis time $\tau_{tot}$ over all cells (excluding portions with noisy/unstable baseline), and calculated the total number of pores divided by the total analysis time, $F_{tot} = N_{tot}/\tau_{tot}$ for the indicated conditions. This estimate (blue) is compared with the per cell estimate used throughout (red).

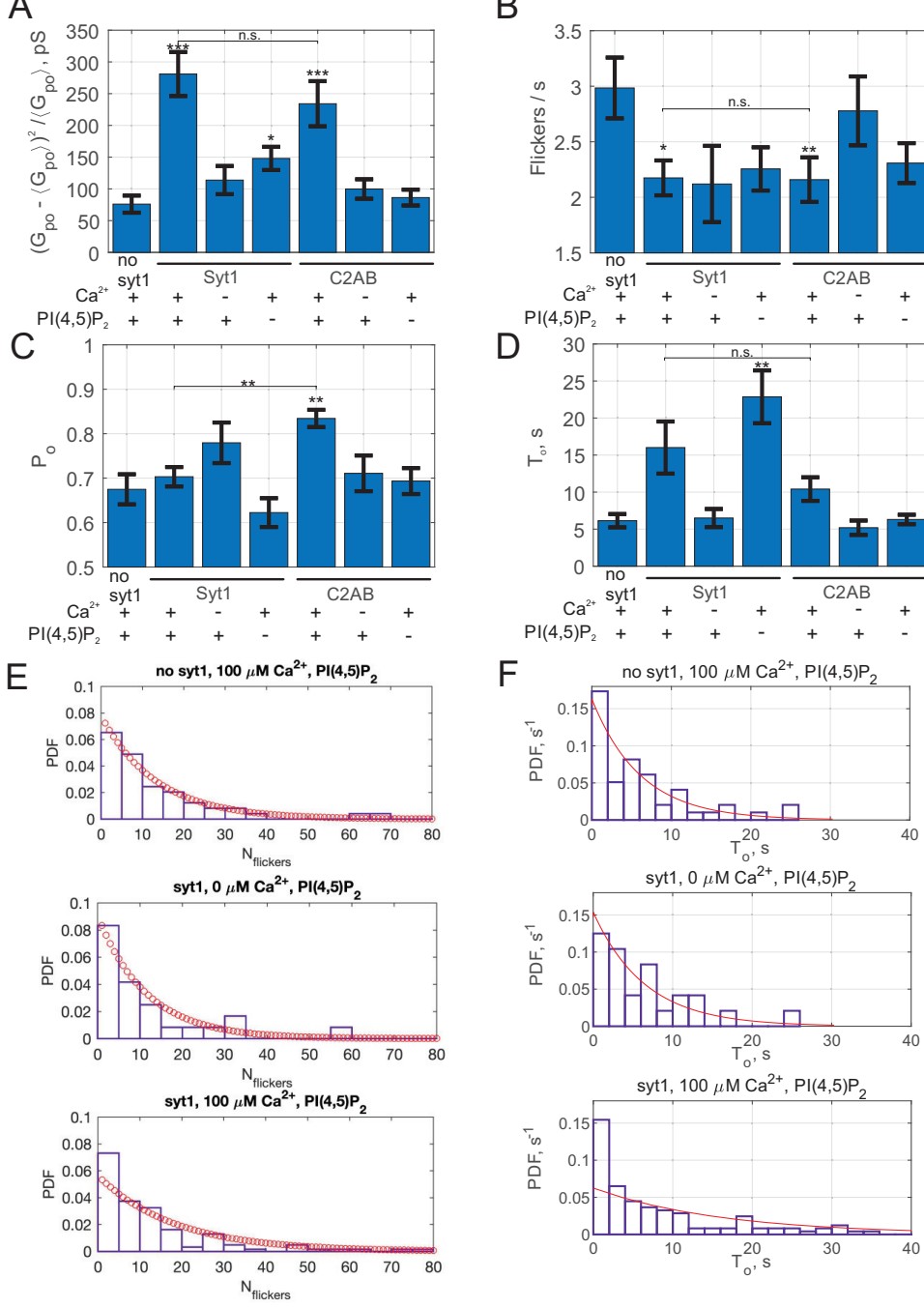

**Appendix 1—figure 3.** Additional properties of single fusion pores in the presence of full-length

*Appendix 1—figure 3 continued on next page*

*Appendix 1—figure 3 continued*

Syt1 or soluble C2AB. Open-pore conductance fluctuations relative to mean (**A**), average flicker rate during a burst (**B**), average open-pore probability, $P_o$, during a current burst (fraction of time pore is in the open state during a burst) (**C**), and average burst lifetime, $T_o$, (**D**) for the indicated conditions. (**E**) Distributions of the number of flickers per burst, $N_{flickers}$, for the indicated conditions. Fits to geometric distributions are shown in red, $y = p(1-p)^{n-1}, n = 1, 2, 3, \ldots$. Best fit parameters (with $\pm$ 95% confidence intervals) are $p = 0.072(0.053, 0.092)$ (no Syt1, 100 $\mu$M Ca$^{2+}$, averaged over 49 individual fusion pores from 10 cells, mean $N_{flickers} = 12.8$), $0.083(0.051, 0115)$ (Syt1, 0 $\mu$M Ca$^{2+}$; averaged over 24 individual fusion pores from 11 cells, mean $N_{flickers} = 11.0$), $0.053(0.044, 0.063)$ (Syt1, 100 $\mu$M Ca$^{2+}$; averaged over 123 individual fusion pores from 20 cells, mean $N_{flickers} = 17.7$). (**F**) Distribution of burst lifetimes, $T_o$ for the indicated conditions. Best fits to single exponentials are shown as red curves, with means (and 95% confidence intervals) as follows. No Syt1, 100 $\mu$M Ca$^{2+}$: 6.1 s (4.7 to 8.3 s, 49 fusion pores from 10 cells), Syt1, 0 $\mu$M Ca$^{2+}$: 6.5 s (4.5 to 10.1 s, 24 fusion pores from 11 cells), Syt1, 100 $\mu$M Ca$^{2+}$: 16 s (13.5 to 19.3 s, 123 fusion pores from 20 cells). In A-D, the two-sample Kolmogorov-Smirnov test was used to assess significant differences between the "no C2AB" group and the rest. *, **, *** indicate p<0.05, 0.01, and 0.001, respectively. Comparison between Syt1 and C2AB in the presence of Ca$^{2+}$ and PI(4,5)P$_2$ are also indicated (using the two-sample Kolmogorov-Smirnov test).

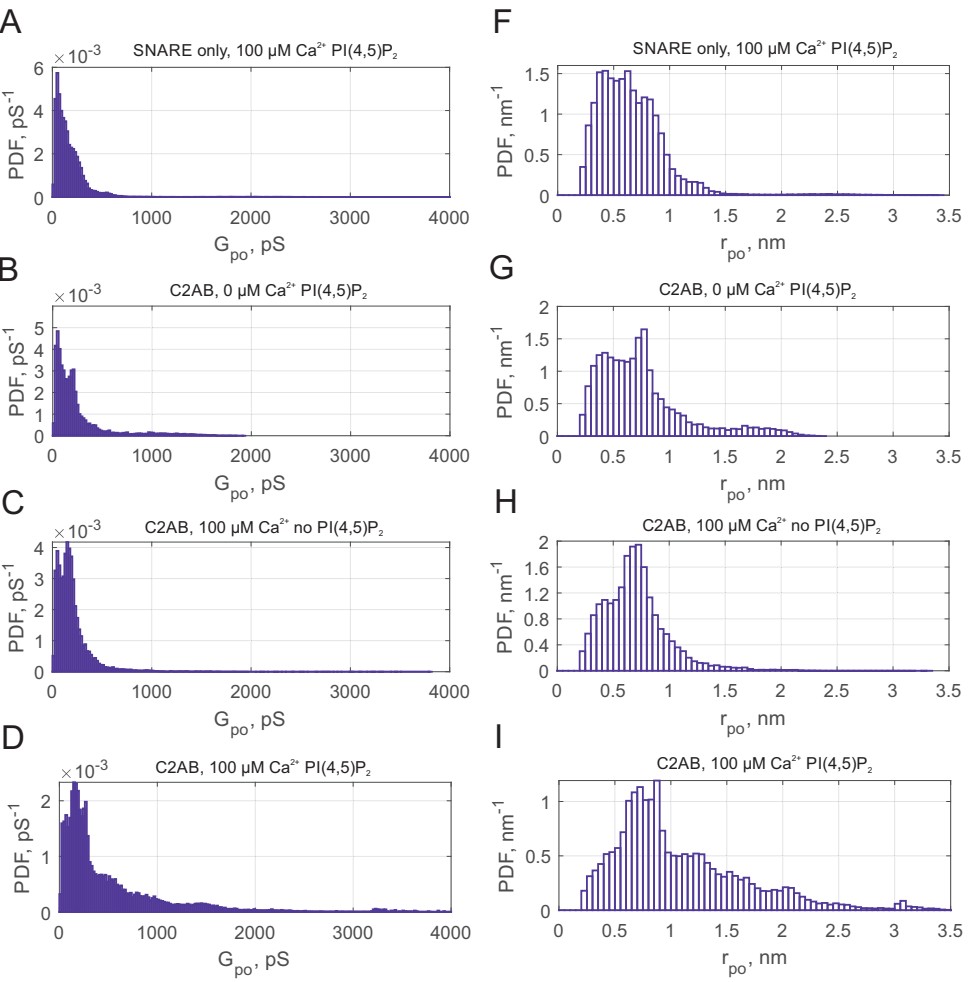

**Appendix 1—figure 4.** Syt1 C2AB dilates fusion pores in a calcium and PI(4,5)P$_2$ dependent manner. (**A–D**) Probability density function (PDF) for point-by-point open-pore conductance values for the indicated conditions. Substantial density is present for $G_{po} \geq 500$ pS only when C2AB, calcium, and PI (4,5)P$_2$ were all present. (**F–I**) PDFs for open-pore radii corresponding to the conductance distributions in A-D, assuming pores are 15 nm long cylinders. Data were from 49 fusion pores/10 cells (SNARE only), 44 fusion pores/12 cells (0 µM Ca$^{2+}$), 84 fusion pores/19 cells (no PI(4,5)P$_2$) and 98 fusion pores/17 cells (100 µM Ca$^{2+}$ plus PI(4,5)P$_2$).

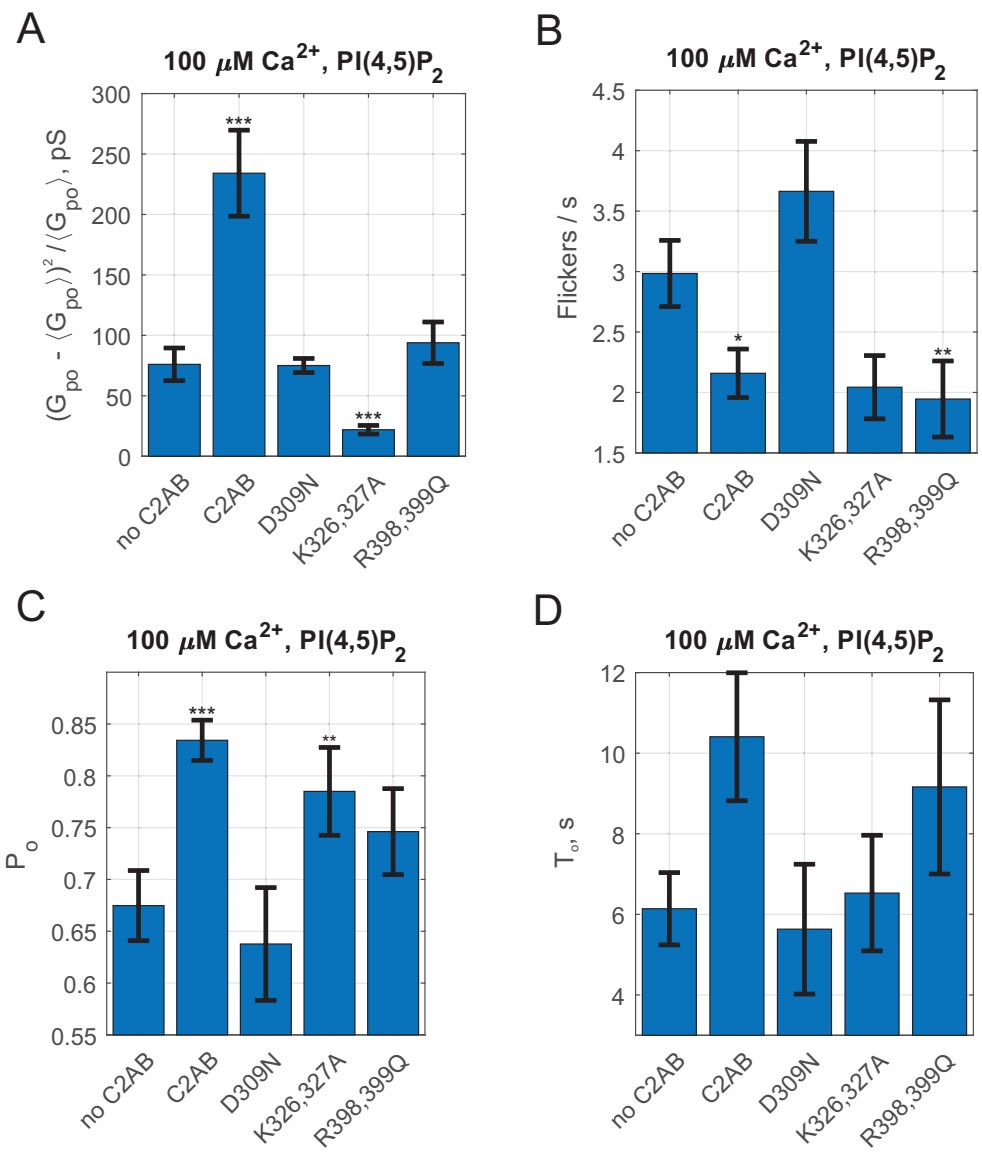

**Appendix 1—figure 5.** Additional fusion pore properties for Syt1 C2AB domains carrying mutations in D309, K326-327 and R398-399. (**A**) Open-pore conductance fluctuations relative to mean. Compared with the SNARE-alone (no C2AB) group, fluctuations were larger for wild-type C2AB, and lower for C2AB$^{K326A,327A}$. (**B**) Average flicker rate for the same conditions as in A. Compared with the SNARE alone group (no C2AB), C2AB and C2AB$^{R398Q,R399Q}$ decreased the flicker rate. (**C**) Average pore open probability during a burst, $P_o$, for the indicated conditions. Compared with the
*Appendix 1—figure 5 continued on next page*

SNARE alone group (no C2AB), C2AB and C2AB[K326A,K327A] had larger pore open probabilities. (**D**) Average burst lifetimes for the same conditions. Error bars are +/- S.E.M. Data were from no C2AB: 49 pores/10 cells, C2AB: 98 pores/17 cells, C2AB[K326A,327A]: 42 pores/14 cells, C2AB[D309N] (18 pores/ 7 cells), C2AB[R398Q,R399Q] (42 pores/18 cells). For A-D, two-sample Kolmogorov-Smirnov test was used to assess significant differences between the "no C2AB" group and the rest. *, **, *** indicate p<0.05, 0.01, and 0.001, respectively.

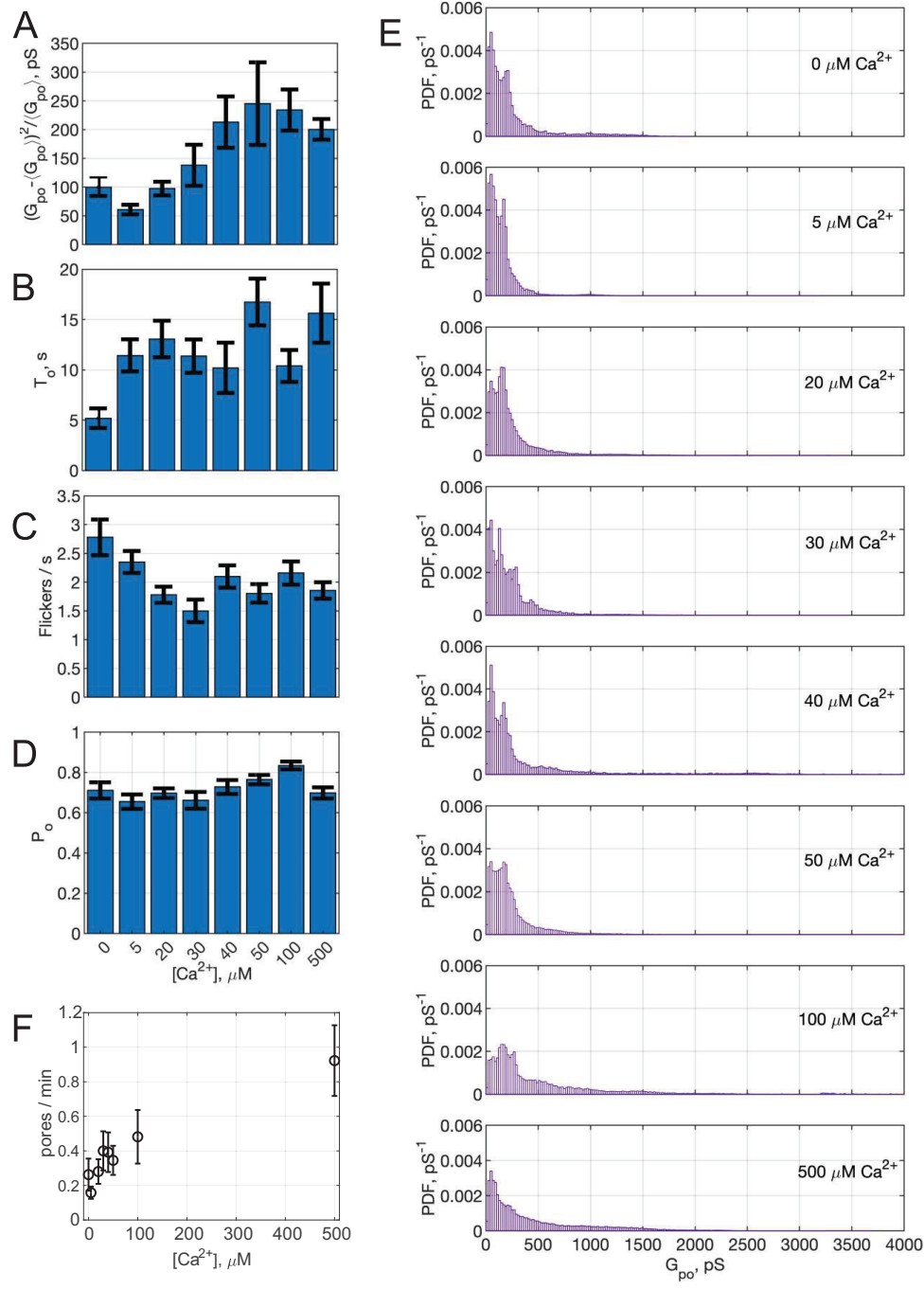

*Appendix 1—figure 6 continued*

**Appendix 1—figure 6.** Additional fusion pore properties as a function of free calcium concentration, $[Ca^{2+}]_{free}$. (A–D) Average single-pore conductance fluctuations relative to mean (A), average burst lifetime (B), average flicker rate (C), and the pore open probability $P_o$ (D) as a function of $[Ca^{2+}]_{free}$. Error bars are ± S.E.M. (E). Probability density functions (PDFs) for point-by-point open-pore conductance values at different $[Ca^{2+}]_{free}$. The probability density for $G_{po} \geq 500$ pS increases as a function of calcium. (0 µM $Ca^{2+}$: 44 pores/12 cells; 5 µM $Ca^{2+}$: 54 pores/20 cells; 20 µM $Ca^{2+}$: 114 pores/18 cells; 50 µM $Ca^{2+}$: 88 pores/26 cells; 100 µM $Ca^{2+}$: 98 pores/17 cells). (F) The fusion rate increases as a function of $[Ca^{2+}]_{free}$. For A-D, two-sample Kolmogorov-Smirnov test was used to assess significant differences between the "no C2AB" group and the rest. *, **, *** indicate $p<0.05$, 0.01, and 0.001, respectively.

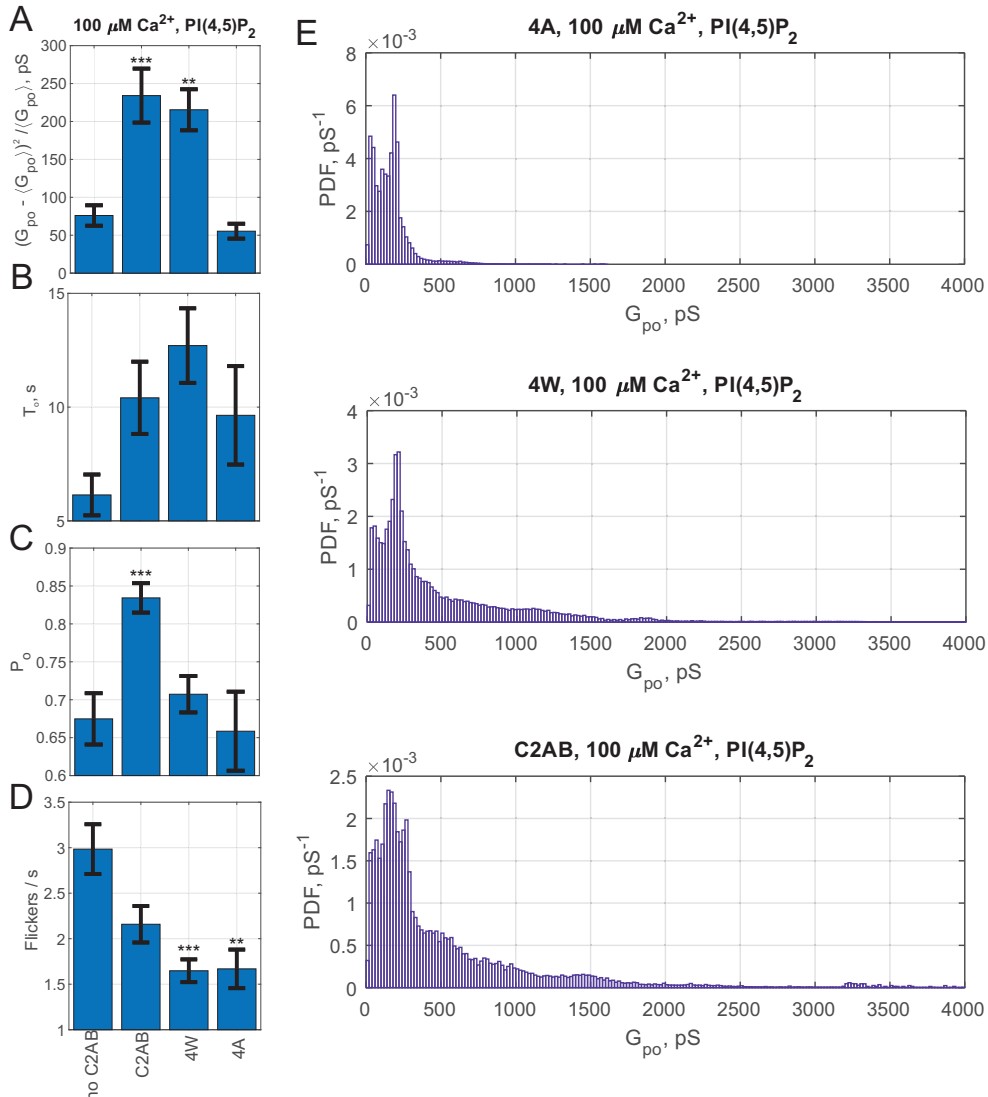

**Appendix 1—figure 7.** Additional fusion pore properties for Syt1 C2AB membrane penetration mutants. (A–D) Average single-pore conductance fluctuations relative to mean (A), burst lifetime (B), pore open probability during a burst, $P_o$ (C), and flicker rate (D) for SNAREs alone (no C2AB), wild-type Syt1 C2AB (C2AB), the 4W mutant with enhanced membrane-penetration ability (M173W,

*Appendix 1—figure 7 continued on next page*

*Appendix 1—figure 7 continued*

F234W, V304W and I367W), and the 4A mutant which cannot penetrate membranes in response to calcium (M173A, F234A, V304A and I367A). Error bars are ± S.E.M. (**E**) Probability density functions (PDFs) for point-by-point open-pore conductance values for wild-type C2AB, and the membrane penetration mutants 4A and 4W. (4W: 115 pores/20 cells, 4A: 31 pores/9 cells). For A-D, two-sample Kolmogorov-Smirnov test was used to assess significant differences between the "no C2AB" group and the rest. *, **, *** indicate p<0.05, 0.01, and 0.001, respectively.

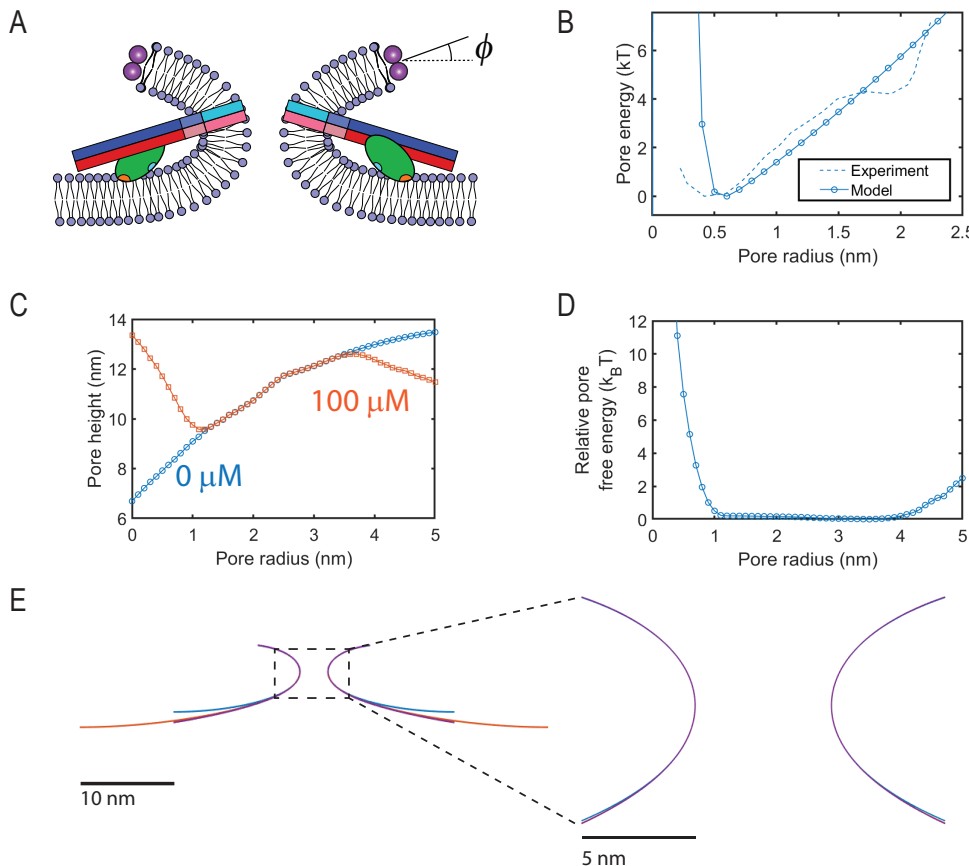

**Appendix 1—figure 8.** Results of the mathematical model of the fusion in the presence of SNAREs and Syt1 C2AB domains. Data in (**C**) and (**D**) was smoothed using a moving average with width 0.5 nm. (**A**) Schematic illustrating a fusion pore with buried SNARE-Syt1 levers. $\phi$ represents the angle of twisting of the ApoE proteins. (**B**) Pore free energy as a function of radius predicted by the model and measured in a previous study (*Wu et al., 2017b*); the membrane tension was tuned to reproduce the experimental curve here. (**C**) Pore height, defined as the maximal separation between the NLP and the tCell membranes, as a function of pore radius with and without $Ca^{2+}$. (**D**) Free energy difference between pores in the expanded state and those in the unexpanded state. Syt-SNARE complex-driven pore expansion costs a maximum of $\sim 10 - 12k_BT$ at low pore radii. (**E**) Fusion pores determined from the model using varying boundary conditions at the location where the fusion pore membrane joins the tCell (lower edge). The blue curve shows the boundary conditions used throughout the paper, while the orange curve uses $R_\infty = 50\text{nm}$ (instead of 30 nm), and the purple curve imposes freely hinged boundary conditions at this location. These changes have very little effect, especially in the region close to the fusion pore (zoom in, right).

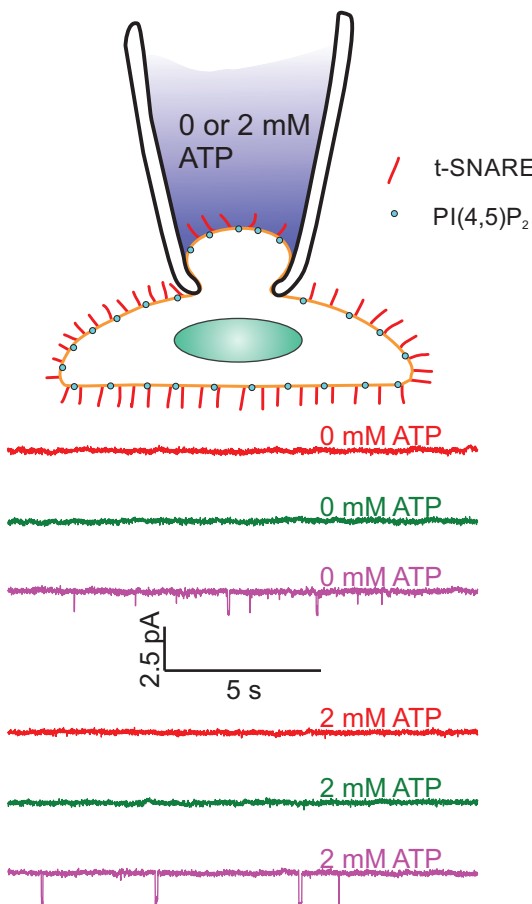

**Appendix 1—figure 9.** Lack of ATP-regulated channel activity in flipped t-SNARE cells. Top: Schematic of the cell-attached recordings to test for ATP-regulated channel activity. Pipette solutions were the same as for single-pore measurements with 100 µM free calcium, but adjusted to contain either 0 or 2 mM ATP. Pipette potential was −40 mV. Middle: Current recordings under voltage clamp from three different patches (out of 21 total) in the absence of ATP. Bottom: Current recordings as in the middle panels, but in the presence of 2 mM freshly prepared ATP (26 cells were recorded). Occasionally, channel activity is recorded both with and without ATP.

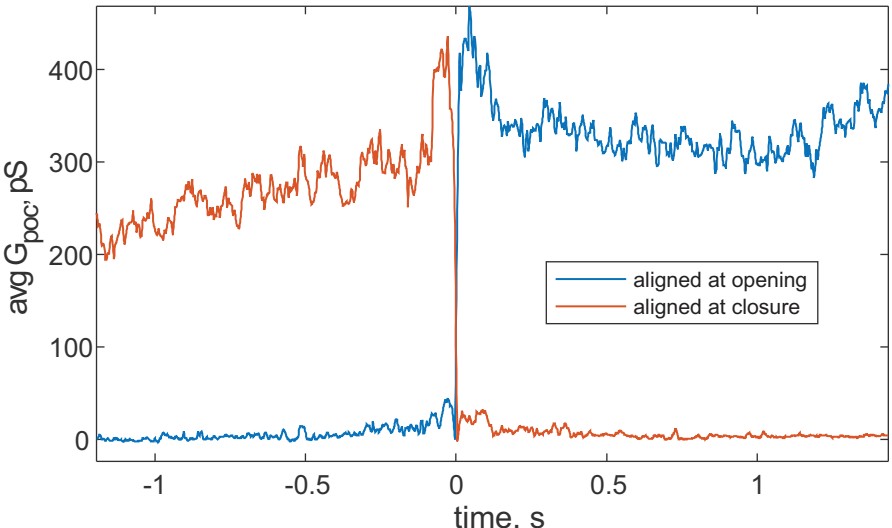

**Appendix 1—figure 10.** Average pore conductance as a function of time, after aligning pores to the moment of opening (blue) or closure (red). Data for full-length Syt1, in the presence of 100 μM free Ca$^{2+}$ and PI(4,5)P$_2$. Other conditions also failed to yield large differences between pore opening or closure.

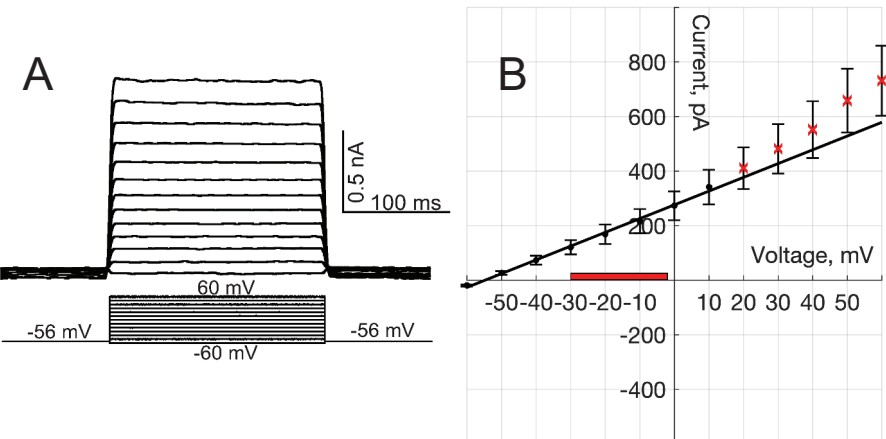

**Appendix 1—figure 11.** Whole-cell conductance of flipped t-SNARE HeLa cells. (**A**) Whole-cell current responses to step changes in membrane potential under voltage-clamp, from a HeLa cell line expressing flipped t-SNAREs. (**B**) Current-voltage relationship. Currents were averaged for 27 cells. The average slope is $G_{cell} = 5.04 \pm 0.32$ nS (95% confidence interval), excluding the five highest voltages (red crosses). If all points are included, $G_{cell} = 6.20 \pm 0.51$ nS (95 % confidence interval). The range of mean open-pore currents and transmembrane voltages comprising 95% of the data values for C2AB in the presence of PI(4,5)P$_2$ and 100 $\mu$M calcium are indicated as a red-colored box.

