## [Decision Letter]

**Acceptance summary:**

This paper will be of interest to neuroscientists, and more broadly to scientists working on membrane fusion. A combination of experiments fusing an elegant nanodisc-cell fusion assay and of mechanical models reveals a new mechanism by which Synaptotagmin-1 mechanically promotes the opening of the fusion pore induced by SNARE proteins.

**Decision letter after peer review:**

[Editors’ note: the authors submitted for reconsideration following the decision after peer review. What follows is the decision letter after the first round of review.]

Thank you for submitting your work entitled "The neuronal calcium sensor Synaptotagmin-1 and SNARE proteins cooperate to dilate fusion pores" for consideration by *eLife*. Your article has been reviewed by 3 peer reviewers, one of whom is a member of our Board of Reviewing Editors, and the evaluation has been overseen by a Senior Editor.

Having considered your revision plan. our decision has been reached after consultation between the reviewers. Based on these discussions and the individual reviews below, we regret to inform you that your work will not be considered further for publication in *eLife*.

Two reviewers read the author response. Their comments are presented below. In brief, the reviewers and the reviewing editor understand that the additional experiments, which are necessary for revision of the experimental part of the article, could be hard to perform in the near future because of the pandemic. At the same time, the author replies to the reviewer comments on the theoretical part of the work, do not appear satisfactory.

Computational part

"In cell-attached recordings, the patch of membrane that is under study is under high tension due to adhesion to the walls of the pipette. Strong adhesion of the membrane to the walls of the glass pipette is needed to obtain a high-resistance seal (the so-called "Gigaseal") for low-noise recordings. Previous measurements show the membrane tension in the patch is ~1 pN/nm [37]. This is consistent with our model that obtained a tension of 0.66 pN nm-1 as a best-fit parameter by comparing the model-predicted slope of the free energy of pore dilation for pores of size ~ 1 nm between the model prediction and the experimental measurement when only SNAREs were present in a previous study of ours [3].

Forces that oppose membrane tension at the nanodisc boundary arise presumably from the bending rigidity of the ApoE proteins, and from their interactions with the lipids. A radially inward force per unit length equal to the membrane tension acts on the ApoE scaffold. Assuming only the bending rigidity opposes this inward force, we can compare the membrane tension to the threshold force per unit length where a circular ring buckles, kT Lp /d2 , where : and d and Lp are the radius of the cross-section of the circular ring and its persistence length respectively. As the scaffold proteins consist of a series of α-helical regions interspersed with unstructured regions [38], and a typical persistence length of α-helices is ~ 100 nm [39], the buckling threshold is ~400 pN nm-1 assuming a thickness : d~ 1 nm, much larger than the membrane tension of 0.66 pN nm-1"

The should be a misunderstanding here. According to the classical literature on the buckling instability of a ring, the radial force per unit length of the ring (the membrane tension in the present case) leading to the instability is, approximately, γ~(kT L_p_)/r^3^ , where r is the ring radius. Taking Lp = 100nm, as suggested by the authors, and r~10"nm" , according to the dimension of nanodiscs used in the work, one gets, practically, a vanishing critical tension. This must mean that, as a result of the pore nucleation, the tension drops to zero.

This means that the model is inconsistent with the reality in this essential point.

"We agree that the toroidal shape is not an exact fusion pore solution, but we use this shape for simplicity, and we believe it captures all the important features of a more exact solution. It has the advantage that the biophysical aspects are clearly articulated. For this reason, toroidal pores have been assumed in many previous theoretical studies (e.g. [40-42])."

This is not a convincing argument. In the previous studies the toroidal pore approximation was used for the cases of two infinitely large membranes connected by a fusion pore. In the present study, the dimension of the nanodisk is of the same order of magnitude as the pore radius so that the deviation from the toroidal shape may have essential consequences for the results.

One of the consequences of the toroidal pore assumption is the conclusion that SNARE reorientation results in the expansion of the pore waist, which is one of the central conclusions of the model. An alternative outcome could be an increase of the tangent angle to the membrane profile at the ND rim without the waist expansion. A detailed computational analysis avoiding the toroidal shape assumption is needed to resolve this issue.

"Regarding the torque at the nanodisc edge, the question raised by the reviewer, it can be shown that the downward bending is small, so that the flat annulus approximation at the outer nanodisc edge does not deviate by a large amount from the exact solution. The argument is as follows. (1) The angle made by an annular piece of membrane of width : at the nanodisc edge is ? ~ @:!/A, where @ is the downward force per unit length at the nanodisc edge and A is the membrane bending modulus. (2) The net downward force is the same at any cross section of the pore. At the pore waist net downward force is ~ 2C(D + F/2)H, assuming the mean curvature of the waist *+,- is negligible. Here, D , F and H refer to the radius of the pore, the membrane thickness, and *+,- tension respectively. Thus, for pores of radii similar to the membrane thickness, the force per unit length at the ND edge is @ ~ FH/J./ where J./ is the radius of the ND. (3) This force per unit length gives an angle ? ~ FH:!/AJ./. Using : ~ 5 ->, and other values as in Table S1, we get an angle ? ~ 4. Thus, the bending of the membrane is negligible near the nanodisc"

This reasoning is very hard to follow. (i) For some reason, the expressions include the membrane thickness, while the latter is already accounted for by the bending rigidity, which is also a part of the same expressions. (ii) Further, the authors estimate the angle of 4o at the ND edge, while the question is about the curvature (mean curvature) there rather than the angle. The expression for the curvature includes the angle derivative along the contour length, not just the angle itself. (iii) Moreover, the origin of the constant downforce, used by the authors for their argumentation, is unclear. Indeed, the cell membrane is, practically, horizontally oriented so that the downforce at the bottom of the pore should vanish. (iv) The suggestion of an almost vanishing mean curvature in the pore waist is incompatible with the assumption of a considerable lateral tension.

"The protein scaffold around the nanodisc can be formed by various configurations of the apolipoprotein E variant containing the N-terminal 22 kDa fragment (ApoE422K) [38]. In all cases, it is thought that two rows of stacked α-helical segments form a belt at the boundary of the nanodisc. The α-helical segments interact laterally and are connected by short linkers [38]. The torque contribution could arise either from the twisting rigidity of the α-helices or from the interactions between the proteins and the lipid molecules.

As the reviewer says, we assumed constant torque. We found a potential quadratic in the twisting angle could not reproduce the energetics of large pores of radii ~ 3 – 6 nm when ~15 SNARE complexes are present at the fusion pore without Synaptotagmin-1 [3].

In fact, we did consider the effect of the torque for pores of smaller radii. The proteins are twisted only when the radius or the height of the pore are large enough so that the rim of the toroidal pore where it joins with the flat membranes reaches the boundary of the nanodisc. Such radii or heights are larger than the equilibrium values predicted by the model, so it costs energy to access them. Thus, such states do not contribute much to the statistical average at the equilibrium values of radius and height."

A proposal about a specific mechanism of a constant (independent of deformation) torque generation should enable quantitative estimation of this factor. Otherwise, this energy contribution appears too speculative to be published.

Experimental part

The reviewers were disappointed to learn that the authors chose not to do an additional experiment, in particular, of determining the binding kinetics of nanodiscs to the cell membranes. The reviewers believe this experiment would yield an important piece of data showing what constitutes the rate-limiting step in the fusion pore assay the authors have developed. Moreover, the technical difficulties stated in the rebuttal, mainly high fluorescence backgrounds, may be bypassed using techniques such as FRET.

However, the senior author professed difficult situations in his lab midst the current pandemic situation. Therefore, at least regarding Comment 1, the authors need to state these potential issues in their experimental assay-including low rates of fusion pore opening, low cooperativity and unawareness of the rate-limiting step-and discuss the limitations and reservations in interpreting their data sets. It may then be possible to proceed as the authors have suggested (on page 7 of the rebuttal).

*Reviewer #1:*

The article by Z. Wu et al. addresses experimentally and theoretically the mechanism by which Syt-1 contributes to fusion pore dilation. The system used consists of nanodiscs fused with SNARE-containing cell membranes. The new proposal of the work is that the fusion pore expansion is driven by an intra-membrane reorientation of Syt1 hydrophobic loops, which leads to rotation of SNARE complexes and the resulting increase of the inter-membrane distance. The latter leads to an increase of the fusion pore radius.

In my view the article can not be published in the present form.

The suggested mechanism of the fusion pore dilation is crucially based on the theoretical model which, in its present form, raises questions.

1. The model assumes that the whole membrane is exposed to a constant lateral tension. I am wondering whether any relevant level of tension can exist in this system given that the edge of the nano-disc is free, i.e. is not subjected to any external force. It is true that the apolipoprotein scaffold at the disc edge could sustain some tension due to the scaffold compression and bending/twisting rigidities provided that the latter are sufficiently large.

Estimations of sustainable tensions based on the feasible values of the protein scaffold rigidities are necessary to support the model.

2. It is assumed that for a substantial range of the pore radii the pore can be described by a toroidal shape. This implies that the fragment of the nano-disc membrane between the ND edge and the rim of the pore is flat. This does not seem to be feasible mechanically since it would violate the torque equilibrium at the disc edge. Indeed, the tension in the cell membrane generates a rotational moment with respect to the ND edge (the pore height serving as a lever), which must be counteracted by torque at the edge. On the other hand, the existence of the edge torque necessarily means generation of the membrane curvature next to the disc edge.

Consideration and estimation of this torque and the related shape of the ND membrane, including their dependence on the pore radius are absolutely necessary to justify the model and its major prediction on the increase of the pore radius upon increasing of the pore height. Also, in case the scaffold is too soft to develop large enough torques to equilibrate the rotational moment of the membrane tension, the whole description becomes questionable.

3. Considering the partially toroidal shapes of the pore, the authors assume the torque at ND edge to be constant since the related energy contribution is linear in the angle. I am wondering about the physical origin of such a constant torque and why its effect was not considered for smaller pore radii.

*Reviewer #2:*

The authors have done a remarkable job of creating an assay for the fusion pore that does not require capacitance measurements or amperometry, but detects a transient in the DC current through a cell membrane , interpreted as the fusion pore, when a nanodisc is applied to the cell surface.

I do not understand why the conductance is limited in time, and what the meaning of the time course is. If the conductance rises and then falls, does it mean that competing factors are changing in time? In that case, what does the mean pore conductance mean for this complex time course?

While this system shows an effect of Syt1, it only explains a very small part of the biological effect of Syt1. The effect of calcium on synaptic exocytosis follows a fourth-power continuously increasing curve with presynaptic [Ca^2+^]free, this is a large effect by 20 uM, and not in keeping with the rather modest (3 fold) increase in activity in the author's model with [Ca^2+^]free in an S-shaped curve that saturates rapidly. An explanation for the saturation of the Ca effect is needed.

The section, "Calcium-dependent membrane-insertion of Syt1 C2AB, but not curvature generation, is necessary for pore dilation" did not seem particularly convincing. I did not follow this explanation: "Membrane insertion of these hydrophobic residues expands the membrane in one leaflet, creating an area mismatch, which relaxes as the membrane curves away from the wedge-like insertion. This membrane buckling is thought to contribute to the triggering of release (37, 52, 53). Nor was I convinced that "Thus, membrane penetration is required for pore expansion by Syt1, but curvature generation is not." Or that " It is well known that calcium induced loop insertion causes a reorientation of the C2 domains (86, 87)."

*Reviewer #3:*

This manuscript by Wu et al. reports on use of in vitro assays to study opening of fusion pores catalyzed by SNAREs and syanptotagmin1 (Syt1). One of the persisting questions in Ca^2+^-triggered exocytosis is how these machineries catalyze fusion pore opening, which is allegedly energetically-expensive, on millisecond scales as in the case of central nerve systems.

Recent incorporation of the electrophysiology tools is a welcoming addition to the field because it makes characterization of the fusion pores more quantitative, to the extent that would be likely unattainable with other methods. The authors formed nanodiscs reconstituted with 8 copies of R-SNAREs and Syt1 each (~25 nm diameter), and expressed Q-SNARE proteins in a flipped orientation on HeLa cell membranes. Fusion events between the nanodiscs and the cell membranes led to conductance of ions through fusion pores, which was monitored via the electrophysiology tool. The authors carried out detailed and quantitative analysis on the various physical properties of fusion pores including the average conductance and fluctuation rates. The authors took one step further to introduce various mutations to the C2B domain and studied their effects on the fusion pore properties.

These observations convincingly showed that in the in vitro assay the authors developed, SNAREs and Syt1 work in a concerted manner to open fusion pores with larger diameters and increased fluctuations with Ca^2+^ and PI(4,5)P2 working as important cofactors. The authors propose a model where enhanced membrane penetration of the C2B domains in the presence of these cofactors tilts up the SNARE complexes. This increases height of the pore structure, which in turn leads to dilation of fusion pores.

Although these results provide some interesting insights into the molecular mechanisms underlying fusion pore opening, there are issues that need to be addressed before recommendation of this manuscript for publication in *eLife*.

Comment 1. One concern is the rate of fusion pore opening is still low, a few times within 10 minutes, which is orders of magnitude slower than what is observed and Ca^2+^-evoked exocytosis in physiological milieu. Although it would be too demanding to expect reconstitution of ms-scale Ca^2+^-triggered pore opening in the current work, it is still important to identify the rate-limiting step for fusion pore opening in the in vitro assay the authors have developed.

The authors may want to study of binding kinetics of the nanodiscs to cell membranes under the reaction conditions used in this study probably using single-particle fluorescence microscopy or capacity measurement. If the binding latency is only a small fraction of the observed long latency, the priming step after binding could define the rate-limiting step. This priming may involve many molecular re-arrangement between two fusing membranes, such as formation of SNARE complexes (in trans or cis forms) and inter-SNARE complex arrangement (shown in Figure 6E) and recently reported interactions between SNARE complexes and the C2B domains (Ref. 76, 77). To sum up, the authors need to characterize the binding kinetics of nanodiscs and cell membranes and specify the rate-limiting step of the current in vitro fusion pore assay in the manuscript.

In this vein, it is also important to see how the pore open frequency is modulated as a function of Ca^2+^ concentration, which is currently missing in Figures 4 and S5. This is because the binding between fusing membranes is known to be substantially accelerated in the presence of Ca^2+^.

Comment 2. In the model shown in Figure 6, the authors suggest that enhanced binding and penetration of the C2B domain mechanically tilt up the SNARE complexes, thereby catalyzing larger fusion pores. The authors further suggest that this levering action can withstand restoring force of the fusion pore up to 14-16 pN.

This estimation, however, appears to be at odds with the observations made in previous single-molecule force spectroscopy studies. For example, the single neuronal SNARE complex is stabilized by a huge free energy of 65 kBT, but even the SNARE complexes are fully unzipped upon application of 16 pN tension (Ref. 92, 93). In addition, binding between the negatively charged membranes (with 5 mol% PI(4,5)P2) and C2B already shows repetitive binding and unbinding under 3 to 4 pN tension (Ref 62). Thus, although the model proposed by the authors is interesting, it may be more realistic to expect that membrane penetration of multiple C2B domains has a moderate steering effects for the SNARE complexes lining the fusion pores, rather than working as strong mechanical supports maintaining the suggested elongated pore structure.

[Editors’ note: further revisions were suggested prior to acceptance, as described below.]

Thank you for submitting your article "The neuronal calcium sensor Synaptotagmin-1 and SNARE proteins cooperate to dilate fusion pores" for consideration by *eLife*. Your article has been reviewed by 3 peer reviewers, including Felix Campelo as the Reviewing Editor and Reviewer #1, and the evaluation has been overseen by Vivek Malhotra as the Senior Editor. The following individual involved in review of your submission has agreed to reveal their identity: Patricia Bassereau (Reviewer #2).

Essential Revisions:

We request no additional experiments. We are only asking for some revisions that are mostly aimed at improving the clarity the manuscript as well as some extra computations regarding the model's assumptions. These additional computations should be relatively straight-forward for the authors. For these required revisions, see the detailed reports from Reviewer #1 and Reviewer #3. In particular, we'd like to emphasize on the following:

1. About the BCs used for the computations (see e.g. Eq. (8) of the SI Appendix): the authors should study (or provide a solid argumentation on why this might not be necessary) the shape and energies when the tCell membrane does not relax to a perfectly flat state but to a catenoidal shape.

2. The final section of Results presents some modeling work with C2A. There are no relevant experiments and the results are not discussed in the Discussion section. It is not clear how this section fits into this paper, so unless the authors can adapt it to provide a logical explanation in the context of this paper, this section should probably be removed.

Reviewer 1:

The calcium sensor Synaptotagmin-1 (Syt1) is known to be a key element for neuronal SNARE-mediated fusion, but how fusion proceeds is still debated and many different scenarios and physical models have been proposed. In this work, Z. Wu and collaborators use an interesting assay for measuring single fusion event with a very good time resolution (ms): v-SNARE proteins are reconstituted in large nanodiscs and fuse with "flipped cells" expressing t-SNAREs facing the extracellular media. Syt1 interacts either as a full length protein co-reconstituted in the nanodisc or as a truncated soluble version (C2A+C2B domains, or mutants) added in the patch pipette. Single fusion events are followed by electrophysiology using voltage-clamp in the cell-attached mode, as a function of PIP2 in the cell membrane or calcium in the medium. The authors focus mostly on the effect of Syt1 on pore expansion. In parallel, they have developed a mathematical model for fusion of nanodisc and cells based on Helfrich energy for the membrane fusion part and theory of elasticity for the scaffolding by ApoE around the nanodisc. The experiments clearly show that Syt1 promotes the growth of the fusion pore in a way dependent on binding to PiP2 and calcium concentration. Its interaction with SNAREs also contributes to pore expansion. The C2B+C2A domain (C2AB) is sufficient to induce this effect and Ca^2+^-dependent penetration of the hydrophobic loop is essential. These dependences and the pore growth rates are nicely reproduced by a model where at low calcium, the C2B domain binds to t-SNARES and to PiP2, remaining parallel to the membrane and keeping the pore small or closed. When calcium increases, the membrane insertion of the hydrophobic loop while being connected to SNAREs produces a tilt of about 15{degree sign} of the SNARE-Syt1 complex. SNARE-Syt1 acts as a mechanical lever on the pore edge, which changes the pore shape and tends to open it more. The C2A domain's loop strongly boosts the kinetics of the opening process. Altogether, the lever model accounts well for the experiments presented in the study, in contrast with different models previously published.

With this work, we can expect conceptual advances on the contribution of Syt1 to neuronal membrane fusion. In addition, the modelling of the mechanical deformation of a nanodisc and of its scaffold in the presence of a pore should be more generally relevant to experiments involving mechanosensitive membrane proteins or other pore forming structures.

I was not a reviewer in the initial version of the manuscript, but I have now read carefully both the current revised manuscript and the rebuttal letter including the detailed responses to the reviewers of the previous submission.

In general, I think that this manuscript deals with an interesting topic (fusion pore expansion by Syt and SNARES) using a variety of tools (experimental study of pore size and dynamics using nanodisks, as well as detailed modeling of the pore shape).

Regarding the experimental part, and following the discussion from the previous submission, although I agree that the ND binding kinetic measurements would have been an important piece to be added, I accept the statements made by the authors in this revised manuscript.

Regarding the modeling part, I do appreciate the effort made by the authors in implementing and solving the shape of the pore without the toroidal pore assumption. The finding that the tension required to open up the pore is different in the exact solution as compared to the toroidal pore is interesting and important. It also helps the authors discuss why they think that the scaffold is not buckling as a response to tension (and bending) of the ND membrane.

That being said, I have some specific comments about the model, and about some of the assumptions made there:

1. Perhaps to me the most important one is the BCs shown in Eq. (8) of the SI Appendix. There the authors assume that the membrane (on the tCell side) is flat (phi(L)=0) at a certain distance (R_inf_=30nm) from the pore axis. A couple of thoughts.

– R_inf_ is chosen somehow arbitrarily.

– Why does the membrane need to relax to a flat state far away from the pore and not a catenoidal geometry (which also has zero bending energy)? This latter condition could be implemented by assuming phi(L)≠0. To me, intuitively (so maths might proof me wrong of course), such a geometry could make that zipping does not lead to pore expansion but maybe to a change in the catenoidal angle? I think that the authors should discuss if this makes sense or not.

2. I've noticed some inconsistencies in the "Mathematical Model of the ApoE scaffold" in the SI Appendix. In particular:

– some references are missing in the appendix.

– Eq. (14) in the app. please use U_scaffold_ and not F.

– Two AHs per ApoE scaffold are assumed to estimate K_soft_ and K_hard_, is that correct? This should be explained and reasoned in the text as the readers are left to deduce that from the numbers.

– pg. 11 appendix: with the definition given in Eq. (14) of k=(K_hard_-K_soft_)/(2 R_NLP_^2^), and using the values in the text just above that, I get k=2.1 kBT/nm and not 4.2 kBT/nm as the authors wrote. Please verify. And then the evaluation of the twist angle for a 2 pN torque, in my hands, appears to be phi=7.5 degrees and not 30 degrees. If this is correct, how does it alter the results of the model?

3. What's the relative importance of the hydration energy to the overall free energy of the pore? I'm asking because the pore is assumed toroidal for Eq. (16) in the SI appendix, but the shape is not toroidal anymore. Could the authors elaborate on that?

4. Figure 1B: Is it fair to compare IF signal of PIP2 in permeabilized cells to the signal in non-permeabilized cells?

5. Figure 2A: Did the authors considered a multiple comparison test when inferring the statistical significance, or just performed one-to-one Student's t-test between the different conditions? If the answer is the latter, then caution should be taken in interpreting the observed differences (see e.g. PMID: 31596231)

*Reviewer #2:*

The calcium sensor Synaptotagmin-1 (Syt1) is known to be a key element for neuronal SNARE-mediated fusion, but how it proceeds is still debated and many different scenarios and physical models have been proposed. In this work, Z. Wu and collaborators use an interesting assay for measuring single fusion events with a very good time resolution (ms): v-SNARE proteins are reconstituted in large nanodiscs and fuse with "flipped cells" expressing t-SNAREs facing the extracellular media. Syt1 interacts either as a full length protein co-reconstituted in the nanodisc or as a truncated soluble version (C2A+C2B domains, or mutants) added in the patch pipette. Single fusion events are followed by electrophysiology using voltage-clamp in the cell-attached mode, as a function of PIP2 in the cell membrane or calcium in the medium. The authors focus mostly on the effect of Syt1 on pore expansion. In parallel, they have developed a mathematical model for fusion of nanodisc and cells based on Helfrich energy for the membrane fusion part and theory of elasticity for the scaffolding by ApoE around the nanodisc. The experiments clearly show that Syt1 promotes the growth of the fusion pore in a way dependent on binding to PiP2 and calcium concentration. Its interaction with SNAREs also contributes to pore expansion. The C2B+C2A domain (C2AB) is sufficient to induce this effect and Ca^2+^-dependent penetration of the hydrophobic loop is essential. These dependences and the pore growth rates are nicely reproduced by a model where at low calcium, the C2B domain binds to t-SNARES and to PiP2, remaining parallel to the membrane and keeping the pore small or closed. When calcium increases, the membrane insertion of the hydrophobic loop while being connected to SNAREs produces a tilt of about 15{degree sign} of the SNARE-Syt1 complex. SNARE-Syt1 acts as a mechanical lever on the pore edge, which changes the pore shape and tends to open it more. The C2A domain's loop strongly boosts the kinetics of the opening process. Altogether, the lever model accounts well for the experiments presented in the study, in contrast with different models previously published.

With this work, we can expect conceptual advances on the contribution of Syt1 to neuronal membrane fusion. But in addition, the modeling of the mechanical deformation of a nanodisc and of its scaffold in the presence of a pore should be more generally relevant to experiments involving mechanosensitive membrane proteins or other pore forming structures.

This manuscript is an extensive revision of a paper that was previously submitted to *eLife*. The main critics on the previous version were on the mechanical models behind the interpretation of the data, in particular on the mechanical resistance of the ND assembly to pore opening, the toroidal pore model and the torque exerted by the protein scaffold on the nanodisc periphery. It is true that these models are essential for providing a mechanism. As far as I can tell and not being a theoretician myself, I think the authors did a good job at addressing the different issues that were raised by the former editor and reviewers. These new additions reinforce their conclusions and their interpretation of their data. On the experimental side, considering the requests after the previous review and the still difficult situation related to the COVID pandemic, I consider that the new version includes the necessary discussions and precautions, in particular on the limitation of the assay to assess the binding kinetics. Personally, I think that the paper is ready for publication.

*Reviewer #3:*

This study by Wu et al. presents interesting new results obtained with a very sophisticated experimental system and incorporating very sophisticated modeling into their interpretations. The principal results advance the idea that synaptotagmin promotes fusion pore expansion through its interactions with membranes. The idea of a role for synaptotagmin in pore expansion has been around for quite some time but the present results extend what we know about this process. In particular, Syt1 binding to the lipid bilayer to exert force and change fusion pore shape is novel and interesting. All in all, it is a strong paper but a number of concerns require attention.

1. When the authors state ~4 copies of syt1 and VAMP2 per disc face (page 4 and 5), this is an average and the actual number will have a wide range due to inherent fluctuations when numbers are small. This variation in copy number must be incorporated into the discussion.

2. How do the authors arrive at 10 nm maximum diameter (top of P 5)? There must be a limit in a 25 nm NLP but the actual value is hard to specify, and the 10 nm value seems like a guess.

3. The authors use the Boltzmann distribution to go from the observed pore size distribution to the size dependence of pore energy. This is an impressive leap of insight and creativity. But the Boltzmann distribution applies to systems at equilibrium and the observed conductance time course is probably very far from equilibrium. The underlying process appears to be irreversible. Each pore opening episode looks like a trajectory through a complex energy landscape. Some acknowledgement of these shortcomings must be made.

4. On P 9 the authors discuss changes in pore radius, height, and shape. Doesn't that complicate the relation between diameter and conductance?

5. The last sentence of the Mathematical modeling section on P 10 states that increasing pore height increases membrane bending energy. Ref 109 shows that increasing height reduces bending energy.

6. The final section of Results presents some modeling work with C2A. There are no relevant experiments and the results are not discussed in the Discussion section. I do not see how this section fits into this paper.

---

## [Author Response]

[Editors’ note: the authors resubmitted a revised version of the paper for consideration. What follows is the authors’ response to the first round of review.]

Computational part"In cell-attached recordings, the patch of membrane that is under study is under high tension due to adhesion to the walls of the pipette. Strong adhesion of the membrane to the walls of the glass pipette is needed to obtain a high-resistance seal (the so-called "Gigaseal") for low-noise recordings. Previous measurements show the membrane tension in the patch is ~1 pN/nm [37]. This is consistent with our model that obtained a tension of 0.66 pN nm-1 as a best-fit parameter by comparing the model-predicted slope of the free energy of pore dilation for pores of size ~ 1 nm between the model prediction and the experimental measurement when only SNAREs were present in a previous study of ours [3].Forces that oppose membrane tension at the nanodisc boundary arise presumably from the bending rigidity of the ApoE proteins, and from their interactions with the lipids. A radially inward force per unit length equal to the membrane tension acts on the ApoE scaffold. Assuming only the bending rigidity opposes this inward force, we can compare the membrane tension to the threshold force per unit length where a circular ring buckles, kT Lp /d2 , where : and d and Lp are the radius of the cross-section of the circular ring and its persistence length respectively. As the scaffold proteins consist of a series of α-helical regions interspersed with unstructured regions [38], and a typical persistence length of α-helices is ~ 100 nm [39], the buckling threshold is ~400 pN nm-1 assuming a thickness : d~ 1 nm, much larger than the membrane tension of 0.66 pN nm-1"The should be a misunderstanding here. According to the classical literature on the buckling instability of a ring, the radial force per unit length of the ring (the membrane tension in the present case) leading to the instability is, approximately, γ~(kT L_p_)/r^3^ , where r is the ring radius. Taking Lp = 100nm, as suggested by the authors, and r~10"nm" , according to the dimension of nanodiscs used in the work, one gets, practically, a vanishing critical tension. This must mean that, as a result of the pore nucleation, the tension drops to zero.This means that the model is inconsistent with the reality in this essential point.

We apologize for our incorrect and extremely careless response, and we thank the editor for pointing out the error. Our “estimate” of the critical buckling force per unit length for the ND scaffold tension did not even have the correct dimensions. Below, we have done the job carefully. The conclusion is that the ND scaffold is easily strong enough to withstand the forces from the membrane.

In fact, the critical buckling force is not small. As the editor stated, calculation of the stability of an elastic ring gives a buckling threshold 3kTLp/RND3 (we include the prefactor; see e.g. S. P. Timoshenko, J. M. Gere, *Theory of elastic stability* (Courier Corporation, 2009)). Noting that the ND scaffold of radius R_ND_ ~ 12 nm in fact comprises two α helices, the estimated persistence length is L_p_ ~ 200nm, giving a substantial critical force per unit length ~1.4 pN/nm. This value is well above the estimated tension ~ 0.1 pN/nm. (Previously we estimated ~ 0.7 pN/nm using the toroidal pore; with our exact calculations of pore shapes – see below – we find a smaller best fit tension).

However, the forces acting on the ND boundary are due not only to tension, but also to bending. The force on the scaffold is determined by the in-plane membrane stress (R. Capovilla et al., *J. Phys. A,* 2002)σa=[2kC (Kab−CgAB)−γgAB]eb−2K∇aCn,whose projection onto the outward pointing membrane tangent vector at the ND boundary 𝝂 = 𝝂_𝒂_𝒆^𝒂^ and the radial unit vector r^ gives the radial force per unit length acting on the scaffoldfr=r^•σava=[2K(C⊥2−C∥2)−γ]cosα+2K∂C∂ssinαwhere α is the angle the membrane makes with the “horizontal” direction at the ND boundary, i.e. the angle between the outward pointing membrane tangent vector 𝝂 and the radial direction. Here C_-_, C_∥_ are the membrane curvatures perpendicular and parallel to the ND boundary, 𝒔 is arclength along the meridian of the membrane (whose rotation about the symmetry axis generates the membrane surface), 𝜿 is the bending stiffness, 𝑪 and *K*^𝒂𝒃^ denote the mean curvature and curvature tensor, 𝒈^𝒂𝒃^ the metric tensor, 𝒆_𝒃_ the membrane surface tangent vectors, 𝛻^𝒂^ the covariant derivative, and 𝒏 the membrane unit normal vector (𝒂, 𝒃 = 𝟏, 𝟐 label directions in the membrane tangent plane, repeated indices summed).

One can get a sense of the magnitude of the force per unit length 𝒇_𝒓_ by considering the simple case 𝜶 = 𝟎 (neglecting rotation of the ND boundary). Then 𝑪_∥_ = 𝐬𝐢𝐧 𝜶/𝑹_𝑵𝑫_ vanishes, sofr=2kC2−γ

From numerically determined fusion pore shapes (see below) we find a perpendicular radius of curvature r⊥≡C⊥−1≈30nm, weakly dependent on tension in the range 𝟎 < 𝜸 < 𝟏 𝐩𝐍 𝐧𝐦^-𝟏^. Using 𝜿 = 𝟐𝟎 𝐤𝐓, this gives an expansive bending contribution ~ 0.18 pN, and a net force per unit length 𝒇_𝒓_~𝟎. 𝟏 pN/nm in the outward direction (i.e. a net expansive effect, not contractile). Thus there is no buckling threat to the ND boundary.

These conclusions are qualitatively unchanged when one allows 𝜶 to vary due to rotation of the ND boundary. Solving the membrane shape equation numerically (without toroidal or other variational approximations, see below) and imposing a torque-balance boundary condition at the ND boundary, we find the net inward force per unit length is even smaller in magnitude, 𝒇_𝒓_~ −0.01 pN/nm, valid for a range of tensions 𝟎. 𝟏 < 𝜸 < 𝟏 𝐩𝐍/𝐧𝐦. This weak contractile effect is well below the buckling threshold.

"We agree that the toroidal shape is not an exact fusion pore solution, but we use this shape for simplicity, and we believe it captures all the important features of a more exact solution. It has the advantage that the biophysical aspects are clearly articulated. For this reason, toroidal pores have been assumed in many previous theoretical studies (e.g. [40-42])."This is not a convincing argument. In the previous studies the toroidal pore approximation was used for the cases of two infinitely large membranes connected by a fusion pore. In the present study, the dimension of the nanodisk is of the same order of magnitude as the pore radius so that the deviation from the toroidal shape may have essential consequences for the results.One of the consequences of the toroidal pore assumption is the conclusion that SNARE reorientation results in the expansion of the pore waist, which is one of the central conclusions of the model. An alternative outcome could be an increase of the tangent angle to the membrane profile at the ND rim without the waist expansion. A detailed computational analysis avoiding the toroidal shape assumption is needed to resolve this issue.

In response to the editor’s suggestion, we performed detailed computational analysis without assuming a toroidal pore (TP). Overall, the exact fusion pore shapes are similar to those predicted by the TP model, and the exact calculations (within the Helfrich framework) show that the pore radius increases when the pore height is increased, as in the TP model (this qualitative effect was central to our model of Syt-mediated pore expansion). The pore free energy versus pore radius is well approximated by the TP model. The biggest failure of the TP model is that the predicted membrane tension, based on fitting the TP model predictions to experiment, is too large. The tension from fitting experiment using the exact solution is lower, because the “exact” pore is more difficult to open up against applied tension than is the TP. Other than this (important) difference, our new numerical calculations have not qualitatively altered our conclusions.

Previous works using the TP approximation typically use infinite planar membranes to obtain solutions, but these solutions are then interpreted in terms of situations with far from infinite membranes. For example, Chizmadzhev et al. and Nanavati et al. (cited in our response above) interpret their results in terms of exocytosis, where vesicles with radii as small as 20 nm (synaptic vesicles) are involved (Chizmadzhev et al. *Biophys. J.*, 2000; Nanavati et al., *Biophys. J.*, 1992). This is clearly inconsistent with the infinite planar membrane boundary condition. Chizmadzhev et al. and Nanavati et al. use the TP to model pores with radii of up to ~30 nm and ~100 nm, respectively, comparable to the size of even large secretory granules. Overall, TP models are widely used to analyze fusion pores between finite sized membranes whose size is of order the fusion pore itself. Given this, it is of interest to note that the TP approximation performs reasonably well.

"Regarding the torque at the nanodisc edge, the question raised by the reviewer, it can be shown that the downward bending is small, so that the flat annulus approximation at the outer nanodisc edge does not deviate by a large amount from the exact solution. The argument is as follows. (1) The angle made by an annular piece of membrane of width : at the nanodisc edge is ? ~ @:!/A, where @ is the downward force per unit length at the nanodisc edge and A is the membrane bending modulus. (2) The net downward force is the same at any cross section of the pore. At the pore waist net downward force is ~ 2C(D + F/2)H, assuming the mean curvature of the waist *+,- is negligible. Here, D , F and H refer to the radius of the pore, the membrane thickness, and *+,- tension respectively. Thus, for pores of radii similar to the membrane thickness, the force per unit length at the ND edge is @ ~ FH/J./ where J./ is the radius of the ND. (3) This force per unit length gives an angle ? ~ FH:!/AJ./. Using : ~ 5 ->, and other values as in Table S1, we get an angle ? ~ 4. Thus, the bending of the membrane is negligible near the nanodisc"This reasoning is very hard to follow. (i) For some reason, the expressions include the membrane thickness, while the latter is already accounted for by the bending rigidity, which is also a part of the same expressions. (ii) Further, the authors estimate the angle of 4o at the ND edge, while the question is about the curvature (mean curvature) there rather than the angle. The expression for the curvature includes the angle derivative along the contour length, not just the angle itself. (iii) Moreover, the origin of the constant downforce, used by the authors for their argumentation, is unclear. Indeed, the cell membrane is, practically, horizontally oriented so that the downforce at the bottom of the pore should vanish. (iv) The suggestion of an almost vanishing mean curvature in the pore waist is incompatible with the assumption of a considerable lateral tension.

Again, we apologize for our incorrect and confusing previous response, and again we thank the editor for pointing out the errors. We tried to argue that the true fusion pore curvature at the ND boundary is small, so the TP shape (with its assumed flat outer edge) is not dramatically wrong. Roughly we used the tension contribution to the membrane stress to estimate the downward force 𝒇_𝒛_ (per unit length) bending the membrane at the edge. In reality 𝒇_𝒛_ = 𝟎 as the ND is free, so the tension contribution is balanced by a bending contribution.

Here is the correct argument. Our calculations using the TP model predicted that, with 100 µM Ca, the pore minimizing the membrane free energy has a flat region that extends ~2 nm inward from the ND scaffold. We will argue that the exact solution to the membrane shape equation is not far from flat in this region as well. Using the expression above for the stress tensor, the vertical force per unit length acting on the circle 𝒔 = 𝐜𝐨𝐧𝐬𝐭. of radius 𝑹 lying in the membrane isfz=z^⋅σava=[2K(C⊥2−C∥2)−γ]sinα−2K∂C∂scos αwhere 𝑪_⊥_ = 𝝏𝜶/𝝏𝒔, 𝑪_∥_ = 𝐬𝐢𝐧 𝜶/𝑹 are the curvatures perpendicular and parallel to 𝒔 = 𝐜𝐨𝐧𝐬𝐭. For simplicity, assume 𝜶 = 𝟎 at the ND boundary (clamped boundary conditions), so 𝝏𝑪/𝝏𝒔 = 𝟎 at the boundary. Then for a location close to the ND boundary 𝜶 and 𝝏𝑪/𝝏𝒔 are small so, to leading order,fZ=[2K•(∂C∂s)2−γ]α−2K∂C∂s+𝓋(a2)=0.

At this point it is not possible to close the perturbative argument, since 𝝏𝑪/𝝏𝒔 is unknown. Thus we used our numerical solutions to calculate this derivative, and we find it contributes negligibly to the vertical force. Hence the square bracket term approximately vanishes, i.e. the curvature 𝝏𝜶/𝝏𝒔 ≈ √(𝜸/𝟐𝜿) ≈. 𝟎𝟐𝟓 𝐧𝐦^$𝟏^ after using 𝜸 = 𝟎. 𝟏 pN/nm.

This shows that the exact solution has a small curvature ~. 𝟎𝟐𝟓𝐧𝐦^-𝟏^ at the ND edge. For example, the membrane angle 𝜶 is only ~𝟑^𝟎^ a distance 𝟐 𝐧𝐦 inwards from the ND edge, corresponding to the width of the flat outer annulus of the TP. This is consistent with our numerical calculations of the membrane shape, which generated angles ~𝟑° − 𝟐𝟎° and curvatures ~𝟎. 𝟎𝟐 − 𝟎. 𝟏 𝐧𝐦-^𝟏^ within 2 nm of the ND edge. In conclusion, the exact solution of the membrane shape shows a small downward displacement of the membrane in the flat annulus region assumed by the TP model, showing that the TP model provides a reasonable approximation to the actual shape of the fusion pore in this region.

"The protein scaffold around the nanodisc can be formed by various configurations of the apolipoprotein E variant containing the N-terminal 22 kDa fragment (ApoE422K) [38]. In all cases, it is thought that two rows of stacked α-helical segments form a belt at the boundary of the nanodisc. The α-helical segments interact laterally and are connected by short linkers [38]. The torque contribution could arise either from the twisting rigidity of the α-helices or from the interactions between the proteins and the lipid molecules.As the reviewer says, we assumed constant torque. We found a potential quadratic in the twisting angle could not reproduce the energetics of large pores of radii ~ 3 – 6 nm when ~15 SNARE complexes are present at the fusion pore without Synaptotagmin-1 [3].In fact, we did consider the effect of the torque for pores of smaller radii. The proteins are twisted only when the radius or the height of the pore are large enough so that the rim of the toroidal pore where it joins with the flat membranes reaches the boundary of the nanodisc. Such radii or heights are larger than the equilibrium values predicted by the model, so it costs energy to access them. Thus, such states do not contribute much to the statistical average at the equilibrium values of radius and height."A proposal about a specific mechanism of a constant (independent of deformation) torque generation should enable quantitative estimation of this factor. Otherwise, this energy contribution appears too speculative to be published.

We have discarded the constant torque assumption. We developed a simple mechanical model of ND scaffold rotation. The model predicts a torque linear in the scaffold rotation angle for small deflections, and non-linear for large deflections. Since our solutions of the fusion pore shape equations predict small deflections at the ND edge (~𝟏𝟎°), the linear torque response is appropriate. We predict deflection angles somewhat smaller than the ~ 45° deflections reported in Martini simulations of SNARE-mediated ND-planar membrane fusion (Sharma and Lindau, 2019). We expect our deflection angles to be smaller, since the NDs in our experiments are larger than the NDs used in these Martini simulations.

Our model treats the scaffold as an elastic rod with rectangular cross-section, a simple representation of the two parallel α helices. The elastic bending energy of the rod isF=∫dL(Ksoft2Csoft2+Khard2Chard2)where 𝑲_𝐬𝐨𝐟𝐭_ and 𝑲_𝐡𝐚𝐫𝐝_ are the bending moduli of the rod in the material directions across its narrow and wide faces, respectively, 𝑪_𝐬𝐨𝐟𝐭_ and 𝑪_𝐡𝐚𝐫𝐝_ are the respective material curvatures, and 𝑳 measures the arclength around the scaffold. (The rod is inherently more difficult to bend across its wider face, i.e. is less soft in this direction.) The material curvatures represent the curvature of the scaffold projected onto the basis vectors of the material cross section. Now we treat the scaffold as if it were a homogeneous material with a Young’s modulus 𝑬 and a rectangular cross-section of width 𝒘 and height 𝒉, and we use a classical result from elasticity theory that 𝑲_𝐬𝐨𝐟𝐭_ = 𝑬𝒘^𝟑^𝒉/𝟏𝟐, 𝑲_𝐡𝐚𝐫𝐝_ = 𝑬𝒘𝒉^𝟑^/𝟏𝟐 (Landau and Lifshitz, *Theory of Elasticity*, 1959). Thus 𝑲_𝐬𝐨𝐟𝐭_ = 𝟐 𝑲_𝜶_ and 𝑲_𝐡𝐚𝐫𝐝_ = 𝟖 𝑲_𝜶_, where 𝒌_𝜶_ = 𝒌𝑻 𝑳_𝐩_ is the bending modulus of a single α helix of persistence length 𝑳_𝑷_ ≈ 𝟏𝟎𝟎 𝐧𝐦. Thus 𝒌_𝐬𝐨𝐟𝐭_ ≈ 𝟐𝟎𝟎 𝒌𝑻 𝐧𝐦 and 𝒌_𝐡𝐚𝐫𝐝_ ≈ 𝟖𝟎𝟎 𝒌𝑻 𝐧𝐦.

Suppose the cross section of the scaffold is rotated such that the long axis of the cross section makes an angle 𝜽 with the vertical direction, while maintaining the shape of the scaffold as a ring of radius 𝑹_𝑵𝑫_. Then the material curvatures areCsoft=cosθRNDChard=sinθRND

This gives an elastic bending energyF=2πRND(ksoft2cos2θRND2+Khard2sin2θRND2)

The torque per unit length to twist the scaffold through an angle 𝜽 is thusτ=12πRND∂F∂θ=Khard−Ksoft2RND2sin2θ≈Khard−KsoftRND2θ

Using the parameter values above, the model predicts that a torque per unit length ~2 pN twists the boundary scaffold of the ND an amount ~𝟑𝟎^𝟎^. We incorporated these effects in our calculation of the fusion pore shape by imposing a torque equilibrium condition at the ND edge, so that membrane torque is resisted by the ND scaffold. Numerical solutions of the membrane shape equation with this boundary condition predicted deflections ~𝟏𝟎° at the ND edge, and curvatures ~0.05 nm^-1^.

Summary of new model calculations and changes to text in the revised manuscript

Exact numerical solutions of the nanodisc-cell membrane fusion pore shape and energetics with and without Syt-SNARE complexes

In response to comments from the reviewers and editor, we abandoned the toroidal pore assumption and instead we solved the exact membrane shape equation for the ND-cell membrane fusion pore in the framework of the Helfrich model that accounts for membrane tension and bending energy. The model includes a torque equilibrium between the outer edge of the ND membrane and the ND scaffold, using a simple mathematical model of the elastic bending energy of the ND scaffold to predict the energy of scaffold rotation. This torque equilibrium is implemented as a boundary condition in our solutions of the shape equation. From numerical solutions, we determined the free energy of the pore versus pore radius and inferred the ND membrane tension by comparing these predictions to our previous experimental measurements of fusion pore conductances. We then added a contribution to the free energy representing the effect of SNARE complexes, dependent on the number of SNAREs present. (This parallels the procedure we used in our earlier toroidal pore-based calculation.)

We then repeated this procedure in the presence of Syt-SNARE complexes at the fusion site. These complexes imposed a constraint on the possible fusion pore shapes, since a Syt-SNARE complex must bind the membrane at a certain angle at both the C-terminal end of the SNARE complex, where SNARE TMDs insert into the bilayer, and the N-terminal end, where the Syt C2B domain Ca^2+^-binding loops tightly bind acidic phospholipids. In the presence of the Syt-SNARE complexes, smaller fusion pores had higher free energies. Thus, in the presence of Ca^2+^ when the Syt Ca^2+^-binding loops bind the membrane, pores of larger radius are energetically favored due to this geometric lever-like constraint. We find the predicted free energy versus pore radius profile is consistent with our experiments, both in the presence and absence of Ca^2+^. The model quantitatively reproduces the experimentally observed increase in conductance in the presence of Ca^2+^ with high accuracy.

Summary of changes to the manuscript related to the new mathematical modeling results

1. The changes to our model and our new results are detailed in the section “Mathematical modelling suggests that Syt1 and SNARE proteins cooperatively dilate fusion pores in a mechanical lever action.” Figure 6, referenced in that section, was updated with our new results. Figure 6B shows examples of our numerical solutions of the membrane shape equation with and without the constraint imposed by the Syt-SNARE complex. Figures 6C and 6D show quantitative data from the updated model. Figure 6E was updated to be consistent with our model of the ND scaffold that accounts for its finite twisting rigidity.

2. The SI subsections: “Mathematical model of the fusion pore with SNAREs and Synaptotagmin-1,” “Membrane free energy,” “Short-ranged steric hydration free energy”, “Free energy of SNAREs”, “Total free energy as a function of pore size, minimum membrane separation, total number of SNAREs”, and “Pushing forces from the membranes are insufficient to disrupt the SNARE-Syt primary-interface interaction” were updated to be consistent with our new model. New SI subsections were added entitled “Geometric constraints imposed by the Syt-SNARE complex,” “Mathematical Model of the ApoE scaffold,” and “Numerical method of solving the membrane shape equation.” Figure S8 was updated to include our fit of the membrane tension to previously obtained experimental data and to include predictions of the new model.

Experimental partThe reviewers were disappointed to learn that the authors chose not to do an additional experiment, in particular, of determining the binding kinetics of nanodiscs to the cell membranes. The reviewers believe this experiment would yield an important piece of data showing what constitutes the rate-limiting step in the fusion pore assay the authors have developed. Moreover, the technical difficulties stated in the rebuttal, mainly high fluorescence backgrounds, may be bypassed using techniques such as FRET.However, the senior author professed difficult situations in his lab midst the current pandemic situation. Therefore, at least regarding Comment 1, the authors need to state these potential issues in their experimental assay-including low rates of fusion pore opening, low cooperativity and unawareness of the rate-limiting step-and discuss the limitations and reservations in interpreting their data sets. It may then be possible to proceed as the authors have suggested (on page 7 of the rebuttal).

We thank the reviewers and the editor for accommodating pandemic-related issues, which are still ongoing. We now explicitly state throughout the manuscript the potential issues raised by the reviewers and the editor. Specifically:

a. We now explain the low rates of fusion pore opening in the first paragraph of the Results section (p. 4).

b. We have added two paragraphs to *Discussion* (pp.13-14) explaining the low cooperativity, the lack of knowledge about pre-fusion stages, and the need to be careful about interpretation of our results.

We have then implemented the revision plan suggested on p. 7 of the rebuttal. Namely:

c. To clarify what is meant by the rate of fusion pore opening (pores/min), we have edited the first paragraph in *Results* (p. 4), and the first paragraph in section *"Syt1 promotes fusion pore expansion"* (p. 5). We also moved SI Figure S8 to an earlier position (it is now SI Figure S2) and made changes accordingly to the text. This figure shows examples of current recordings, explains how pore currents are selected, how the fusion rate increases with nanodisc concentration in the pipette, and an alternative calculation of the fusion rate.

d. We have added a new panel (F) to Figure S6, showing how the fusion rate (pores/min) increases with calcium concentration. This panel is now referenced in section

"Calcium-dependence of pore dilation by Syt1 C2AB" on p. 7.

We made additional changes in response to issue #3 ("Some parts of the presentation need clarification"), as we had outlined in the revision plan:

e. We added some clarifications about pore lifetimes in the first paragraph of *Results* (p. 4) and referred to Karatekin, *FEBS Lett,* 2018 for a discussion of the relevance of nanodisc-based single pore assays to exocytotic fusion pores.

f. We edited the sections regarding curvature generation by Syt1, such that the relevant literature is cited, but not overemphasized.

In response to a minor comment, we moved the paragraph about the effects of the R398QR399Q mutation to *Discussion*.

[Editors’ note: what follows is the authors’ response to the second round of review.]

Essential Revisions:We request no additional experiments. We are only asking for some revisions that are mostly aimed at improving the clarity the manuscript as well as some extra computations regarding the model's assumptions. These additional computations should be relatively straight-forward for the authors. For these required revisions, see the detailed reports from Reviewer #1 and Reviewer #3. In particular, we'd like to emphasize on the following:1. About the BCs used for the computations (see e.g. Eq. (8) of the SI Appendix): the authors should study (or provide a solid argumentation on why this might not be necessary) the shape and energies when the tCell membrane does not relax to a perfectly flat state but to a catenoidal shape.

We performed additional calculations relaxing our assumptions about how the fusion pore membrane connects up to the tCell membrane. We found that relaxing these assumptions makes very little difference to the shape of the fusion pore. We added a figure to the appendix demonstrating this, see below.

2. The final section of Results presents some modeling work with C2A. There are no relevant experiments and the results are not discussed in the Discussion section. It is not clear how this section fits into this paper, so unless the authors can adapt it to provide a logical explanation in the context of this paper, this section should probably be removed.

We have removed this section from the paper.

Reviewer 1:[…]That being said, I have some specific comments about the model, and about some of the assumptions made there:1. Perhaps to me the most important one is the BCs shown in Eq. (8) of the SI Appendix. There the authors assume that the membrane (on the tCell side) is flat (phi(L)=0) at a certain distance (R_inf_=30nm) from the pore axis. A couple of thoughts.– R_inf_ is chosen somehow arbitrarily.– Why does the membrane need to relax to a flat state far away from the pore and not a catenoidal geometry (which also has zero bending energy)? This latter condition could be implemented by assuming phi(L)≠0. To me, intuitively (so maths might proof me wrong of course), such a geometry could make that zipping does not lead to pore expansion but maybe to a change in the catenoidal angle? I think that the authors should discuss if this makes sense or not.

We thank the reviewers for these constructive comments. Our choice of R_inf_ was actually not completely arbitrary, although no systematic analysis was performed to choose an optimal value. The only requirement we insisted on is that R_inf_ is sufficiently large that the boundary conditions imposed at the location (r=R_inf_) where the pore joins the tCell membrane have little to no effect on the pore shape. This is expected to be true provided R_inf_ > 𝑙_cap_ where lcap =k/2γ is the capillary length. Thus, we expect that provided R_inf_ is significantly greater than 20 nm, our model should reasonably describe the true pore shape. This observation is now mentioned in the appendix, in the section “Numerical method of solving the membrane shape equation”.

To verify this, we performed additional calculations of the fusion pore shape assuming a larger value of R_inf_ (50 nm). The fusion pore shapes were nearly identical to those with the original R_inf_ value in the region where their domains overlap. In particular, in the vicinity of the fusion pore, the two shapes were visually indistinguishable.

Our assumption that the membrane is flat where it meets the tCell membrane is motivated by the assumption that the large tCell membrane is itself flat (far enough away from the site of the fusion pore). To verify that this boundary condition did not influence the model results, we performed additional calculations, relaxing the assumption that the tCell membrane is asymptotically flat far away from the pore and instead implementing freely hinged boundary conditions where the pore meets the tCell membrane. These boundary conditions allow the membrane to choose the angle it makes at the point where it joins the tCell, equivalent to imposing zero torque at this location. This boundary condition produced very little change in the pore shape, as expected since the torque at this location was very small even with the previous zero slope boundary condition. Again, in the vicinity of the fusion pore the pore shape was indistinguishable from the shape obtained with zero slope boundary condition. This insensitivity to the boundary conditions used is now stated in the main text in the subsection “Mathematical modelling suggests that Syt1 and SNARE proteins cooperatively dilate fusion pores in a mechanical lever action” and in the appendix in the section “Numerical method of solving the membrane shape equation”.

We added a supplementary figure (Figure S8E) showing solutions of the fusion pore shape for each of these boundary conditions, so readers can easily and instantly compare them visually.

2. I've noticed some inconsistencies in the "Mathematical Model of the ApoE scaffold" in the SI Appendix. In particular:– some references are missing in the appendix.

Thank you, this has been fixed.

– Eq. (14) in the app. please use U_scaffold_ and not F.

Thank you, this has been fixed.

– pg. 11 appendix: with the definition given in Eq. (14) of k=(K_hard_-K_soft_)/(2 R_NLP_^2^), and using the values in the text just above that, I get k=2.1 kBT/nm and not 4.2 kBT/nm as the authors wrote. Please verify. And then the evaluation of the twist angle for a 2 pN torque, in my hands, appears to be phi=7.5 degrees and not 30 degrees. If this is correct, how does it alter the results of the model?

We thank the reviewer very much for alerting us to these arithmetic errors. We corrected these errors. Note that these errors were present only in the text of the SI appendix, and were not used in the calculations of the model, and therefore have no impact on our results.

3. What's the relative importance of the hydration energy to the overall free energy of the pore? I'm asking because the pore is assumed toroidal for Eq. (16) in the SI appendix, but the shape is not toroidal anymore. Could the authors elaborate on that?

The hydration energy only affects the energetics of the pore at very small pore radii, less than ~0.5 nm, due to the very rapid drop off of the hydration forces (exp (−𝑟/(0.1 nm))). Thus, this term has little effect on our results; it primarily serves to account for the repulsive self-interaction of the pore at very small radii. Unfortunately, this effect cannot be accounted for analytically without a closed form expression describing the shape of the fusion pore, and an assumption about the shape is therefore required at this step.

4. Figure 1B: Is it fair to compare IF signal of PIP2 in permeabilized cells to the signal in non-permeabilized cells?

Our goal was to compare the amount of exogenous, short-chain PI(4,5)P_2_ incorporated into the outer leaflet of the plasma membrane with the amount of endogenous PI(4,5)P_2_ found in the inner leaflet. To access the endogenous PI(4,5)P_2_ in the inner leaflet, permeabilization was required. These results are meant to show that the amount of exogenous PI(4,5)P_2_ incorporated into the outer leaflet is not completely off the mark which might put into question the relevance of the PI(4,5)P_2_ dependence we find. This type of comparison is the best we could do, and we believe it is adequate for the question we asked.

5. Figure 2A: Did the authors considered a multiple comparison test when inferring the statistical significance, or just performed one-to-one Student's t-test between the different conditions? If the answer is the latter, then caution should be taken in interpreting the observed differences (see e.g. PMID: 31596231)

We do not think a multiple comparison test is appropriate for the data presented in Figure 2A, because the results summarize two different biological questions with associated control experiments.

In order to determine which statistical test is most appropriate, McDonald

(http://www.biostathandbook.com/) recommends starting with a biological question, then formulating the question in the form of a biological null-hypothesis and an alternative hypothesis. The statistical null-hypothesis and the alternative hypothesis (which are about the data only) are formulated next, as they must follow from the biological hypotheses.

The first biological question we asked was whether the presence of Syt1 affected the rate of SNAREmediated membrane fusion. The biological null-hypothesis is that Syt1 does not affect the rate of SNARE-mediated membrane fusion. The statistical null hypothesis is that the mean fusion rate in the presence of Syt1 is the same as the rate in its absence. The other three conditions tested in the presence of Syt1 were control experiments. We know from the literature that Syt1 binds calcium and acidic lipids, and these interactions are needed for its function. Thus, if either of these co-factors are omitted, no effect of Syt1 is expected. Treatment with TeNT has in fact nothing to do with Syt1; TeNT cleaves the vSNARE Syb2/VAMP2 and is expected to reduce the fusion rate if fusion is indeed SNARE-dependent.

The second biological question we asked was whether the soluble C2AB domain of Syt1 alone behaves similarly to the full-length protein, *given* that Syt1 increased the rate of membrane fusion. The biological null-hypothesis is that C2AB does no increase the fusion rate. The statistical null hypothesis can be formulated as "the mean fusion rate in the presence of C2AB is not higher than in its absence". If the statistical null-hypothesis is rejected, then we would conclude that C2AB behaved similarly to Syt1. Alternatively, we could formulate the statistical null hypothesis as "the fusion rate in the presence of Syt1 or C2AB are the same" (which is likely not to be rejected given the scatter in the data, leading to the conclusion that the C2AB domain behaves like Syt1). The other three conditions tested in the presence of C2AB were controls as above.

We could of course ask: "does Syt1 or its soluble domain promote SNARE-mediated membrane fusion under any condition shown in Figure 2A?", which would require an adjustment for multiple comparisons, but this question would be too general. The biological null-hypothesis would then be: "Syt1 or its C2AB domain do not affect membrane fusion under any conditions tested", and the statistical null-hypothesis would be: "the mean fusion rate is the same under any condition tested".

Multiple comparison tests were developed because if one performs pair-wise comparisons over multiple conditions, the probability of falsely rejecting the null-hypothesis increases with the number of conditions. Multiple comparison tests add a penalty to avoid this type of error. However, if one includes in the multiple comparisons conditions that are not directly related to the biological hypothesis, then the probability of finding an actual effect decreases, because there is an unnecessary penalty paid for the comparisons that should not be included in the first place.

Of course, the biological question and the null-hypothesis can be formulated *after* performing the experiments in such a way that only some conditions appear relevant and are included in multiple comparisons. In our case, the TeNT experiments should clearly be excluded from multiple comparisons (because they do not directly test the role of Syt1 in fusion), but one might argue that Syt1 might affect fusion without its co-factors and that the conditions omitting calcium or PI(4,5)P_2_ should be considered nominal variables that should be included in a multiple comparison test.

A one-way ANOVA between the first four conditions in Figure 2A resulted in a p-value of 0.0064, suggesting the group means are different. A comparison between the control (SNARE-alone) and Syt1+calcium+PI(4,5)P_2_ adjusted for multiple comparisons using Tukey’s Honestly Significant Difference Procedure yielded a p-value of 0.0128. Thus, the mean fusion rate in the presence of Syt1 and its cofactors is higher than the SNARE-alone case (at the %5 level of significance). (The conditions omitting either co-factor are not significantly different than the control.) Repeating this procedure with C2AB, we find a p-value of 0.1231 for one-way ANOVA and 0.1661 when comparing the control to the condition with C2AB+calcium+PI(4,5)P_2_ (using the Tukey procedure as above). That is, at face value, the multiple comparison test would change the earlier conclusion that the fusion rate with C2AB in the presence of calcium and PI(4,5)P_2_ is significantly larger than the rate observed with the control group.

However, this raises an issue regarding the use of multiple comparisons, and more broadly of null hypothesis significance testing. The same SNARE-alone (i.e. "no C2AB") control group and the "C2AB" (with calcium and PIP_2_) group are re-plotted in Figure 3B and Figure 5B for comparison with other conditions. These different figures address different biological questions. Because the number of groups are different in each case, the p-value for comparing the control group against the C2AB group changes depending on how we present the data, which is somewhat arbitrary.

Due to its shortcomings, alternatives to null-hypothesis significance testing are being developed. We explored the use of the estimation graphic introduced by Ho et al.^1^ to visualize the effect size together with an indicator of precision (the 95% confidence interval, CI), derived from bootstrapping. Compared to parametric methods, this approach is more robust for data sets with non-normal distributions. Author response image 1 shows the mean difference between control (SNAREs alone) and Syt1 in the presence of calcium and PI(4,5)P_2_ (Syt1CaPIP2) is 0.535 pores/min, with 95.0%CI=[0.258, 1.35], whereas the mean difference between control and C2AB in the presence of calcium and PI(4,5)P_2_ (C2ABCaPIP2) is 0.294 pores/min, with 95.0%CI = [0.0685, 0.785]. That is, this approach supports the conclusion that the fusion rate with C2AB in the presence of calcium and PI(4,5)P_2_ is larger than the rate observed with the control group.

**Author response image 1. respfig1:** The mean difference for 8 comparisons against the shared control are shown in the above Cumming estimation plot. The raw data is plotted on the upper axes. On the lower axes, mean differences are plotted as bootstrap sampling distributions. Each mean difference is depicted as a dot. Each 95% confidence interval is indicated by the ends of the vertical error bars. 5000 bootstrap samples were taken; the confidence interval is bias-corrected and accelerated. (see ref. 1).

Our fusion rate data in Figure 2A and other figures should indeed be interpreted with caution, not so much because adjustments for multiple comparisons were omitted in null-hypothesis testing, but because the assay is not optimal for reliably estimating fusion rates, which are not a focus of our study. We require the fusion rate to be low by design in order to detect single events. As the rate increases, the number of events counted does not increase as rapidly, because events that may be overlapping (which increase in frequency with increasing fusion rate) are not counted in the analysis – an issue not captured by the statistical analyses above.

In summary, our interpretation of these results is that in the presence of calcium and PI(4,5)P_2_, Syt1 increases the fusion rate. Fusion is still dependent on SNAREs, as TeNT inhibits fusion. The soluble C2AB domain of Syt1 seems to recapitulate the effects of Syt1 on fusion rates, but we are less confident about whether C2AB really causes an increase in the fusion rate in the presence of its co-factors.

We note that our main conclusions are based on single-pore conductance measurements which are more robust and not affected by our fusion rate estimates.

We have left the pair-wise comparisons as they are, but we added two notes of caution in the text:

last paragraph on p. 5:

“Note that our fusion rate estimates throughout should be interpreted with caution, because they are inherently noisy and they systematically underestimate fusion rates when the rates are high. Both effects are due to the fact that in the assay only a few fusion pores can be analyzed per patch (see SI Methods and ref. (2)).”

last paragraph on p. 6:

“Similar to the results with full-length Syt1, there was little change in the fusion rate compared to the SNARE-alone case if either calcium or exogenous PI(4,5)P_2_ was omitted (Figure 2A). When both calcium (100 μM) and PI(4,5)P_2_ were present, the fusion rate was higher, but we are not as confident about this increase as in the case of Syt1. The mean conductance was significantly above the SNARE-only value in the presence of calcium and PI(4,5)P_2_, but not when either was omitted (Figure 2B).”

Reviewer #3:This study by Wu et al. presents interesting new results obtained with a very sophisticated experimental system and incorporating very sophisticated modeling into their interpretations. The principal results advance the idea that synaptotagmin promotes fusion pore expansion through its interactions with membranes. The idea of a role for synaptotagmin in pore expansion has been around for quite some time but the present results extend what we know about this process. In particular, Syt1 binding to the lipid bilayer to exert force and change fusion pore shape is novel and interesting. All in all, it is a strong paper but a number of concerns require attention.1. When the authors state ~4 copies of syt1 and VAMP2 per disc face (page 4 and 5), this is an average and the actual number will have a wide range due to inherent fluctuations when numbers are small. This variation in copy number must be incorporated into the discussion.

We agree. We have added the following sentence to Discussion:

"Another, possibly related, limitation is that due to the small numbers of proteins that can be incorporated into nanodiscs, large fluctuations are expected in the actual copy numbers from disc-to-disc. Such fluctuations likely contribute to the variability observed in our single-pore measurements, e.g., of mean conductance values. "

2. How do the authors arrive at 10 nm maximum diameter (top of P 5)? There must be a limit in a 25 nm NLP but the actual value is hard to specify, and the 10 nm value seems like a guess.

This is an estimate based on geometrical constraints and measurements of maximum conductance values by Wu et al.^3^. Shi et al.^4^ previously used similar arguments for smaller, MSP-based nanodiscs. A more direct estimate was provided by Bello et al.^5^ who monitored the efflux of encapsulated fluorescent dextrans of different sizes during fusion between v-SNARE reconstituted NLPs and t-SNARE liposomes. They concluded that NLP-liposome fusion pores can reach >9 nm in size. We have added this reference to the relevant passage (top of p. 5).

3. The authors use the Boltzmann distribution to go from the observed pore size distribution to the size dependence of pore energy. This is an impressive leap of insight and creativity. But the Boltzmann distribution applies to systems at equilibrium and the observed conductance time course is probably very far from equilibrium. The underlying process appears to be irreversible. Each pore opening episode looks like a trajectory through a complex energy landscape. Some acknowledgement of these shortcomings must be made.

We agree. We have added ref. Wu et al.3 where a similar approach was used (and the assumptions discussed). We also added the following passage on p. 6:

“Invoking the Boltzmann distribution amounts to assuming the membrane-protein system is approximately in equilibrium, i.e. the conductance measurements are approximately passive and only weakly perturb the fusion pore. We cannot exclude substantial non-equilibrium effects, as application of a potential difference may in itself promote pore formation and affect the structure and dynamics of the pores that result, as seen in lipid bilayer electroporation studies (84), although the potential difference used in our studies is much lower (<20 mV). Generally, the profiles we report should be interpreted as effective free energies.”

4. On P 9 the authors discuss changes in pore radius, height, and shape. Doesn't that complicate the relation between diameter and conductance?

Changes in the shape of the pore do in general change the relation between diameter and conductance. We used a general formula to determine the conductance of the fusion pores predicted by our model, see equation (20) of the SI appendix. This formula makes no assumptions about the pore shape other than it being axially symmetric.

5. The last sentence of the Mathematical modeling section on P 10 states that increasing pore height increases membrane bending energy. Ref 109 shows that increasing height reduces bending energy.

The membrane energy does not depend monotonically on the height of the pore: there is a height that minimizes the energy of the pore, which we assume is selected in the absence of forces associated with the Syt-SNARE complex. Whether a change in the pore height increases or decreases the energy of the pore thus depends on the initial state of the pore. Our statement refers specifically to the situation when the initial condition is the pore that minimizes the bending energy as a function of height; from this state, increasing (or decreasing) the height of the pore will by definition increase the bending energy. In Ref 109 increasing the height of the pore decreases the energy because the initial state of the pore is a compressed state. (In fact it could not be the case that increasing the height of the pore decreases the membrane energy quite generally, since were this true the height of fusion pores would increase without bound.)

6. The final section of Results presents some modeling work with C2A. There are no relevant experiments and the results are not discussed in the Discussion section. I do not see how this section fits into this paper.

We agree with the reviewer that this section is disconnected from the remainder of the paper. We have removed this section from the paper.

References

1. Ho J, Tumkaya T, Aryal S, Choi H, Claridge-Chang A. Moving beyond P values: data analysis with estimation graphics. *Nature Methods***16**, 565-566 (2019).

2. Karatekin E. Toward a unified picture of the exocytotic fusion pore. *FEBS Lett***592**, 3563-3585 (2018).

3. Wu Z*, et al.* Dilation of fusion pores by crowding of SNARE proteins. e*life***6**, e22964 (2017).

4. Shi L*, et al.* SNARE Proteins: One to Fuse and Three to Keep the Nascent Fusion Pore Open. *Science***335**, 1355-1359 (2012).

5. Bello OD, Auclair SM, Rothman JE, Krishnakumar SS. Using ApoE Nanolipoprotein Particles To Analyze SNARE-Induced Fusion Pores. *Langmuir***32**, 3015-3023 (2016).

6. Wang S, Li Y, Ma C. Synaptotagmin-1 C2B domain interacts simultaneously with SNAREs and membranes to promote membrane fusion. e*life***5**, (2016).